# ON THE CONVERGENCE OF MUON AND BEYOND

## ABSTRACT

The Muon optimizer has demonstrated remarkable empirical success in handling matrix-structured parameters for training neural networks. However, a significant gap remains between its practical performance and theoretical understanding. Existing analyses show that the Muon variants achieve only a suboptimal iteration complexity of $\mathcal{O}(T^{-1/4})$ in stochastic non-convex settings, where $T$ denotes the number of iterations. To explore the theoretical limits of the Muon framework, we analyze two Momentum-based Variance-Reduced variants: a one-batch version (Muon-MVR1) and a two-batch version (Muon-MVR2). We provide the first rigorous proof that incorporating variance reduction enables Muon-MVR2 to attain the optimal iteration complexity of $\tilde{\mathcal{O}}(T^{-1/3})$, thereby matching the theoretical lower bound for this class of problems. Furthermore, our analysis establishes last-iterate convergence guarantees for Muon variants under the Polyak-Łojasiewicz (PŁ) condition. Extensive experiments on vision (CIFAR-10) and language (C4) benchmarks corroborate our theoretical findings on per-iteration convergence. Overall, this work offers the first proof of optimality for a Muon-style optimizer and clarifies the path toward developing more practically efficient, accelerated variants.

## 1 INTRODUCTION

The immense computational cost of pre-training Large Language Models (LLMs) has spurred a surge of research into novel optimization methods designed to enhance parameter efficiency and training stability Hoffmann et al. (2022); Liu et al. (2023); Chen et al. (2023); Vyas et al. (2025); Pethick et al. (2025); Yuan et al. (2024). Among these, methods based on matrix orthogonalization have recently garnered significant attention from both academia and industry Jordan al. (2024); Liu et al. (2025a). In particular, the Muon optimizer has emerged as a notable milestone due to its impressive empirical performance Liu et al. (2025a); An et al. (2025); Liu et al. (2025b); Shah et al. (2025). However, despite its practical success, the theoretical understanding of Muon's underlying mechanisms has surprisingly lagged behind, with existing convergence analyses being fraught with limitations and even critical fallacies.

Specifically, the current theoretical exploration of Muon's convergence faces three primary obstacles. First, existing analyses have not established convergence to a stationary point without reliance on problem dimension or batch size; the available bounds become valid only when the batch size is sufficiently large, so the basic requirement for a trustworthy optimizer remains unfulfilled Sato et al. (2025); Sfyraki & Wang (2025). Second, some analyses are predicated on flawed mathematical assumptions, such as the erroneous use of the inequality $\|\mathbf{S}^{-1}\|_2 \leq 1/\|\mathbf{S}\|_2$ in a key part of their proof, which casts serious doubt on the validity of their convergence claims Li & Hong (2025). Finally, the most rigorous existing work analyzes both the standard Muon algorithm (like Algorithm 1, Option MVR1 with $\gamma = 0$) and its simplified variant , considering their convergence to a non-standard $\epsilon$-nuclear norm stationary point. This leaves the behavior of the Nesterov-Accelerated Muon algorithm (like Algorithm 1, Option MVR1), as discussed in Liu et al. (2025a); Sato et al. (2025), and Variance-Reduction Muon (like Algorithm 1, Option MVR2) under standard settings as an open question Shen et al. (2025).

To bridge this critical theoretical gap, this work establishes a rigorous and complete theoretical foundation for the Muon optimizer. Our main contributions are threefold:

- For general non-convex settings, we provide a convergence analysis for both the standard Muon (Algorithm 1, Option MVR1 with $\gamma = 0$) and the Muon-MVR1 algorithm (Algorithm 1, Option MVR1).

- Regarding iteration complexity, we provide, to the best of our knowledge, the first analysis showing that, in an unconstrained Muon-style setting, the variance-reduced Muon-MVR2 algorithm (Algorithm 1, Option MVR2) attains the $\tilde{\mathcal{O}}(T^{-1/3})$ convergence rate, matching the best-known complexity for variance-reduced momentum methods.

- Under the Polyak-Łojasiewicz (PL) condition, we prove that our proposed algorithms, Muon-MVR1, and Muon-MVR2, all achieve sublinear convergence rates. More precisely, we demonstrate that Muon-MVR1 converge at a rate of $\tilde{\mathcal{O}}(T^{-1/2})$, and Muon-MVR2 converges at an accelerated rate of $\tilde{\mathcal{O}}(T^{-2/3})$

Table 1 summarizes the main contributions of our work and compares them with existing methods.

Table 1: Comparison of Existing Muon-type Analyses with Ours.

|  | Smooth[a] | Stoc. Gradient Estimator [b] | Batch Size | Iteration Complexity | Last-iterate Conv. Rate |
|---|---|---|---|---|---|
| Li & Hong (2025) | $L$ | MVR1($\gamma = 0$) | $\mathcal{O}(1)$ | $\mathcal{O}(T^{-1/4})$ | ✗ |
| Sato et al. (2025) | $L$ | MVR1($\gamma = 0$) | $\mathcal{O}(\epsilon^{-1})$ | $\mathcal{O}(T^{-1}) + \mathcal{O}(1)^c$ | ✗ |
| Sato et al. (2025) | $L$ | MVR1 | $\mathcal{O}(\epsilon^{-1})$ | $\mathcal{O}(T^{-1}) + \mathcal{O}(1)^c$ | ✗ |
| Sfyraki & Wang (2025) | $L_+$ | **MVR2** | $\mathcal{O}(\epsilon^{-1})^d$ | $\mathcal{O}(T^{-1/3})$ | ✗ |
| Shen et al. (2025) | $L$ | MVR1($\gamma = 0$) | $\mathcal{O}(1)$ | $\mathcal{O}(T^{-1/4})$ | ✗ |
| Ours | $L$ | MVR1($\gamma = 0$) | $\mathcal{O}(1)$ | $\tilde{\mathcal{O}}(T^{-1/4})$ | $\tilde{\mathcal{O}}(T^{-1/2})$ |
| Ours | $L$ | MVR1 | $\mathcal{O}(1)$ | $\tilde{\mathcal{O}}(T^{-1/4})$ | $\tilde{\mathcal{O}}(T^{-1/2})$ |
| Ours | $L_+$ | **MVR2** | $\mathcal{O}(1)$ | $\tilde{\mathcal{O}}(T^{-1/3})$ | $\tilde{\mathcal{O}}(T^{-2/3})$ |

[a] $L$ pertains to the smoothness of the overall function $f$, while $L_+$ pertains to the smoothness of its stochastic components $f(\cdot; \xi)$.
[b] Option MVR1 ($\gamma = 0$) is the standard momentum method, Option MVR1 is the one-batch variance-reduction momentum method, and Option MVR2 is the two-batch variance-reduction momentum method. We summarize them in Algorithm 1.
[c] Although increasing the batch size can mitigate the impact of stochastic noise Sato et al. (2025), these methods still fail to converge to a stationary point and cannot eliminate the influence of dimensionality.
[d] The results from Sfyraki & Wang (2025) on Option MVR2 are the closest to ours. However, their method requires a large initial batch of size $\mathcal{O}(\epsilon^{-1})$, although the batch size can be reduced to 1 in subsequent iterations.

• **Organizations**. The rest of the paper is organized as follows. Section 2 reviews existing variants of the Muon algorithm. Section 3 addresses iteration complexity and last-iterate convergence rate. Section 5 presents experimental results demonstrating the effectiveness of our method. Related work is reviewed in Section 6, and conclusions are drawn in Section 7.

• **Notations**. We denote scalars by non-bold letters (e.g., $a, A$), vectors in $\mathbb{R}^d$ by bold lowercase letters (e.g., $\mathbf{a}$), and matrices by bold uppercase letters (e.g., $\mathbf{A}$). The space $\mathbb{R}^d$ is endowed with the Euclidean inner product $\langle \mathbf{x}, \mathbf{y} \rangle_2 := \mathbf{x}^\top \mathbf{y}$ and norm $\|\mathbf{x}\|_2$. For matrices, we employ the Frobenius inner product $\langle \mathbf{A}, \mathbf{B} \rangle_F := \mathrm{tr}(\mathbf{A}^\top \mathbf{B})$ and the corresponding norm $\|\mathbf{A}\|_F$. The nuclear norm, denoted by $\|\mathbf{A}\|_*$, is defined as the sum of the singular values of the matrix, $\|\mathbf{A}\|_* = \sum_i \sigma_i(\mathbf{A})$. Throughout the paper, $[m]$ denotes the set of integers $\{1, 2, \ldots, m\}$, and $\mathbb{N}$ denotes the set of non-negative integers. The model is parameterized by a matrix $\mathbf{X} \in \mathbb{R}^{m \times n}$. Without loss of generality, we assume $m \geq n$, so the rank of the matrix is at most $n$. The model is optimized by minimizing the empirical loss function $f(\mathbf{X}) := \frac{1}{N} \sum_{i \in [N]} f_i(\mathbf{X})$, where $N$ is the number of training data points, and $f_i(\mathbf{X})$ is the loss function for $\mathbf{X} \in \mathbb{R}^{m \times n}$ with respect to the $i$-th training data point $\mathbf{z}_i$ (for $i \in [N]$). Let $\xi$ be a random variable that is independent of $\mathbf{X} \in \mathbb{R}^{m \times n}$, and let $\mathbb{E}_\xi[\mathbf{X}]$ denote the expectation of a random variable $\mathbf{X}$ with respect to $\xi$.

## 2 REVISITING THE MUON ALGORITHMS

We consider the following optimization problem:

$$\min_{\mathbf{X} \in \mathbb{R}^{m \times n}} f(\mathbf{X}), \text{ where } f(\mathbf{X}) = \mathbb{E}_{\xi \sim \mathcal{D}}[f(\mathbf{X}; \xi)], \tag{1}$$

where $f : \mathbb{R}^{m \times n} \to \mathbb{R}$ is the loss function, $\mathbf{X}$ denotes the decision variable, and $\xi$ represents a random variable(e.g., a training data sample) drawn from an unknown distribution $\mathcal{D}$. We assume that $f$ is differentiable and possibly nonconvex.

The Muon optimizer begins by computing a momentum-based variance-reduced gradient update, similar in spirit to ADAM Kingma & Ba (2014), STORM Cutkosky & Orabona (2019), and SGD with Nesterov momentum Sutskever et al. (2013). The momentum term $\mathbf{M}_t$ is then projected onto the set of orthogonal matrices. This orthogonalization step equalizes the singular values, ensuring that no principal component direction dominates the optimization. Finally, the resulting scaled orthogonal matrix is used to update the model parameters. The Muon algorithm is summarized in Algorithm 1.

---

**Algorithm 1** Muon-style Algorithm

---

1: **Input:** Initial parameters $\mathbf{X}_0 \in \mathbb{R}^{m \times n}$, learning rate $\eta_t > 0$, momentum parameter $\beta_t \in [0, 1)$, variance-reduction parameter $\gamma \in [0, 1]$, initial momentum $\mathbf{M}_0 = 0$, $\nabla f(\mathbf{X}_0; \xi) = 0$.
2: **for** $t = 1$ **to** $T$ **do**
3:     Compute stochastic gradient: $\nabla f(\mathbf{X}_t; \xi_t)$
4:     Option MVR1: One-batch Momentum Variance-Reduction (MVR1)
5:     $\mathbf{M}_t = \beta_t \mathbf{M}_{t-1} + (1 - \beta_t) \nabla f(\mathbf{X}_t; \xi_t) + \gamma \cdot \beta_t \cdot (\nabla f(\mathbf{X}_t; \xi_t) - \nabla f(\mathbf{X}_{t-1}; \xi_{t-1}))$
6:     Option MVR2: Two-batch Momentum Variance-Reduction (MVR2)
7:     $\mathbf{M}_t = \beta_t \mathbf{M}_{t-1} + (1 - \beta_t) \nabla f(\mathbf{X}_t; \xi_t) + \gamma \cdot \beta_t \cdot (\nabla f(\mathbf{X}_t; \xi_t) - \nabla f(\mathbf{X}_{t-1}; \xi_t))$
8:     $\mathbf{O}_t \in \arg\min_{\mathbf{O}} \|\mathbf{O} - \mathbf{M}_t\|_F$, s.t. $\mathbf{O}^\top \mathbf{O} = \mathbf{I}_n$
9:     $\mathbf{X}_{t+1} = \mathbf{X}_t - \eta_t \mathbf{O}_t$
10: **end for**
11: **Output:** Final parameters $\mathbf{X}_t$

---

This orthogonalization step can be obtained from the Singular Value Decomposition (SVD) of $\mathbf{M}_t$:

$$\mathbf{M}_t = \mathbf{U}\mathbf{\Sigma}\mathbf{V}^\top, \qquad \mathbf{O}_t = \mathbf{U}\mathbf{V}^\top.$$

Equivalently, we have $\mathbf{O}_t = (\mathbf{M}_t \mathbf{M}_t^\top)^{-1/2} \mathbf{M}_t$, which shows that orthogonalization reduces to computing an inverse square root rather than a full SVD. Since forming and decomposing $\mathbf{M}_t \mathbf{M}_t^\top$ remains costly, the popular Muon implementation Jordan et al. (2024) uses the quintic Newton–Schulz iteration to approximate the inverse square root. This recurrence converges in only a few steps (typically five), producing a numerically stable, rank-preserving orthogonalization of $\mathbf{M}_t$ that is nearly as accurate as SVD but far more efficient.

Algorithm 1 incorporates two distinct strategies for momentum-based variance reduction, termed Muon-MVR1 and Muon-MVR2. These options present a fundamental trade-off between computational efficiency and theoretical rigor. MVR2 implements a principled variance reduction scheme at the cost of two gradient evaluations per step, while MVR1 serves as a computationally cheaper, single-gradient approximation. We detail both below.

▶ **Option 1: Muon-MVR1 (One-batch Approximation).** The first option, MVR1, augments the classical momentum update with a variance-reducing term that reuses the gradient from the previous step:

$$\mathbf{M}_t = \beta_t \mathbf{M}_{t-1} + (1 - \beta_t) \nabla f(\mathbf{X}_t; \xi_t) + \gamma \cdot \beta_t \cdot (\nabla f(\mathbf{X}_t; \xi_t) - \nabla f(\mathbf{X}_{t-1}; \xi_{t-1})). \tag{2}$$

The primary advantage of this formulation is its computational efficiency, as it requires only one stochastic gradient evaluation per iteration. This update rule is flexible:

*(i)* When $\gamma = 0$, Rule (2) degenerates to the standard exponential moving average (EMA) of gradients (Rule (3)), a stochastic gradient estimator widely used in optimizers like Adam Kingma & Ba (2014).

$$\mathbf{M}_t = \beta_t \mathbf{M}_{t-1} + (1 - \beta_t) \nabla f(\mathbf{X}_t; \xi_t). \tag{3}$$

*(ii)* By setting $\beta_t = \mu$ and $\gamma = 1 - \mu$, the update rule (2) yields a momentum term that, after rescaling by $1/(1 - \mu)$, satisfies the recurrence $\tilde{\mathbf{M}}_t = \mu \tilde{\mathbf{M}}_{t-1} + \nabla f(\mathbf{X}_t; \xi_t) + \mu(\nabla f(\mathbf{X}_t; \xi_t) - \nabla f(\mathbf{X}_{t-1}; \xi_{t-1}))$. This form is algebraically equivalent to the standard Muon optimizer Jordan et al. (2024); Liu et al. (2025a) derived from Eq. (4) Yuan et al. (2024), and inherently implements Nesterov acceleration via a first-order Taylor approximation of the gradient Xie et al. (2024).

$$\begin{aligned}
\mathbf{C}_t &= \mu \mathbf{C}_{t-1} + \nabla f(\mathbf{X}_t; \xi_t), \\
\mathbf{M}_t &= \mu \mathbf{C}_t + \nabla f(\mathbf{X}_t; \xi_t).
\end{aligned} \tag{4}$$

▶ **Option 2: Muon-MVR2 (Two-batch Principled VR).** The second option, MVR2, incorporates a more rigorous variance-reduction mechanism inspired by methods like SPIDER Fang et al. (2018), STORM Cutkosky & Orabona (2019), SVRG Zhou et al. (2020), SUPER-Adam Huang et al. (2021), and MARS Yuan et al. (2024):

$$\mathbf{M}_t = \beta_t \mathbf{M}_{t-1} + (1 - \beta_t) \nabla f(\mathbf{X}_t; \xi_t) + \gamma \cdot \beta_t \cdot (\nabla f(\mathbf{X}_t; \xi_t) - \nabla f(\mathbf{X}_{t-1}; \xi_t)). \tag{5}$$

The key distinction from MVR1 is the correction term. MVR2 subtracts the gradient computed on the previous parameters but with the current mini-batch, i.e., $\nabla f(\mathbf{X}_{t-1}; \xi_t)$. This modification is crucial as it is designed to directly cancel the variance introduced by the mini-batch $\xi_t$ Cutkosky & Orabona (2019); Huang et al. (2021); Yuan et al. (2024). However, this theoretical benefit comes at the cost of requiring two gradient evaluations per step. MVR1 can be formally understood as a practical approximation of MVR2. The difference between their update rules is a single noise term, $\Delta_{t-1}^{\text{Noise}} = \nabla f(\mathbf{X}_{t-1}; \xi_t) - \nabla f(\mathbf{X}_{t-1}; \xi_{t-1})$. Under the standard assumption of bounded variance ($\mathbb{E}_\xi[\|\nabla f(\mathbf{X}; \xi) - \nabla f(\mathbf{X})\|_F^2] \leq \sigma^2$), the variance of this noise is well-controlled, satisfying $\mathbb{E}[\|\Delta_{t-1}^{\text{Noise}}\|_F^2] \leq 2\sigma^2$. While MVR1 is often sufficient in practice, this structural difference leads to fundamentally different theoretical guarantees. The principled variance cancellation in MVR2 allows our algorithm to achieve a superior iteration complexity of $\tilde{\mathcal{O}}(T^{-1/3})$, as we will formally establish in Theorem 3.2.

## 3 CONVERGENCE ANALYSIS

We begin our analysis by situating it within the established context of first-order stochastic optimization. The $\mathcal{O}(T^{-1/4})$ iteration complexity is a well-known bottleneck for methods like SGD, and recent work by Shen et al. (2025) confirmed this limitation for the standard Muon algorithm. Accordingly, our first step in this section is to analyze the Muon-MVR1 variant and formally reaffirm this baseline iteration complexity in Theorem 3.1. While the rate itself is standard, our primary contribution here lies in the novel and unified analytical framework we develop to prove it, as this framework will be instrumental for subsequent results.

Our analysis is conducted under a diminishing stepsize schedule, a standard setting that guarantees convergence to an optimal solution. Nevertheless, our theoretical framework is general enough to encompass the constant-stepsize setting, where it yields a tighter convergence bound devoid of the logarithmic factor $\ln(T)$.

To facilitate the analysis of convergence for Muon, we make the following assumptions:

**Assumption 3.1.** *The function $f$ is bounded from below. There exists $f^* > -\infty$ such that $f(\mathbf{X}) \geq f^*$, for all $\mathbf{X} \in \mathbb{R}^{m \times n}$.*

**Assumption 3.2.** *The function $f$ is $L$-smooth: $\|\nabla f(\mathbf{Y}) - \nabla f(\mathbf{X})\|_F \leq L\|\mathbf{Y} - \mathbf{X}\|_F$.*

**Assumption 3.3.** *The function $f$ is $L$-smooth for any $\xi$: $\|\nabla f(\mathbf{Y}; \xi) - \nabla f(\mathbf{X}; \xi)\|_F \leq L\|\mathbf{Y} - \mathbf{X}\|_F$.*

**Assumption 3.4.** *The variance of unbiased stochastic gradient is finite. Specifically, there exists a constant $\sigma > 0$ such that for all $\mathbf{X} \in \mathbb{R}^{m \times n}$, the following holds: $\mathbb{E}[\nabla f(\mathbf{X}; \xi)] = \nabla f(\mathbf{X})$ and $\mathbb{E}\|\nabla f(\mathbf{X}; \xi) - \nabla f(\mathbf{X})\|_F^2 \leq \sigma^2$.*

These assumptions are quite common Zhou et al. (2018); Chen et al. (2018); Huang et al. (2021); Guo et al. (2021); Li et al. (2023); Wang et al. (2023); Xie et al. (2024); Yuan et al. (2024).

### 3.1 EROGDIC CONVERGENCE OF MUON

In this subsection, we establish the ergodic convergence of Muon-MVR1 and Muon-MVR2.

### 3.1.1 OPTION MVR1

We begin our analysis with Option MVR1, a straightforward implementation of one-batch momentum-based variance reduction. The following theorem establishes its ergodic convergence rate, demonstrating that the algorithm converges to a stationary point at a rate of $\tilde{\mathcal{O}}(T^{-1/4})$ for specific choices of learning rate and momentum schedules.

**Theorem 3.1.** *Suppose Assumptions 3.1, 3.2, and 3.4 hold. Consider Algorithm 1 with a learning rate of $\eta_t = t^{-3/4}$. The expected average squared norm of the gradient is bounded for the following options:*

1. *For **Option MVR1** ($\gamma = 0$), with momentum parameter $\beta_t = 1 - t^{-1/2}$, the bound is given by:*

$$\frac{1}{T}\sum_{t=1}^{T}\mathbb{E}\|\nabla f(\mathbf{X}_t)\|_F \leq \frac{f(\mathbf{X}_1) - f^*}{T^{1/4}} + \frac{A_1 \ln T + A_2}{T^{1/4}},$$

   *where $A_1 = 2L^{-1}\sigma^2 + 4\sqrt{2}Ln + Ln + L/2$ and $A_2 = 4L^{-1}\sigma^2 + 4\sqrt{2}Ln + Ln + L/2$.*

2. *For **Option MVR1** ($\gamma_t = t^{-1/2}$), with momentum parameter $\beta_t = 1 - (t+1)^{-1/2}$, the bound is given by:*

$$\frac{1}{T}\sum_{t=1}^{T}\mathbb{E}\|\nabla f(\mathbf{X}_t)\|_F \leq \frac{f(\mathbf{X}_1) - f^*}{T^{1/4}} + \frac{A_1 \ln T + A_2}{T^{1/4}},$$

   *where $A_1 = 4L^{-1}\sigma^2 + 8\sqrt{2}Ln + Ln + L/2$ and $A_2 = 10L^{-1}\sigma^2 + 8\sqrt{2}Ln + Ln + L/2$.*

*See Appendix B for details.*

**Remark 3.1.** *Theorems 3.1 establishes that as $T \to \infty$, the leading terms diminish to zero, and the algorithm converges to a neighborhood of a stationary point. The size of this neighborhood is determined by the stochastic gradient variance, the learning rate, and the momentum parameter. Notably, this convergence bound is free of any non-vanishing additive error term that depends on the dimension $n$ Sato et al. (2025). We absorb the dimensional dependence into the $\mathcal{O}$-notation to define $\tilde{\mathcal{O}}$, which yields the following complexity for the algorithm:*

$$\min_{t=1,\dots,T}\mathbb{E}\|\nabla f(\mathbf{X}_t)\|_F \leq \tilde{\mathcal{O}}(T^{-1/4}).$$

### 3.1.2 OPTION MVR2

While prior work has established a iteration complexity of $\tilde{\mathcal{O}}(T^{-1/3})$ for variance-reduction in non-convex settings for various algorithms, such as SGD Fang et al. (2018); Cutkosky & Orabona (2019); Zhou et al. (2020) and Adam Huang et al. (2021); Yuan et al. (2024), the theoretical convergence properties of this technique when applied to the Muon optimizer have remained an open question. We bridge this theoretical gap in Theorem 3.2, which rigorously proves that Option MVR2 of Algorithm 1 (Muon-MVR2) achieves the same $\tilde{\mathcal{O}}(T^{-1/3})$ iteration complexity in the general non-convex setting.

**Theorem 3.2.** *Under Assumptions 3.1, 3.3, and 3.4, consider Algorithm 1 with **Option MVR2** and update rule (5) parameters set as $\beta_t = 1 - \eta_t$, $\eta_t = t^{-2/3}$ for $t \geq 1$, and $\gamma = 1$. Then, the following bounds hold:*

$$\frac{1}{T}\sum_{t=1}^{T}\mathbb{E}\|\nabla f(\mathbf{X}_t)\|_F \leq \frac{\sqrt{4\sigma^2 + (32L^2n + 8\sigma^2)(1 + \ln T)}}{T^{1/3}} + \frac{G}{T^{1/3}},$$

*where*

$$G = f(\mathbf{X}_1) - f^* + 2\sigma^2 L^{-1} + 2Ln + \left(16Ln + 4\sigma^2 L^{-1} + \frac{Ln}{2}\right)(1 + \ln T).$$

*See Appendix D for details.*

**Remark 3.2.** *Theorem 3.2 shows that, for an unconstrained Muon-style algorithm with momentum-based variance reduction, we can match the current state-of-the-art $\tilde{\mathcal{O}}(T^{-1/3})$ iteration complexity in a practical fixed mini-batch setting. This complements prior work Sfyraki & Wang (2025), which achieves the same rate using a growing batch size $b = \mathcal{O}(T^{1/3})$ to control gradient variance. This is achieved by a specific hyperparameter schedule in which the learning rate $\eta_t = t^{-2/3}$ and the momentum parameter $\beta_t = 1 - \eta_t$ are tightly coupled. This schedule balances optimization progress with control of the stochastic gradient variance.*

**Remark 3.3.** *As noted in Yuan et al. (2024), a more sophisticated, adaptive setting for the variance-reduction parameter $\gamma$ can be employed. Specifically, by setting $\gamma = \gamma_t = 1 - \frac{A_t}{\beta_t}$, where $A_t$ is defined in Lemma C.2, a key term in the analysis becomes $P_t = -\mathbb{E}\|\nabla f(\mathbf{X}_t; \xi_t) - \nabla f(\mathbf{X}_{t-1}; \xi_t)\|_F^2 \cdot A_t^2$, which is strictly negative. This leads to a tighter convergence bound, as it effectively introduces an additional beneficial term into the recurrence. However, computing this adaptive $\gamma_t$ is often impractical as it depends on quantities that are difficult to estimate during training. Consequently, we adhere to the common and more practical approach of using a constant $\gamma \leq 1$.*

**Remark 3.4.** *For **Option MVR1**, the prefactor in the $\tilde{\mathcal{O}}(T^{-1/4})$ bound depends linearly on the noise variance and the dimension, i.e., it is of order $L^{-1}\sigma^2 + Ln + L$, so there is no super-linear growth in $n$. For **Option MVR2**, the leading stochastic term behaves like $\sigma + L\sqrt{n}$ (up to a $\sqrt{1 + \ln T}$ factor), while the remaining constant term scales as $f(\mathbf{X}_1) - f^* + L^{-1}\sigma^2 + Ln$, which reveals a mixed $\sqrt{n}$- and $n$-dependence. In contrast to standard parameter-agnostic complexity results for variance-reduced SGD and adaptive methods, which typically hide the dependence on $L$, $\sigma$, and $n$ inside the $\tilde{\mathcal{O}}(\cdot)$ notation (see, e.g., Fang et al. (2018); Cutkosky & Orabona (2019); Zhou et al. (2020); Huang et al. (2021); Yuan et al. (2024)), our analysis keeps this structure explicit and highlights how the Muon geometry interacts with variance reduction in the matrix-valued setting.*

## 3.2 NON-ERGODIC CONVERGENCE OF MUON

In this subsection, we examine the performance of Muon-MVR1 and Muon-MVR2 under the setting where the non-convex objective functions satisfy the Polyak-Łojasiewicz (PL) condition. Our analysis is based on the following additional assumption:

**Assumption 3.5.** *We assume the function $f$ is $\mu$-PL, i.e., $\|\nabla f(\mathbf{X}_t)\|_F^2 \geq 2\mu(f(\mathbf{X}_t) - f^*)$.*

**Remark 3.5.** *The PL condition has been widely employed in the convergence analysis of various first-order algorithms Karimi et al. (2016); Xie et al. (2020a); Li & Li (2022), though typically under restricted settings. Note that if $f(\cdot)$ is strongly convex, then it is necessarily convex and satisfies the PL condition. However, the converse does not hold in general; a counterexample is given by $f(x) = x^2 + 3\sin(2x)$.*

**Theorem 3.3.** *Suppose Assumptions 3.1, 3.2, 3.4, and 3.5 hold. Let $\{\mathbf{X}_t\}$ be the sequence of iterates generated by Algorithm 1 with a step size of $\eta_t = t^{-3/4}$. We analyze the following two MVR1 schemes:*

1. *Scheme 1 ($\gamma = 0$): Using the update rule (3) with momentum $\beta_t = 1 - t^{-1/2}$. For this case, we define the constant $\mathcal{A}_1$ as:*

$$\mathcal{A}_1 = 6L^{-1}\sigma^2 + (8\sqrt{2} + 2)Ln + L.$$

2. *Scheme 2 ($\gamma_t = t^{-1/2}$): Using the update rule (2) with parameters $\beta_t = 1 - (t+1)^{-1/2}$. For this case, we define the constant $\mathcal{A}_2$ as:*

$$\mathcal{A}_2 = 14L^{-1}\sigma^2 + (16\sqrt{2} + 2)Ln + L.$$

*Then, for either scheme $i \in \{1, 2\}$, there exists a constant $T_0 = 2e^4$ such that for all iterations $T \geq T_0$, the expected suboptimality gap is bounded as follows:*

$$\mathbb{E}[f(\mathbf{X}_{T+1})] - f^* \leq \frac{\mathcal{A}_i^2}{8\mu} \cdot \frac{(\ln T)^2}{T^{1/2}}.$$

**Theorem 3.4.** *Suppose Assumptions 3.1, 3.2, 3.4, and 3.5 hold. Let $\{\mathbf{X}_t\}$ be the sequence of iterates generated by Algorithm 1 with **Option MVR2** (see update rule (5)). By setting the parameters to $\eta_t = t^{-2/3}$, $\beta_t = 1 - t^{-2/3}$, and $\gamma = 1$, we define the constant $\mathcal{A}_3$ as :*

$$\mathcal{A}_3 = 20L^{-1}\sigma^2 + 66Ln + L.$$

*Then, there exists a constant $T_0 = 2e^3$ such that for all iterations $T \geq T_0$, the expected suboptimality gap is bounded as follows:*

$$\mathbb{E}[f(\mathbf{X}_{T+1})] - f^* \leq \frac{2\mathcal{A}_3^2}{9\mu} \cdot \frac{(\ln T)^2}{T^{2/3}}.$$

Detailed proofs for Theorems 3.3 and 3.4 are provided in Appendix F and G, respectively.

**Remark 3.6.** *Our non-ergodic convergence proofs are unified by a recursive inequality on the potential function $\Delta_t = \mathbb{E}[f(\mathbf{X}_t)] - f^*$:*

$$\Delta_{t+1} \leq \Delta_t - \frac{\sqrt{2\mu}}{t^p}\sqrt{\Delta_t} + \Gamma_t,$$

*where the error terms $\Gamma_t$ satisfy $\sum_{i=1}^t \Gamma_i \leq \mathcal{O}(\ln t)$, and $\Gamma_t$ is defined in the Appendix, equations (20) and (21). The exponent $p \in (0,1)$ dictates the final convergence rate. For Theorem 3.3, we establish this inequality with $p = 3/4$, which leads to a rate of $\tilde{\mathcal{O}}(T^{-1/2})$. The superior variance control in Theorem 3.4 enables a tighter analysis with $p = 2/3$, which in turn yields the accelerated convergence rate of $\tilde{\mathcal{O}}(T^{-2/3})$.*

## 4 CLARIFICATION OF THEORETICAL NOVELTY

▶ **Relation to concurrent work.** Our work is distinct from concurrent studies Sfyraki & Wang (2025); Kovalev (2025) in setting and scope. Sfyraki & Wang (2025) focuses on compact solution sets using a stochastic Frank–Wolfe scheme with constant step sizes; in contrast, we analyze the original Muon update without compactness assumptions, employing decaying step sizes to establish gradient-norm guarantees. Similarly, while Kovalev (2025) provides a general non-Euclidean trust-region framework yielding SGD-type $\mathcal{O}(\varepsilon^{-4})$ complexity, we exploit the specific Muon structure with variance reduction (MVR1/MVR2). This specialized analysis improves the rate to $\mathcal{O}(\varepsilon^{-3})$ under the same stochastic assumptions and establishes PL-type non-ergodic guarantees distinct from the general framework.

▶ **Relation to SGDM-style analyses.** While related to normalization-based gradient-scaling methods Cutkosky & Mehta (2020; 2021); Chen et al. (2023), our analysis differs in three key aspects. *(i)* **Step sizes**: Unlike normalized SGDM Cutkosky & Mehta (2020) which typically uses constant or $\mathcal{O}(T^{-1/2})$ steps, MVR1/MVR2 employ decaying step sizes $\eta_t = \Theta(t^{-3/4})$ and $\Theta(t^{-2/3})$. This yields a sharper nonconvex complexity of $\tilde{\mathcal{O}}(\varepsilon^{-3})$ compared to the standard $\tilde{\mathcal{O}}(\varepsilon^{-4})$. *(ii)* **Manifold Optimization**: We analyze matrix-valued Muon on the Stiefel manifold using a practical two-batch MVR scheme, proving optimal order convergence with constant mini-batches—a setting not covered by prior Euclidean MVR or parameter-free results Yang et al. (2023). *(iii)* **PL Analysis**: Under the PL condition, we establish non-ergodic convergence via a novel recursion involving $\sqrt{\Delta_t}$ rather than the standard linear form, necessitating a tailored analysis beyond existing techniques.

## 5 EXPERIMENTS

In this section, we evaluate the performance of the Muon-variant optimizers on pretraining tasks. All experiments were conducted using 48x Ascend 910C (64GB) NPUs and 4x NVIDIA RTX 4090 (24GB) GPUs. Detailed experimental settings are provided in Appendix H.

▶ **ResNet18 on CIFAR10 Dataset.** We train ResNet-18 He et al. (2016) on CIFAR-10 for 100 epochs (batch size 128), comparing Muon variants against SGD and Adam over five random seeds. For each optimizer, the learning rate is tuned via grid search over $\{1e-4, 5e-4, 1e-3, 5e-3, 1e-2, 5e-2, 1e-1\}$. As shown in Figures 1a and 1b, Muon variants demonstrate faster initial convergence and lower final test error than the baselines, with Muon-MVR2 achieving the best overall performance.

▶ **LLaMA2 on C4 Dataset.** We pre-train LLaMA2-130M Touvron et al. (2023) on C4 to benchmark Muon-MVR variants against AdamW and MARS-AdamW. We perform a grid search over learning rates $\{3e-4, 5e-4, 8e-4, 1e-3, 2e-3, 4e-3, 6e-3, 8e-3\}$ and, for MARS-AdamW, Muon-MVR1, and Muon-MVR2, over the gamma parameter $\gamma \in \{0.01, 0.025, 0.05\}$. Models are

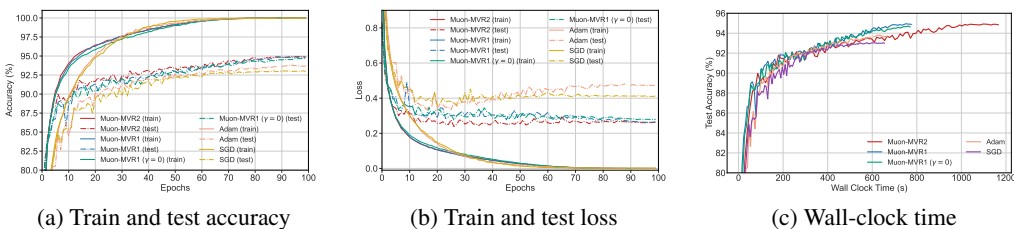

(a) Train and test accuracy     (b) Train and test loss     (c) Wall-clock time

Figure 1: Training dynamics of Muon-MVR2, Muon-MVR1, Muon-MVR1 ($\gamma = 0$), and AdamW on CIFAR-10 with ResNet-18. The plots show (a) accuracy and (b) loss versus epochs for both training and testing, along with (c) test accuracy versus wall-clock time.

trained for 20k steps (∼12B tokens); refer to Appendix H for full details. Figure 2 shows that while Muon-MVR2 achieves the lowest per-step loss, it doubles the wall-clock time. Consequently, we prioritize the highly efficient Muon-MVR1 in subsequent experiments, as it attains comparable performance despite the theoretical complexity gap.

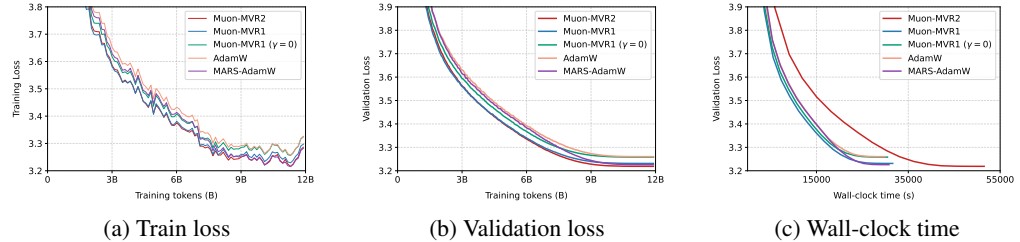

(a) Train loss     (b) Validation loss     (c) Wall-clock time

Figure 2: LLaMA2-130M train and validation curves on C4 Dataset Training dynamics of Muon-MVR2, Muon-MVR1, Muon-MVR1 ($\gamma = 0$), MARS-AdamW, and AdamW on C4 Dataset with LLaMA2-130M. The plots show (a) train loss and (b) validation loss, along with (c) validation loss versus wall-clock time using 8x Ascend 910C.

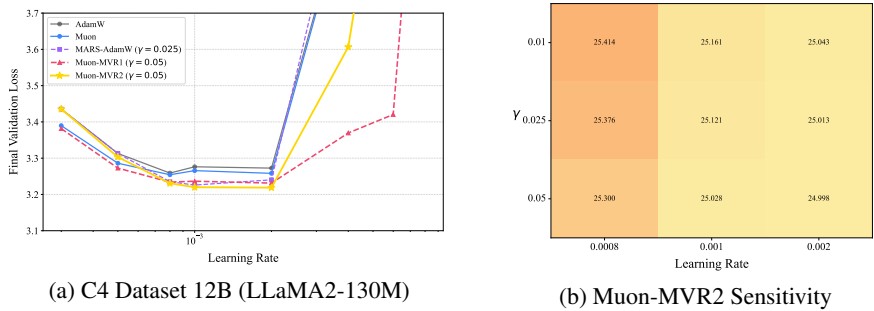

(a) C4 Dataset 12B (LLaMA2-130M)     (b) Muon-MVR2 Sensitivity

Figure 3: (a) Final validation loss with varying learning rates on C4 Dataset 12B ; (b) Heatmap of the final validation perplexity of the Muon-MVR2 model for different $\gamma$ values around the optimal learning rate.

Figure 3a reports the final validation loss on C4 Dataset 12B across different learning rates for all optimizers. Each method exhibits a reasonably wide range of stable learning rates, with Muon-type optimizers achieving lower validation loss than AdamW at their respective best settings. For Muon-MVR2, we additionally sweep the algorithmic parameter $\gamma$ around the optimal learning rate and visualize the resulting validation perplexity as a heatmap in Figure 3b. The heatmap shows that Muon-MVR2 is relatively insensitive to the choice of $\gamma$ in a neighborhood of the best learning rate, suggesting that $\gamma$ does not require fine-grained tuning in practice.

**Remark 5.1.** *The precise variance-reduction formulation Muon-MVR2 (Eq. 5) is more closely aligned with stochastic optimization theory and, as confirmed by our experiments, consistently attains the highest validation accuracy. However, it requires two gradient evaluations per step and can be prohibitive in large-scale settings (see Fig. 1c and 2c). In regimes where the cost difference between one and two mini-batch gradient evaluations is negligible, we therefore recommend Muon-MVR2; when computational efficiency is the primary concern, the approximate Muon-MVR1 variant offers a pragmatic alternative with only minimal performance degradation. Thus, the two variants serve complementary roles rather than one uniformly dominating the other.*

**Remark 5.2.** *We employ standard weight decay in our experiments, following common practice in deep learning. For clarity, this regularization term is omitted from the theoretical analysis, but the same analysis framework naturally extends to this setting.*

# 6 RELATED WORK

## 6.1 THE EVOLUTION OF OPTIMIZATION ALGORITHMS

The advancement of deep learning relies on first-order optimization methods. Nesterov (1983) proposed the Momentum method, which leverages historical gradients to accelerate convergence. Subsequently, Hinton et al. (2012) introduced RMSprop, enabling per-parameter adaptive learning rates. Kingma & Ba (2014) then integrated these ideas in Adam, an optimizer that adapts using the first and second moments of the gradients, establishing it as a standard choice in the field.

Numerous variants have been proposed to address the limitations of Adam. Reddi et al. (2018) introduced AMSGrad to ensure a non-increasing learning rate, while Dozat (2016) created NAdam by incorporating Nesterov momentum. To improve regularization, Loshchilov (2017) developed AdamW, which decouples weight decay from the optimization step. Other methods focus on controlling the learning rate and variance: Luo et al. (2019) proposed AdaBound to dynamically clip learning rates; Liu et al. (2019) designed RAdam to rectify variance estimates in early training; and Zhuang et al. (2020) created AdaBelief to adapt step sizes based on gradient belief. Recent developments continue this trend, such as Xie et al. (2020b) which decouples adaptation from momentum in the Adai framework, and Xie et al. (2024) which introduced Adan with a novel Nesterov momentum estimation. The latest works from Liang et al. (2024) and Yuan et al. (2024) further enhance efficiency with new masking and variance-reduction strategies, respectively. Additionally, the MGUP strategy proposed by Chang & Yuan (2025) smooths zero masks into small non-zero values to alleviate potential non-convergence issues that the Cautious mask may cause in Adam.

Beyond Adam variants, research has explored other paradigms such as preconditioning methods that use parameter curvature. Gupta et al. (2018) pioneered this direction with the Shampoo optimizer. Building on this work, Jordan et al. (2024) proposed Muon, which adapts to curvature by orthogonalizing gradient momentum. Subsequent variants emerged, such as AdaMuon Si et al. (2025) which adds element-wise adaptivity, and COSMOS Liu et al. (2025b) which integrates ideas from SOAP Vyas et al. (2025) for large model training. While these methods showed practical benefits, they often lacked convergence proofs. To bridge this theoretical gap for LMO-based methods, Gluon Riabinin et al. (2025) introduces a novel layer-wise smoothness assumption, providing convergence guarantees that align with the practical implementations of optimizers like Muon and Scion Pethick et al. (2025). In the literature, other related preconditioning methods include ASGO An et al. (2025), PolarGrad Lau et al. (2025), and AdaGO Zhang et al. (2025), which introduces the Adagrad-Norm step size Ward et al. (2020) into a simplified version of Muon. Meanwhile, other high-performing optimizers not belonging to the Shampoo family also warrant attention, such as Sophia Liu et al. (2023), which improves second-moment estimation through efficient diagonal Hessian approximation and coordinate clipping, and Lion Chen et al. (2023), a lightweight optimizer that only tracks momentum and uses the sign function to normalize updates. These methods are closely related to normalized SGD with momentum variants, where the gradient is rescaled or truncated before the update Cutkosky & Mehta (2020; 2021), in contrast to Muon and our Muon-MVR variants, which perform spectral-norm-based matrix orthogonalization.

## 6.2 Analysis of Muon

Recent theoretical analyses have clarified the mechanisms and convergence of the Muon optimizer. One central line of work models modern optimizers as steepest-descent or trust-region methods under non-Euclidean norm constraints Bernstein & Newhouse (2024); Kovalev (2025), clarifying how spectral-norm constraints shape the orthogonalized update direction and, in the case of Kovalev (2025). Building on this, subsequent work has linked Muon with weight decay to the stochastic Frank-Wolfe method, identifying it as an instance operating under a spectral norm constraint Sfyraki & Wang (2025). From a constrained optimization viewpoint, Muon has also been characterized as a special case of Lion-$\mathcal{K}$, with its convergence to a KKT point proven in both deterministic and stochastic settings Chen et al. (2025). These analyses, alongside broader research on norm-constrained stochastic conditional gradient methods, form Muon's theoretical underpinnings Pethick et al. (2025). Beyond its formal framework, Muon's implicit bias has become a key area of investigation. It has been demonstrated that Muon tends to converge toward max-margin solutions with respect to the spectral norm of the weight matrix, revealing a distinct implicit regularization and generalization preference compared to Adam Fan et al. (2025). A unifying preconditioning perspective, based on matrix polar decomposition, explains this behavior by distinguishing between curvature and gradient anisotropy. This decomposition clarifies the differing ways Muon and Adam handle various parameter types Lau et al. (2025). The convergence analysis of Muon has also seen active development. Although rigorous proofs and iteration complexity in non-convex settings have been established under various smoothness assumptions Shen et al. (2025), the validity of some early results has been challenged. For instance, certain proofs were found to rely on incorrect mathematical inequalities, casting doubt on their conclusions Li & Hong (2025). Other analyses, while confirming the convergence of several Muon variants, report slow convergence rates or the need for stringent conditions to reach a stationary point Sato et al. (2025). On a practical level, especially for pretraining large-scale language models (LLMs), weight decay has been identified as an indispensable component for Muon. To enhance its scalability and utility, methods based on RMS analysis have also been proposed to effectively transfer learning rates from Adam to Muon Liu et al. (2025a).

## 7 Conclusion

In this work, we establish a rigorous theoretical foundation for the Muon optimizer, addressing the gap between its empirical success and formal analysis. We analyze two momentum-based variance-reduced variants of Muon: a one-batch version (Muon-MVR1) and a two-batch version (Muon-MVR2). We provide the first proof that Muon-MVR2 achieves the optimal iteration complexity of $\tilde{\mathcal{O}}(T^{-1/3})$ in stochastic nonconvex settings, matching the theoretical lower bound Arjevani et al. (2023). Furthermore, we show that under the Polyak–Łojasiewicz condition, Muon-MVR1 and Muon-MVR2 attain sublinear nonergodic convergence rates of $\tilde{\mathcal{O}}(T^{-1/2})$ and $\tilde{\mathcal{O}}(T^{-2/3})$, respectively. Our theoretical results are validated by extensive experiments on the CIFAR-10 and C4 benchmarks, which confirm the practical acceleration and superior performance of Muon-MVR2 over its standard counterpart and other widely used optimizers. Overall, this research strengthens the Muon framework by providing robust theoretical guarantees and a practically effective new variant for deep learning training.

## 8 Limitations and future work

▶ **Limitations**: First, a systematic comparison with other Muon-type optimizers is currently lacking. Second, a gap remains between the theoretical assumption of exact orthogonalization and the practical use of finite Newton–Schulz iterations, particularly for the inexact Muon-MVR variant.

▶ **Future work**: Future directions include conducting a large-scale, unified evaluation of Muon variants with thorough tuning. Furthermore, it is valuable to derive rigorous guarantees for finite-step Newton–Schulz orthogonalization and to improve the theoretical convergence rate of Muon-MVR1 beyond $\tilde{\mathcal{O}}(T^{-1/4})$.

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

# APPENDIX

## A  LEMMAS FOR THEOREM 3.1

### A.1  LEMMA A.1

**Lemma A.1.** *For Algorithm 1, choosing an arbitrary parameter $\alpha > 0$, we have the following inequality:*

$$f(\mathbf{X}_{t+1}) \leq f(\mathbf{X}_t) - \eta_t \|\mathbf{M}_t\|_F + \frac{\eta_t \alpha}{2} \|\nabla f(\mathbf{X}_t) - \mathbf{M}_t\|_F^2 + \frac{\eta_t n}{2\alpha} + \frac{L\eta^2 n}{2}.$$

*Proof.* According to Assumption 3.2, we have the upper bound for the function value:

$$f(\mathbf{X}_{t+1}) \leq f(\mathbf{X}_t) + \langle \nabla f(\mathbf{X}_t), \mathbf{X}_{t+1} - \mathbf{X}_t \rangle + \frac{L}{2} \|\mathbf{X}_{t+1} - \mathbf{X}_t\|_F^2$$

$$\leq f(\mathbf{X}_t) - \eta_t \langle \nabla f(\mathbf{X}_t), \mathbf{O}_t \rangle + \frac{L\eta_t^2}{2} \|\mathbf{O}_t\|_F^2$$

$$\leq f(\mathbf{X}_t) - \eta_t \langle \mathbf{M}_t, \mathbf{O}_t \rangle - \eta_t \langle \nabla f(\mathbf{X}_t) - \mathbf{M}_t, \mathbf{O}_t \rangle + \frac{L\eta_t^2}{2} \|\mathbf{O}_t\|_F^2.$$

Now we bound the three terms on the right-hand side respectively:

Descent Term: According to the definition of $\mathbf{O}_t$ and the norm property $\|\mathbf{M}_t\|_* \geq \|\mathbf{M}_t\|_F$, we have:

$$-\eta_t \langle \mathbf{M}_t, \mathbf{O}_t \rangle = -\eta_t \langle \mathbf{M}_t, \mathbf{U}_r \mathbf{V}_r^\top \rangle = -\eta_t \|\mathbf{M}_t\|_* \leq -\eta_t \|\mathbf{M}_t\|_F.$$

Cross Term: This is the key to eliminating the dimension-dependent error. We use Young's inequality with a parameter ($ab \leq \frac{\alpha}{2}a^2 + \frac{1}{2\alpha}b^2$) and the fact that $\|\mathbf{O}_t\|_F^2 \leq \sum_{i=1}^n 1 = n$:

$$-\eta_t \langle \nabla f(\mathbf{X}_t) - \mathbf{M}_t, \mathbf{O}_t \rangle \leq \eta_t \|\nabla f(\mathbf{X}_t) - \mathbf{M}_t\|_F \|\mathbf{O}_t\|_F$$

$$\leq \eta_t \left( \frac{\alpha}{2} \|\nabla f(\mathbf{X}_t) - \mathbf{M}_t\|_F^2 + \frac{1}{2\alpha} \|\mathbf{O}_t\|_F^2 \right)$$

$$\leq \eta_t \left( \frac{\alpha}{2} \|\nabla f(\mathbf{X}_t) - \mathbf{M}_t\|_F^2 + \frac{n}{2\alpha} \right).$$

Quadratic Term:

$$\frac{L\eta_t^2}{2} \|\mathbf{O}_t\|_F^2 \leq \frac{L\eta_t^2 n}{2}.$$

Substituting these three bounds into the inequality for $f(\mathbf{X}_{t+1})$:

$$f(\mathbf{X}_{t+1}) \leq f(\mathbf{X}_t) - \eta_t \|\mathbf{M}_t\|_F + \frac{\eta_t \alpha}{2} \|\nabla f(\mathbf{X}_t) - \mathbf{M}_t\|_F^2 + \frac{\eta_t n}{2\alpha} + \frac{L\eta_t^2 n}{2}.$$

This completes the proof. □

### A.2  LEMMA A.2

**Lemma A.2.** *For Algorithm 1 option MVR1 ($\gamma = 0$), the accumulated error between the momentum term and the true gradient is bounded:*

$$\mathbb{E}\left[ \|\mathbf{M}_{t+1} - \nabla f(\mathbf{X}_{t+1})\|_F^2 \right] \leq \beta_{t+1} \mathbb{E}\left[ \|\mathbf{M}_t - \nabla f(\mathbf{X}_t)\|_F^2 \right] + \frac{\beta_{t+1}^2}{1 - \beta_{t+1}} L^2 \eta_t^2 n + (1 - \beta_{t+1})^2 \sigma^2.$$

*Proof.* First, we have

$$\|\mathbf{M}_{t+1} - \nabla f(\mathbf{X}_{t+1})\|_F^2$$

$$= \|\beta_{t+1}\mathbf{M}_t + (1 - \beta_{t+1})\nabla f(\mathbf{X}_{t+1}; \xi_{t+1}) - \nabla f(\mathbf{X}_{t+1})\|_F^2$$

$$= \|\beta_{t+1}(\mathbf{M}_t - \nabla f(\mathbf{X}_t)) + (1 - \beta_{t+1})(\nabla f(\mathbf{X}_{t+1}; \xi_{t+1}) - \nabla f(\mathbf{X}_{t+1}))$$
$$\quad + \beta_{t+1}(\nabla f(\mathbf{X}_t) - \nabla f(\mathbf{X}_{t+1}))\|_F^2$$

$$= \beta_{t+1}^2 \|\mathbf{M}_t - \nabla f(\mathbf{X}_t)\|_F^2 + \beta_{t+1}^2 \|\nabla f(\mathbf{X}_t) - \nabla f(\mathbf{X}_{t+1})\|_F^2$$
$$\quad + (1 - \beta_{t+1})^2 \|\nabla f(\mathbf{X}_{t+1}; \xi_{t+1}) - \nabla f(\mathbf{X}_{t+1})\|_F^2$$
$$\quad + 2\beta_{t+1}^2 \langle \mathbf{M}_t - \nabla f(\mathbf{X}_t), \nabla f(\mathbf{X}_t) - \nabla f(\mathbf{X}_{t+1})\rangle_F$$
$$\quad + 2\beta_{t+1}(1 - \beta_{t+1})\langle \mathbf{M}_t - \nabla f(\mathbf{X}_t), \nabla f(\mathbf{X}_{t+1}; \xi_{t+1}) - \nabla f(\mathbf{X}_{t+1})\rangle_F$$
$$\quad + 2\beta_{t+1}(1 - \beta_{t+1})\langle \nabla f(\mathbf{X}_t) - \nabla f(\mathbf{X}_{t+1}), \nabla f(\mathbf{X}_{t+1}; \xi_{t+1}) - \nabla f(\mathbf{X}_{t+1})\rangle_F.$$

According to Assumption 3.4. Taking the expectation of its squared norm, and using the unbiasedness and independence of the stochastic gradient, we obtain:

$$\mathbb{E}[\|\mathbf{M}_{t+1} - \nabla f(\mathbf{X}_{t+1})\|_F^2] = \beta_{t+1}^2 \mathbb{E}[\|\mathbf{M}_t - \nabla f(\mathbf{X}_t)\|_F^2]$$
$$\quad + (1 - \beta_{t+1})^2 \mathbb{E}[\|\nabla f(\mathbf{X}_{t+1}; \xi_{t+1}) - \nabla f(\mathbf{X}_{t+1})\|_F^2]$$
$$\quad + \beta_{t+1}^2 \mathbb{E}[\|\nabla f(\mathbf{X}_t) - \nabla f(\mathbf{X}_{t+1})\|_F^2]$$
$$\quad + 2\beta_{t+1}^2 \mathbb{E}[\langle \mathbf{M}_t - \nabla f(\mathbf{X}_t), \nabla f(\mathbf{X}_t) - \nabla f(\mathbf{X}_{t+1})\rangle].$$

Applying Young's inequality with a parameter $(ab \leq \frac{\epsilon}{2}a^2 + \frac{1}{2\epsilon}b^2)$, we have

$$\langle \mathbf{M}_t - \nabla f(\mathbf{X}_t), \nabla f(\mathbf{X}_t) - \nabla f(\mathbf{X}_{t+1})\rangle_F \leq \frac{\epsilon}{2}\|\mathbf{M}_t - \nabla f(\mathbf{X}_t)\|_F^2 + \frac{1}{2\epsilon}\|\nabla f(\mathbf{X}_t) - \nabla f(\mathbf{X}_{t+1})\|_F^2.$$

Thus, we have:

$$\mathbb{E}\left[\|\mathbf{M}_{t+1} - \nabla f(\mathbf{X}_{t+1})\|_F^2\right] \leq \beta_{t+1}^2(1 + \epsilon)\mathbb{E}\left[\|\mathbf{M}_t - \nabla f(\mathbf{X}_t)\|_F^2\right]$$
$$\quad + \beta_{t+1}^2\left(1 + \frac{1}{\epsilon}\right)\mathbb{E}\left[\|\nabla f(\mathbf{X}_t) - \nabla f(\mathbf{X}_{t+1})\|_F^2\right]$$
$$\quad + (1 - \beta_{t+1})^2\mathbb{E}\left[\|\nabla f(\mathbf{X}_{t+1}; \xi_{t+1}) - \nabla f(\mathbf{X}_{t+1})\|_F^2\right].$$

According to Assumption 3.2,

$$\|\nabla f(\mathbf{X}_t) - \nabla f(\mathbf{X}_{t+1})\|_F^2 \leq L^2\|\mathbf{X}_t - \mathbf{X}_{t+1}\|_F^2$$
$$= L^2\eta_t^2\|\mathbf{O}_t\|_F^2$$
$$\leq L^2\eta_t^2 n.$$

Therefore:

$$\mathbb{E}\left[\|\mathbf{M}_{t+1} - \nabla f(\mathbf{X}_{t+1})\|_F^2\right] \leq \beta_{t+1}^2(1 + \epsilon)\mathbb{E}\left[\|\mathbf{M}_t - \nabla f(\mathbf{X}_t)\|_F^2\right]$$
$$\quad + \beta_{t+1}^2\left(1 + \frac{1}{\epsilon}\right)L^2\eta_t^2 n + (1 - \beta_{t+1})^2\sigma^2.$$

Then, by letting $\epsilon := \frac{1 - \beta_{t+1}}{\beta_{t+1}}$, we have

$$\mathbb{E}\left[\|\mathbf{M}_{t+1} - \nabla f(\mathbf{X}_{t+1})\|_F^2\right] \leq \beta_{t+1}\mathbb{E}\left[\|\mathbf{M}_t - \nabla f(\mathbf{X}_t)\|_F^2\right] + \frac{\beta_{t+1}^2}{1 - \beta_{t+1}}L^2\eta_t^2 n + (1 - \beta_{t+1})^2\sigma^2. \tag{6}$$

$\square$

## A.3 LEMMA A.3

**Lemma A.3.** *Suppose that $\{E_i, A_i\}$ are two nonnegative sequences. Assume $E_{t+1} \leq (1 - \alpha_{t+1})E_t + A_{t+1}$ where $\alpha_t = t^{-p}$, $p \in (0, 1]$. Then we have:*

$$\alpha_t E_t \leq 2(E_t - E_{t+1} + A_{t+1}).$$

*Proof.* We derive the following inequalities:

$$\alpha_t E_t - c\left(E_t - E_{t+1} + A_{t+1}\right)$$

$$\overset{(\bullet)}{\leq} \alpha_t E_t - c\left(E_t + A_{t+1}\right) + c \cdot \left(E_t - \alpha_{t+1} E_t + A_{t+1}\right)$$

$$= E_t\left(\alpha_t - c\alpha_{t+1}\right)$$

$$= E_t \cdot (t+1)^{-p} \cdot \left(\left(\frac{t}{t+1}\right)^{-p} - c\right)$$

$$\overset{(\circ)}{\leq} E_t \cdot (t+1)^{-p} \cdot (2 - c)$$

$$\overset{(\star)}{\leq} 0,$$

where $(\bullet)$ follows from $E_{t+1} \leq (1 - \alpha_{t+1})E_t + A_{t+1}$; $(\circ)$ is due to $(\frac{t}{t+1})^{-p} \leq 2^p \leq 2$; $(\star)$ is due to our choice $c = 2$. $\qquad\square$

## B    PROOFS OF THEOREM 3.1

*Proof.* According to Lemma A.1, we have:

$$f(\mathbf{X}_{t+1}) \leq f(\mathbf{X}_t) - \eta_t\|\mathbf{M}_t\|_F + \frac{\eta_t\alpha}{2}\|\nabla f(\mathbf{X}_t) - \mathbf{M}_t\|_F^2 + \frac{\eta_t n}{2\alpha} + \frac{L\eta_t^2 n}{2}$$

$$\overset{(\circ)}{\leq} f(\mathbf{X}_t) - \eta_t\|\mathbf{M}_t\|_F + \frac{\eta_t^{2/3}}{2L}\|\nabla f(\mathbf{X}_t) - \mathbf{M}_t\|_F^2 + \frac{\eta_t^{4/3}Ln}{2} + \frac{L\eta_t^2 n}{2}$$

$$\overset{(\star)}{\leq} f(\mathbf{X}_t) - \eta_t\|\mathbf{M}_t\|_F + \frac{\eta_t^{2/3}}{2L}\|\nabla f(\mathbf{X}_t) - \mathbf{M}_t\|_F^2 + Ln\eta_t^{4/3},$$

where $(\circ)$ by setting $\alpha = \frac{1}{\eta_t^{1/3}L}$; $(\star)$ follows from $\eta_t \leq 1$, we have $L \leq L/\eta_t^{1/3}$.

Thus, taking the expectation yields

$$\mathbb{E}[f(\mathbf{X}_{t+1})] \leq \mathbb{E}[f(\mathbf{X}_t)] - \eta_t\mathbb{E}[\|\mathbf{M}_t\|_F] + \frac{\eta_t^{2/3}}{2L}\mathbb{E}[\|\nabla f(\mathbf{X}_t) - \mathbf{M}_t\|_F^2] + Ln\eta_t^{4/3}$$

$$\overset{(\circ)}{\leq} \mathbb{E}[f(\mathbf{X}_t)] - \eta_t\mathbb{E}[\|\nabla f(\mathbf{X}_t)\|_F] + \eta_t\mathbb{E}[\|\nabla f(\mathbf{X}_t) - \mathbf{M}_t\|_F]$$

$$+ \frac{\eta_t^{2/3}}{2L}\mathbb{E}[\|\nabla f(\mathbf{X}_t) - \mathbf{M}_t\|_F^2] + Ln\eta_t^{4/3}$$

$$\overset{(\star)}{\leq} \mathbb{E}[f(\mathbf{X}_t)] - \eta_t\mathbb{E}[\|\nabla f(\mathbf{X}_t)\|_F] + \frac{1}{2\epsilon}\eta_t^2 + \frac{\epsilon}{2}\mathbb{E}[\|\nabla f(\mathbf{X}_t) - \mathbf{M}_t\|_F^2]$$

$$+ \frac{\eta_t^{2/3}}{2L}\mathbb{E}[\|\nabla f(\mathbf{X}_t) - \mathbf{M}_t\|_F^2] + Ln\eta_t^{4/3} \qquad (7)$$

$$\overset{(\bullet)}{=} \mathbb{E}[f(\mathbf{X}_t)] - \eta_t\mathbb{E}[\|\nabla f(\mathbf{X}_t)\|_F]$$

$$+ \underbrace{\frac{L\eta_t^{4/3}}{2} + \left(\frac{\eta_t^{2/3}}{2L} + \frac{\eta_t^{2/3}}{2L}\right)\mathbb{E}[\|\nabla f(\mathbf{X}_t) - \mathbf{M}_t\|_F^2] + Ln\eta_t^{4/3}}_{\Gamma_t},$$

where $(\circ)$ follows from the reverse triangle inequality $-\|\mathbf{M}_t\|_F \leq \|\nabla f(\mathbf{X}_t) - \mathbf{M}_t\|_F - \|\nabla f(\mathbf{X}_t)\|_F$; $(\star)$ applies Young's inequality to the term $\eta_t\mathbb{E}[\|\nabla f(\mathbf{X}_t) - \mathbf{M}_t\|_F]$; and $(\bullet)$ collects the residual terms into $\Gamma_t$ and sets $\epsilon = \eta_t^{2/3}/L$.

Next, we set $\eta_t = t^{-3/4}$ and $\beta_t = 1 - t^{-1/2}, \alpha_t = t^{-1/2}$.

Case 1: $\gamma = 0$.

By Lemma A.2 inequality (6), we have

$$\mathbb{E}\left[\|\mathbf{M}_{t+1} - \nabla f(\mathbf{X}_{t+1})\|_F^2\right] \le \beta_{t+1}\mathbb{E}\left[\|\mathbf{M}_t - \nabla f(\mathbf{X}_t)\|_F^2\right] + \frac{\beta_{t+1}^2}{1-\beta_{t+1}}L^2\eta_t^2 n + (1-\beta_{t+1})^2\sigma^2$$

$$\le \beta_{t+1}\mathbb{E}\left[\|\mathbf{M}_t - \nabla f(\mathbf{X}_t)\|_F^2\right] + \frac{L^2\eta_t^2 n}{1-\beta_{t+1}} + (1-\beta_{t+1})^2\sigma^2.$$

Let $\mathbf{S}_{t+1} = \mathbf{M}_{t+1} - \nabla f(\mathbf{X}_{t+1})$. Thus, setting $\alpha_{t+1} = (t+1)^{-1/2}$, we observe the following relationship for $t \ge 1$:

$$\frac{\eta_t^2}{1-\beta_{t+1}} = \frac{t^{-3/2}}{(t+1)^{-1/2}} = \frac{\sqrt{t+1}}{t^{3/2}} \le \frac{2\sqrt{2}}{t+1}.$$

This allows us to bound the expectation as follows:

$$\mathbb{E}\|\mathbf{S}_{t+1}\|_F^2 \le (1-\alpha_{t+1})\mathbb{E}\|\mathbf{S}_t\|_F^2 + \alpha_{t+1}^2(2\sqrt{2}L^2 n + \sigma^2).$$

According to Lemma A.3, by letting $A_{t+1} = \alpha_{t+1}^2(2\sqrt{2}L^2 n + \sigma^2)$, we have

$$\alpha_t \mathbb{E}\|\mathbf{S}_t\|_F^2 \le 2(\mathbb{E}\|\mathbf{S}_t\|_F^2 - \mathbb{E}\|\mathbf{S}_{t+1}\|_F^2 + A_{t+1}).$$

Furthermore, since

$$\mathbb{E}\|\mathbf{S}_1\| = \mathbb{E}\|\nabla f(\mathbf{X}_1) - \mathbf{M}_1\|_F^2 = \mathbb{E}\|\nabla f(\mathbf{X}_1) - (1-\beta_1)\nabla f(\mathbf{X}_1;\xi_1)\|_F^2$$

$$= \mathbb{E}\|\nabla f(\mathbf{X}_1) - \nabla f(\mathbf{X}_1;\xi_1)\|_F^2 \le \sigma^2 \qquad (\text{since } \beta_1 = 0).$$

It follows that

$$\sum_{t=1}^{T}\alpha_t\mathbb{E}\|\mathbf{S}_t\|_F^2 \le 2\sum_{t=1}^{T}(\mathbb{E}\|\mathbf{S}_t\|_F^2 - \mathbb{E}\|\mathbf{S}_{t+1}\|_F^2 + A_{t+1})$$

$$\le 2\mathbb{E}\|\mathbf{S}_1\|_F^2 + 2(2\sqrt{2}L^2 n + \sigma^2)\sum_{t=1}^{T}\frac{1}{t+1} \tag{8}$$

$$\le 2\sigma^2 + 2(2\sqrt{2}L^2 n + \sigma^2)(\ln T + 1).$$

Thus,

$$\Gamma_t = \frac{\eta_t^{2/3}}{L}\mathbb{E}\|\mathbf{S}_t\|_F^2 + (L/2 + Ln)\eta_t^{4/3}$$

$$= \frac{\alpha_t}{L}\mathbb{E}\|\mathbf{S}_t\|_F^2 + (L/2 + Ln)\alpha_t^2.$$

Next, we define $A_1 = 4\sqrt{2}Ln + Ln + 2L^{-1}\sigma^2 + L/2$ and $A_2 = 4\sqrt{2}Ln + Ln + 4L^{-1}\sigma^2 + L/2$.

$$\sum_{t=1}^{T}\Gamma_t = \frac{1}{L}\sum_{t=1}^{T}\alpha_t\mathbb{E}\|\mathbf{S}_t\|_F^2 + (L/2 + Ln)\sum_{t=1}^{T}\alpha_t^2$$

$$\overset{(\circ)}{\le} \frac{2\sigma^2 + 2(2\sqrt{2}L^2 n + \sigma^2)(\ln T + 1)}{L} + (L/2 + Ln)(\ln T + 1) \tag{9}$$

$$\le A_1 \ln T + A_2,$$

where $(\circ)$ due to inequality (8).

Therefore, we have

$$\frac{1}{T}\sum_{t=1}^{T}\mathbb{E}\|\nabla f(\mathbf{X}_t)\|_F = \frac{1}{T}\sum_{t=1}^{T}t^{3/4}\cdot t^{-3/4}\mathbb{E}\|\nabla f(\mathbf{X}_t)\|_F$$

$$\le \frac{1}{T}\sum_{t=1}^{T}t^{3/4}\cdot\eta_t\mathbb{E}\|\nabla f(\mathbf{X}_t)\|_F$$

$$\le \frac{T^{3/4}}{T}\sum_{t=1}^{T}f(\mathbf{X}_t) - f(\mathbf{X}_{t+1}) + \frac{T^{3/4}}{T}\sum_{t=1}^{T}\Gamma_t$$

$$\le \frac{f(\mathbf{X}_1) - f^*}{T^{1/4}} + \frac{A_1 \ln T + A_2}{T^{1/4}}$$

$$= \tilde{\mathcal{O}}(T^{-1/4}).$$

Case 2: $\gamma \neq 0$.

We set $\eta_t = t^{-3/4}$ and $\beta_t = 1 - (t+1)^{-1/2}, \alpha_t = t^{-1/2}$, and $\gamma = 1 - \beta_{t-1} = t^{-1/2}$, we first note that an equivalent form of Algorithm 1 Option MVR1 is given by

$$
\begin{aligned}
\mathbf{C}_t &= \beta_{t-1}\mathbf{C}_{t-1} + (1 - \beta_{t-1})\nabla f(\mathbf{X}_t; \xi_t) \\
\mathbf{M}_t &= \beta_t \mathbf{C}_t + (1 - \beta_t)\nabla f(\mathbf{X}_t; \xi_t).
\end{aligned}
\tag{10}
$$

In this case, then we have

$$
\begin{aligned}
&\mathbb{E}\|\mathbf{C}_t - \nabla f(\mathbf{X}_t)\|_F^2 \\
&= \mathbb{E}\|\beta_{t-1}\mathbf{C}_{t-1} + (1 - \beta_{t-1})\nabla f(\mathbf{X}_t; \xi_t) - \nabla f(\mathbf{X}_t)\|_F^2 \\
&= \mathbb{E}\|\beta_{t-1}(\mathbf{C}_{t-1} - \nabla f(\mathbf{X}_{t-1})) + (1 - \beta_{t-1})(\nabla f(\mathbf{X}_t; \xi_t) - \nabla f(\mathbf{X}_t)) \\
&\quad + \beta_{t-1}(\nabla f(\mathbf{X}_{t-1}) - \nabla f(\mathbf{X}_t))\|_F^2 \\
&\overset{(\circ)}{\leq} \beta_{t-1}\mathbb{E}\|\mathbf{C}_{t-1} - \nabla f(\mathbf{X}_{t-1})\|_F^2 + \frac{\beta_{t-1}^2 L\eta_{t-1}^2 n}{1 - \beta_{t-1}} + (1 - \beta_{t-1})^2\sigma^2 \\
&= (1 - \alpha_t)\mathbb{E}\|\mathbf{C}_{t-1} - \nabla f(\mathbf{X}_{t-1})\|_F^2 + \alpha_t^2(2\sqrt{2}Ln + \sigma^2),
\end{aligned}
$$

where $(\circ)$ follows from Lemma A.2.

Then, we define $\mathbf{S}_t' = \|\mathbf{M}_t - \nabla f(\mathbf{X}_t)\|_F^2, \mathbf{S}_t = \|\mathbf{C}_t - \nabla f(\mathbf{X}_t)\|_F^2, A_{t+1} = \alpha_{t+1}^2(2\sqrt{2}Ln + \sigma^2)$. Using the conclusion of Lemma A.3, we have

$$
\mathbb{E}\|\mathbf{S}_{t+1}\|_F^2 \leq (1 - \alpha_{t+1})\mathbb{E}\|\mathbf{S}_t\|_F^2 + A_{t+1}.
$$

Furthermore, since

$$
\begin{aligned}
\mathbb{E}\|\mathbf{S}_1\| &= \mathbb{E}\|\nabla f(\mathbf{X}_1) - \mathbf{C}_1\|_F^2 = \mathbb{E}\|\nabla f(\mathbf{X}_1) - (1 - \beta_0)\nabla f(\mathbf{X}_1; \xi_1)\|_F^2 \\
&= \mathbb{E}\|\nabla f(\mathbf{X}_1) - \nabla f(\mathbf{X}_1; \xi_1)\|_F^2 \leq \sigma^2 \qquad (\text{since } \beta_0 = 0).
\end{aligned}
$$

Thus, as in inequality (8), we have

$$
\begin{aligned}
\sum_{t=1}^T \alpha_t \mathbb{E}\|\mathbf{S}_t\|_F^2 &\leq 2\sum_{t=1}^T (\mathbb{E}\|\mathbf{S}_t\|_F^2 - \mathbb{E}\|\mathbf{S}_{t+1}\|_F^2 + A_{t+1}) \\
&\leq 2\mathbb{E}\|\mathbf{S}_1\|_F^2 + 2(2\sqrt{2}L^2n + \sigma^2)\sum_{t=1}^T \frac{1}{t+1} \\
&\leq 2\sigma^2 + 2(2\sqrt{2}L^2n + \sigma^2)(\ln T + 1).
\end{aligned}
$$

Then,

$$
\begin{aligned}
\|\mathbf{M}_t - \nabla f(\mathbf{X}_t)\|_F &= \|\beta_t\mathbf{C}_t + (1 - \beta_t)\nabla f(\mathbf{X}_t; \xi_t) - \nabla f(\mathbf{X}_t)\|_F \\
&= \|\beta_t(\mathbf{C}_t - \nabla f(\mathbf{X}_t)) + (1 - \beta_t)(\nabla f(\mathbf{X}_t; \xi_t) - \nabla f(\mathbf{X}_t))\|_F.
\end{aligned}
$$

From this, we can bound the squared norm using the inequality $\|\mathbf{A} + \mathbf{B}\|_F^2 \leq 2\|\mathbf{A}\|_F^2 + 2\|\mathbf{B}\|_F^2$:

$$
\|\mathbf{M}_t - \nabla f(\mathbf{X}_t)\|_F^2 \leq 2\beta_t^2\|\mathbf{C}_t - \nabla f(\mathbf{X}_t)\|_F^2 + 2(1 - \beta_t)^2\|\nabla f(\mathbf{X}_t; \xi_t) - \nabla f(\mathbf{X}_t)\|_F^2.
$$

Thus, we have

$$
\begin{aligned}
\mathbb{E}\|\mathbf{M}_t - \nabla f(\mathbf{X}_t)\|_F^2 &\leq 2\beta_t^2\mathbb{E}\|\mathbf{C}_t - \nabla f(\mathbf{X}_t)\|_F^2 + 2(1 - \beta_t)^2\mathbb{E}\|\nabla f(\mathbf{X}_t; \xi_t) - \nabla f(\mathbf{X}_t)\|_F^2 \\
&\leq 2\beta_t^2\mathbb{E}\|\mathbf{C}_t - \nabla f(\mathbf{X}_t)\|_F^2 + 2(1 - \beta_t)^2\sigma^2 \\
&\leq 2\mathbb{E}\|\mathbf{C}_t - \nabla f(\mathbf{X}_t)\|_F^2 + 2(1 - \beta_t)^2\sigma^2.
\end{aligned}
\tag{11}
$$

Thus,

$$
\begin{aligned}
\sum_{t=1}^T \alpha_t\mathbb{E}\|\mathbf{S}_t'\|_F^2 &\leq 2\sum_{t=1}^T \alpha_t\mathbb{E}\|\mathbf{S}_t\|_F^2 + 2\sigma^2\sum_{t=1}^T \frac{1}{t(t+1)} \\
&\overset{(\circ)}{\leq} 4\sigma^2 + 4(2\sqrt{2}L^2n + \sigma^2)(\ln T + 1) + 2\sigma^2 \\
&= 6\sigma^2 + 4(2\sqrt{2}L^2n + \sigma^2)(\ln T + 1),
\end{aligned}
\tag{12}
$$

where $(\circ)$ is due to $\sum_{t=1}^{T} \frac{1}{t(t+1)} = \sum_{t=1}^{T} (\frac{1}{t} - \frac{1}{t+1}) = 1 - \frac{1}{T+1} \leq 1$.

Therefore,

$$
\begin{aligned}
\Gamma_t &= \frac{\eta_t^{2/3}}{L} \mathbb{E}\|\mathbf{S}_t'\|_F^2 + (L/2 + Ln)\eta_t^{4/3} \\
&= \frac{\alpha_t}{L} \mathbb{E}\|\mathbf{S}_t'\|_F^2 + (L/2 + Ln)\alpha_t^2.
\end{aligned}
$$

Then, we define $A_1 = 8\sqrt{2}Ln + Ln + 4L^{-1}\sigma^2 + L/2$ and $A_2 = 8\sqrt{2}Ln + Ln + 10L^{-1}\sigma^2 + L/2$. We have

$$
\begin{aligned}
\sum_{t=1}^{T} \Gamma_t &= \frac{1}{L} \sum_{t=1}^{T} \alpha_t \mathbb{E}\|\mathbf{S}_t'\|_F^2 + (L/2 + Ln) \sum_{t=1}^{T} \alpha_t^2 \\
&\overset{(\circ)}{\leq} \frac{6\sigma^2 + 4(2\sqrt{2}L^2 n + \sigma^2)(\ln T + 1)}{L} + (L/2 + Ln)(\ln T + 1) \\
&\leq A_1 \ln T + A_2,
\end{aligned}
\tag{13}
$$

where $(\circ)$ is due to inequality (12).

Then, we have

$$
\begin{aligned}
\frac{1}{T} \sum_{t=1}^{T} \mathbb{E}\|\nabla f(\mathbf{X}_t)\|_F &= \frac{1}{T} \sum_{t=1}^{T} t^{3/4} \cdot t^{-3/4} \mathbb{E}\|\nabla f(\mathbf{X}_t)\|_F \\
&\leq \frac{1}{T} \sum_{t=1}^{T} t^{3/4} \cdot \eta_t \mathbb{E}\|\nabla f(\mathbf{X}_t)\|_F \\
&\leq \frac{T^{3/4}}{T} \sum_{t=1}^{T} f(\mathbf{X}_t) - f(\mathbf{X}_{t+1}) + \frac{T^{3/4}}{T} \sum_{t=1}^{T} \Gamma_t \\
&\leq \frac{f(\mathbf{X}_1) - f^*}{T^{1/4}} + \frac{A_1 \ln T + A_2}{T^{1/4}} \\
&= \tilde{\mathcal{O}}(T^{-1/4}).
\end{aligned}
$$

This completes the proof. $\qquad\square$

## C  LEMMAS FOR THEOREM 3.2

### C.1  LEMMA C.1

**Lemma C.1.** *Let $\{A_t\}_{t \geq 1}$ and $\{B_t\}_{t \geq 1}$ be non-negative sequences satisfying the relation $A_{t+1} \leq (1 - \varepsilon_{t+1})A_t + B_{t+1}$ for all $t \geq 1$. If we define the sequence $\varepsilon_t = t^{-p}$ for some constant $p \in (0, 1]$, then for all $t \geq 1$, the following inequality holds:*

$$
\sqrt{\varepsilon_t} A_t \leq 4 \left( \frac{A_t}{\sqrt{\varepsilon_t}} - \frac{A_{t+1}}{\sqrt{\varepsilon_{t+1}}} + \frac{B_{t+1}}{\sqrt{\varepsilon_{t+1}}} \right).
$$

*Proof.* First, we define the function $F(t, q) = \frac{1}{4}t^{-q} - t^q + (t+1)^q - (t+1)^{-q}$ for $t \geq 1$ and $q \in (0, 1/2]$. To analyze its properties, let us define $g(t) = F(t, q)$ for a fixed $q \in (0, 1/2]$ and $f(q) = F(t, q)$ for a fixed $t \geq 1$.

Case 1: For $t = 1$. We have $g(1) = \frac{1}{4} - 1 + 2^q - \frac{1}{2^q}$. Since $q \in (0, 1/2]$, this expression is bounded above by its value at $q = 1/2$, yielding $g(1) \leq \frac{1}{4} - 1 + 2^{1/2} - \frac{1}{2^{1/2}} < 0$. Thus, for $t = 1$, the inequality $g(t) \leq 0$ holds.

Case 2: For $t > 1$. For any given $t > 1$ and $q \in (0, 1/2]$, we examine the derivative of $f(q)$:

$$
\begin{aligned}
f'(q) &= \left( \ln(t+1)(t+1)^{-q} - \frac{1}{4}\ln(t)t^{-q} \right) + (\ln(t+1)(t+1)^q - \ln(t)t^q) \\
&\overset{(\circ)}{\geq} \left( \ln(t+1)(t+1)^{-q} - \frac{1}{4}\ln(t)t^{-q} \right) \\
&= \ln(t)t^{-q} \cdot \left\{ \frac{\ln(t+1)}{\ln(t)} \cdot \left( \frac{t}{t+1} \right)^q - \frac{1}{4} \right\} \\
&\overset{(\star)}{\geq} \ln(t)t^{-q} \cdot \left\{ 1 \cdot \sqrt{\frac{t}{t+1}} - \frac{1}{4} \right\} \\
&\overset{(\bullet)}{\geq} \ln(t)t^{-q} \cdot \left\{ \sqrt{\frac{1}{2}} - \frac{1}{4} \right\} \geq 0,
\end{aligned}
\tag{14}
$$

where $(\circ)$ holds because $\ln(t+1) > \ln(t)$ and $(t+1)^q > t^q$ for all $t \geq 1$ and $q \in (0, 1/2]$. The inequality $(\star)$ holds because $\frac{\ln(t+1)}{\ln(t)} > 1$ for $t > 1$, and the function $(\frac{t}{t+1})^q$ is decreasing in $q$, thus its minimum on $(0, 1/2]$ is achieved at $q = 1/2$. The final inequality $(\bullet)$ holds because $t \geq 1$ implies $\sqrt{\frac{t}{t+1}} \geq \sqrt{\frac{1}{2}}$, and $\sqrt{\frac{1}{2}} > 1/4$.

Inequality (14) implies that $f(q)$ is monotonically increasing with respect to $q$ on the interval $(0, 1/2]$.

Next, we consider the boundary condition at $q = 1/2$. Let $h(t) := f(\frac{1}{2}) = \frac{1}{4}t^{-1/2} - t^{1/2} + (t+1)^{1/2} - (t+1)^{-1/2}$. It can be verified that $h(t) \leq 0$ for all $t > 1$. Consequently, we have $F(t, \frac{1}{2}) = f(\frac{1}{2}) \leq 0$ for all $t > 1$.

Finally, for all $t > 1$ and $q \in (0, 1/2]$, we have

$$
F(t, q) \leq F(t, 1/2) \leq 0,
\tag{15}
$$

where the first inequality holds because $f(q)$ is monotonically increasing in $q$ on $(0, 1/2]$ for any fixed $t > 1$.

With this result, we can proceed as follows:

$$
\begin{aligned}
& \sqrt{\varepsilon_t}A_t - 4 \left( \frac{A_t}{\sqrt{\varepsilon_t}} - \frac{A_{t+1}}{\sqrt{\varepsilon_{t+1}}} + \frac{B_{t+1}}{\sqrt{\varepsilon_{t+1}}} \right) \\
&\overset{(\circ)}{\leq} \sqrt{\varepsilon_t}A_t - 4 \left( \frac{A_t}{\sqrt{\varepsilon_t}} + \frac{B_{t+1}}{\sqrt{\varepsilon_{t+1}}} \right) + \frac{4}{\sqrt{\varepsilon_{t+1}}} \cdot (A_t - \varepsilon_{t+1}A_t + B_{t+1}) \\
&= \sqrt{\varepsilon_t}A_t - 4\frac{A_t}{\sqrt{\varepsilon_t}} + \frac{4}{\sqrt{\varepsilon_{t+1}}}(1 - \varepsilon_{t+1})A_t \\
&= A_t \cdot \left( \sqrt{\varepsilon_t} - \frac{4}{\sqrt{\varepsilon_t}} + \frac{4}{\sqrt{\varepsilon_{t+1}}} - 4\sqrt{\varepsilon_{t+1}} \right) \\
&\overset{(\star)}{=} 4A_t \cdot \left( \frac{1}{4}t^{-q} - t^q + (t+1)^q - (t+1)^{-q} \right) \\
&\overset{(\bullet)}{\leq} 0.
\end{aligned}
$$

Here, $(\circ)$ follows from the assumption $A_{t+1} \leq (1 - \varepsilon_{t+1})A_t + B_{t+1}$. The equality $(\star)$ is obtained by substituting $\varepsilon_t = t^{-p}$ and setting $q = \frac{p}{2}$ (note that $p \in (0, 1]$ implies $q \in (0, 1/2]$). Finally, $(\bullet)$ is a direct consequence of our result in inequality (15). $\square$

## C.2 LEMMA C.2

**Lemma C.2.** *For Algorithm 1 option MVR2, let* $\Delta_t = \nabla f(\mathbf{X}_{t+1}; \xi_{t+1}) - \nabla f(\mathbf{X}_t; \xi_{t+1})$, $\delta_t = \nabla f(\mathbf{X}_t) - \nabla f(\mathbf{X}_{t+1})$, $\mathbf{S}_t = \mathbf{M}_t - \nabla f(\mathbf{X}_t)$, *and* $\mathbf{R}_{t+1} = \nabla f(\mathbf{X}_{t+1}; \xi_{t+1}) - \nabla f(\mathbf{X}_{t+1})$. *Then we*

*have the following inequality:*

$$\mathbb{E}\|\mathbf{S}_{t+1}\|_F^2 \le \beta_{t+1}^2\mathbb{E}\|\mathbf{S}_t\|_F^2 + 2\beta_{t+1}^2 L^2\mathbb{E}\|\mathbf{X}_{t+1} - \mathbf{X}_t\|_F^2 + 2(1 - \beta_{t+1})^2\sigma^2 + P_{t+1},$$

*where*

$$A_{t+1} = \frac{B_{t+1} + \beta_{t+1}(\mathbb{E}\|\Delta_t\|_F^2 - \|\mathbb{E}\Delta_t\|_F^2)}{\mathbb{E}\|\Delta_t\|_F^2}$$

$$B_{t+1} = (1 - \beta_{t+1})\mathbb{E}\langle\Delta_t, \mathbf{R}_{t+1}\rangle + \beta_{t+1}\mathbb{E}\langle\Delta_t, \mathbf{S}_t\rangle$$

$$P_{t+1} = \mathbb{E}\|\Delta_t\|_F^2(\beta_{t+1}(1 - \gamma_{t+1}) - A_{t+1})^2 - \mathbb{E}\|\Delta_t\|_F^2 A_{t+1}^2.$$

*If we choose $\gamma_{t+1} = 1 - \frac{A_{t+1}}{\beta_{t+1}}$ or $\gamma_{t+1} = 1$, then*

$$\mathbb{E}\|\mathbf{S}_{t+1}\|_F^2 \le \beta_{t+1}^2\mathbb{E}\|\mathbf{S}_t\|_F^2 + 2\beta_{t+1}^2 L^2\mathbb{E}\|\mathbf{X}_{t+1} - \mathbf{X}_t\|_F^2 + 2(1 - \beta_{t+1})^2\sigma^2.$$

*Proof.* First, we have

$$\begin{aligned}
\mathbf{M}_{t+1} &= \beta_{t+1}\mathbf{M}_t + (1 - \beta_{t+1})\nabla f(\mathbf{X}_{t+1}; \xi_{t+1}) + \gamma_{t+1}\cdot\beta_{t+1}(\nabla f(\mathbf{X}_{t+1}; \xi_{t+1}) - \nabla f(\mathbf{X}_t; \xi_{t+1})) \\
&= (1 - \beta_{t+1})\nabla f(\mathbf{X}_{t+1}; \xi_{t+1}) + \beta_{t+1}(\mathbf{M}_t + \gamma_{t+1}(\nabla f(\mathbf{X}_{t+1}; \xi_{t+1}) - \nabla f(\mathbf{X}_t; \xi_{t+1}))) \\
&= (1 - \beta_{t+1})\nabla f(\mathbf{X}_{t+1}; \xi_{t+1}) + \beta_{t+1}(\mathbf{M}_t + \gamma_{t+1}\Delta_t).
\end{aligned}$$

Hence,

$$\begin{aligned}
\mathbf{M}_{t+1} - \nabla f(\mathbf{X}_{t+1}) &= (1 - \beta_{t+1})\left(\nabla f(\mathbf{X}_{t+1}; \xi_{t+1}) - \nabla f(\mathbf{X}_{t+1})\right) \\
&\quad + \beta_{t+1}\left(\mathbf{M}_t - \nabla f(\mathbf{X}_t) + \delta_t + \gamma_{t+1}\Delta_t\right) \\
&= (1 - \beta_{t+1})\mathbf{R}_{t+1} + \beta_{t+1}(\mathbf{S}_t + \delta_t + \gamma_{t+1}\Delta_t).
\end{aligned}$$

Note that, according to Assumption 3.4:

$$\delta_t = \nabla f(\mathbf{X}_t) - \nabla f(\mathbf{X}_{t+1}) = -\mathbb{E}_{t+1}[\Delta_t].$$

Therefore,

$$\mathbf{S}_{t+1} = (1 - \beta_{t+1})\mathbf{R}_{t+1} + \beta_{t+1}\mathbf{S}_t + \beta_{t+1}(\gamma_{t+1}\Delta_t - \mathbb{E}\Delta_t).$$

Thus,

$$\begin{aligned}
\mathbb{E}\|\mathbf{S}_{t+1}\|_F^2 &= \mathbb{E}\|(1 - \beta_{t+1})\mathbf{R}_{t+1} + \beta_{t+1}\mathbf{S}_t + \beta_{t+1}(\gamma_{t+1}\Delta_t - \mathbb{E}\Delta_t)\|_F^2 \\
&= \mathbb{E}\|(1 - \beta_{t+1})\mathbf{R}_{t+1} + \beta_{t+1}\mathbf{S}_t + \beta_{t+1}((\gamma_{t+1} - 1)\Delta_t + \Delta_t - \mathbb{E}\Delta_t)\|_F^2 \\
&= \underbrace{\mathbb{E}\|(1 - \beta_{t+1})\mathbf{R}_{t+1} + \beta_{t+1}\mathbf{S}_t + \beta_{t+1}(\Delta_t - \mathbb{E}\Delta_t)\|_F^2}_{\text{Term A.1}} \\
&\quad + \underbrace{\beta_{t+1}^2(\gamma_{t+1} - 1)^2\mathbb{E}\|\Delta_t\|_F^2}_{\text{Term A.2}} \\
&\quad + \underbrace{2\beta_{t+1}(\gamma_{t+1} - 1)\mathbb{E}\langle\Delta_t, (1 - \beta_{t+1})\mathbf{R}_{t+1} + \beta_{t+1}\mathbf{S}_t + \beta_{t+1}(\Delta_t - \mathbb{E}\Delta_t)\rangle}_{\text{Term A.3}}.
\end{aligned}$$

First, let's consider Term A.1:

$$\begin{aligned}
\text{A.1} &= \mathbb{E}\|(1 - \beta_{t+1})\mathbf{R}_{t+1} + \beta_{t+1}\mathbf{S}_t + \beta_{t+1}(\Delta_t - \mathbb{E}\Delta_t)\|_F^2 \\
&\overset{(\circ)}{=} \mathbb{E}\|(1 - \beta_{t+1})\mathbf{R}_{t+1} + \beta_{t+1}(\Delta_t - \mathbb{E}\Delta_t)\|_F^2 + \beta_{t+1}^2\mathbb{E}\|\mathbf{S}_t\|_F^2 \\
&\overset{(\star)}{\le} 2(1 - \beta_{t+1})^2\mathbb{E}\|\mathbf{R}_{t+1}\|_F^2 + 2\beta_{t+1}^2\mathbb{E}\|\Delta_t - \mathbb{E}\Delta_t\|_F^2 + \beta_{t+1}^2\mathbb{E}\|\mathbf{S}_t\|_F^2,
\end{aligned}$$

where $(\circ)$ holds because Assumption 3.4 implies $\mathbb{E}[\mathbf{R}_{t+1}] = 0$ and $\mathbb{E}[\Delta_t - \mathbb{E}\Delta_t] = 0$, making the cross-terms with $\mathbf{S}_t$ zero; and $(\star)$ follows from the inequality $\|\mathbf{A} + \mathbf{B}\|_F^2 \le 2\|\mathbf{A}\|_F^2 + 2\|\mathbf{B}\|_F^2$ for any $\mathbf{A}, \mathbf{B} \in \mathbb{R}^{m \times n}$.

Next, considering the sum of Terms A.2 and A.3, let us define

$$A_{t+1} = \frac{B_{t+1} + \beta_{t+1}(\mathbb{E}\|\Delta_t\|_F^2 - \|\mathbb{E}\Delta_t\|_F^2)}{\mathbb{E}\|\Delta_t\|_F^2}$$

$$B_{t+1} = (1 - \beta_{t+1})\mathbb{E}\langle\Delta_t, \mathbf{R}_{t+1}\rangle + \beta_{t+1}\mathbb{E}\langle\Delta_t, \mathbf{S}_t\rangle.$$

Then we have

$$
\begin{aligned}
\text{A.2} + \text{A.3} &= \beta_{t+1}^2(\gamma_{t+1}-1)^2\mathbb{E}\|\Delta_t\|_F^2 \\
&\quad + 2(\gamma_{t+1}-1)\beta_{t+1}\mathbb{E}\langle\Delta_t, (1-\beta_{t+1})\mathbf{R}_{t+1} + \beta_{t+1}\mathbf{S}_t + \beta_{t+1}(\Delta_t - \mathbb{E}\Delta_t)\rangle \\
&= \beta_{t+1}^2\mathbb{E}\|\Delta_t\|_F^2\left((\gamma_{t+1}-1)^2 + 2(\gamma_{t+1}-1)\frac{A_{t+1}}{\beta_{t+1}}\right) \\
&= \beta_{t+1}^2\mathbb{E}\|\Delta_t\|_F^2\left(\gamma_{t+1}-1+\frac{A_{t+1}}{\beta_{t+1}}\right)^2 - \beta_{t+1}^2\mathbb{E}\|\Delta_t\|_F^2\left(\frac{A_{t+1}}{\beta_{t+1}}\right)^2 \\
&= \mathbb{E}\|\Delta_t\|_F^2(\beta_{t+1}(1-\gamma_{t+1}) - A_{t+1})^2 - A_{t+1}^2\mathbb{E}\|\Delta_t\|_F^2 \\
&:= P_{t+1}.
\end{aligned}
$$

Therefore, combining the bounds:

$$
\begin{aligned}
\mathbb{E}\|\mathbf{S}_{t+1}\|_F^2 &\leq 2(1-\beta_{t+1})^2\mathbb{E}\|\mathbf{R}_{t+1}\|_F^2 + 2\beta_{t+1}^2\mathbb{E}\|\Delta_t - \mathbb{E}\Delta_t\|_F^2 + \beta_{t+1}^2\mathbb{E}\|\mathbf{S}_t\|_F^2 + P_{t+1} \\
&\overset{(\circ)}{\leq} 2(1-\beta_{t+1})^2\sigma^2 + 2\beta_{t+1}^2\mathbb{E}\|\Delta_t\|_F^2 + \beta_{t+1}^2\mathbb{E}\|\mathbf{S}_t\|_F^2 + P_{t+1} \\
&\overset{(\star)}{\leq} 2(1-\beta_{t+1})^2\sigma^2 + 2\beta_{t+1}^2 L^2\mathbb{E}\|\mathbf{X}_{t+1} - \mathbf{X}_t\|_F^2 + \beta_{t+1}^2\mathbb{E}\|\mathbf{S}_t\|_F^2 + P_{t+1},
\end{aligned}
$$

where $(\circ)$ uses Assumption 3.4 (bounded variance, i.e., $\mathbb{E}\|\mathbf{R}_{t+1}\|_F^2 \leq \sigma^2$) and the property that $\mathbb{E}\|\Delta_t - \mathbb{E}\Delta_t\|_F^2 \leq \mathbb{E}\|\Delta_t\|_F^2$; and $(\star)$ follows from Assumption 3.3, which implies $\|\Delta_t\|_F^2 \leq L^2\|\mathbf{X}_{t+1} - \mathbf{X}_t\|_F^2$.

Next, we set

$$
\gamma_{t+1} = 1 - \frac{A_{t+1}}{\beta_{t+1}} \text{ or } \gamma_{t+1} = \gamma = 1. \tag{16}
$$

Then,

$$
P_{t+1} = -\mathbb{E}\|\Delta_t\|_F^2 A_{t+1}^2 \leq 0 \text{ or } P_{t+1} = 0.
$$

This leads to the final result:

$$
\begin{aligned}
\mathbb{E}\|\mathbf{S}_{t+1}\|_F^2 &\leq \beta_{t+1}^2\mathbb{E}\|\mathbf{S}_t\|_F^2 + 2\beta_{t+1}^2 L^2\mathbb{E}\|\mathbf{X}_{t+1} - \mathbf{X}_t\|_F^2 + 2(1-\beta_{t+1})^2\sigma^2 \\
&\overset{(\circ)}{\leq} \beta_{t+1}\mathbb{E}\|\mathbf{S}_t\|_F^2 + 2\beta_{t+1}^2 L^2\mathbb{E}\|\mathbf{X}_{t+1} - \mathbf{X}_t\|_F^2 + 2(1-\beta_{t+1})^2\sigma^2,
\end{aligned}
$$

where $(\circ)$ follows from $\beta_{t+1}^2 \leq \beta_{t+1} \leq 1$. $\qquad\square$

### C.3 LEMMA C.3

**Lemma C.3.** *Let $\{\mathbf{S}_t\}_{t\geq 1}$ be a sequence of matrices satisfying the recursive inequality $\mathbb{E}\|\mathbf{S}_{t+1}\|_F^2 \leq (1-\eta_{t+1})\mathbb{E}\|\mathbf{S}_t\|_F^2 + 2\eta_{t+1}^2(2L^2 n + \sigma^2)$ for some constants $L, n, \sigma^2 > 0$. If we set the step size $\eta_t = t^{-2/3}$ and $\gamma = 1$, then we have the following upper bound on the time-averaged expectation:*

$$
\frac{1}{T}\sum_{t=1}^T \mathbb{E}\|\mathbf{S}_t\|_F^2 \leq \frac{4\sigma^2 + 8(2L^2 n + \sigma^2)(1 + \ln T)}{T^{2/3}}.
$$

*Proof.* The proof begins with the recursive inequality derived from a preceding Lemma C.2:

$$
\mathbb{E}\|\mathbf{S}_{t+1}\|_F^2 \leq (1-\eta_{t+1})\mathbb{E}\|\mathbf{S}_t\|_F^2 + 2\eta_t^2 L^2 n + 2\eta_{t+1}^2\sigma^2,
$$

where $\mathbf{S}_{t+1} = \nabla f(\mathbf{X}_{t+1}) - \mathbf{M}_{t+1}$. We have noticed the following facts

$$
\frac{1}{t^{2/3}} \leq \frac{2}{(t+1)^{2/3}}.
$$

Therefore , we have

$$
\mathbb{E}\|\mathbf{S}_{t+1}\|_F^2 \leq (1-\eta_{t+1})\mathbb{E}\|\mathbf{S}_t\|_F^2 + 2\eta_{t+1}^2(4L^2 n + \sigma^2).
$$

Let $A_t = \mathbb{E}\|\mathbf{S}_t\|_F^2$ and $B_{t+1} = 2\eta_{t+1}^2(4L^2n + \sigma^2)$. The inequality can be written as $A_{t+1} \le (1 - \eta_{t+1})A_t + B_{t+1}$. This structure allows us to apply a standard result Lemma C.1 which yields:

$$\sqrt{\eta_t}A_t \le 4\left(\frac{A_t}{\sqrt{\eta_t}} - \frac{A_{t+1}}{\sqrt{\eta_{t+1}}} + \frac{B_{t+1}}{\sqrt{\eta_{t+1}}}\right).$$

Then, we define $P_t = \frac{4A_t}{\sqrt{\eta_t}} = \frac{4\mathbb{E}\|\mathbf{S}_t\|_F^2}{\sqrt{\eta_t}}$. Substituting $P_t$ and the definition of $B_{t+1}$ into the inequality gives:

$$\sqrt{\eta_t}\mathbb{E}\|\mathbf{S}_t\|_F^2 \le P_t - P_{t+1} + \frac{4 \cdot 2\eta_{t+1}^2(4L^2n + \sigma^2)}{\sqrt{\eta_{t+1}}} = P_t - P_{t+1} + 8\eta_{t+1}^{3/2}(4L^2n + \sigma^2).$$

Now, we sum this inequality from $t = 1$ to $T$:

$$\sum_{t=1}^{T} \sqrt{\eta_t}\mathbb{E}\|\mathbf{S}_t\|_F^2 \le \sum_{t=1}^{T}\left(P_t - P_{t+1} + 8\eta_{t+1}^{3/2}(4L^2n + \sigma^2)\right)$$

$$= (P_1 - P_{T+1}) + 8(4L^2n + \sigma^2)\sum_{t=1}^{T}\eta_{t+1}^{3/2}.$$

Since $P_{T+1} \ge 0$, we can drop this term to simplify the bound. By setting the step size $\eta_t = t^{-2/3}$, we have $\eta_{t+1}^{3/2} = ((t+1)^{-2/3})^{3/2} = (t+1)^{-1}$. The summation becomes:

$$\sum_{t=1}^{T} \sqrt{\eta_t}\mathbb{E}\|\mathbf{S}_t\|_F^2 \le P_1 + 8(4L^2n + \sigma^2)\sum_{t=1}^{T}\frac{1}{t+1}$$

$$\le P_1 + 8(4L^2n + \sigma^2)\sum_{t=1}^{T}\frac{1}{t} \tag{17}$$

$$\overset{(\circ)}{\le} P_1 + 8(4L^2n + \sigma^2)(1 + \ln T),$$

where $(\circ)$ follows from the harmonic series, $\sum_{t=1}^{T}\frac{1}{t} \le 1 + \ln T$.

Finally, we establish the bound on the time-averaged expectation. With our choice of $\eta_t = t^{-2/3}$, we have $\sqrt{\eta_t} = t^{-1/3}$. Therefore:

$$\frac{1}{T}\sum_{t=1}^{T}\mathbb{E}\|\mathbf{S}_t\|_F^2 = \frac{1}{T}\sum_{t=1}^{T}t^{1/3} \cdot t^{-1/3}\mathbb{E}\|\mathbf{S}_t\|_F^2$$

$$= \frac{1}{T}\sum_{t=1}^{T}t^{1/3}\sqrt{\eta_t}\mathbb{E}\|\mathbf{S}_t\|_F^2$$

$$\le \frac{T^{1/3}}{T}\sum_{t=1}^{T}\sqrt{\eta_t}\mathbb{E}\|\mathbf{S}_t\|_F^2 \qquad \text{(since } t^{1/3} \le T^{1/3} \text{ for } t \le T)$$

$$\overset{(\circ)}{\le} \frac{T^{1/3}}{T}\left(P_1 + 8(4L^2n + \sigma^2)(1 + \ln T)\right)$$

$$= \frac{P_1 + 8(4L^2n + \sigma^2)(1 + \ln T)}{T^{2/3}},$$

where $(\circ)$ follows from the inequality (17).

Substituting $P_1 = 4\mathbb{E}\|\mathbf{S}_1\|_F^2/\sqrt{\eta_1} = 4\mathbb{E}\|\mathbf{S}_1\|_F^2$ completes the main proof. To establish the final bound, we now analyze the initial term $P_1$. Given the definition $\mathbf{S}_1 = \nabla f(\mathbf{X}_1) - \mathbf{M}_1$, we have:

$$P_1 = 4\mathbb{E}\|\nabla f(\mathbf{X}_1) - \mathbf{M}_1\|_F^2$$

$$= 4\mathbb{E}\|\nabla f(\mathbf{X}_1) - (1 - \beta_1 + \gamma_1\beta_1)\nabla f(\mathbf{X}_1; \xi_1)\|_F^2$$

$$= 4\mathbb{E}\|\nabla f(\mathbf{X}_1) - \nabla f(\mathbf{X}_1; \xi_1)\|_F^2$$

$$\overset{(\star)}{\le} 4\sigma^2.$$

The final inequality $(\star)$ holds by setting the parameter $\gamma_1 = \gamma = 1$. This choice nullifies the first term, as $(\gamma_1 - 1)^2 = 0$, and simplifies the coefficient of the variance to $(1 - \beta_1 + \beta_1)^2 = 1$. $\qquad\square$

## D  PROOFS OF THEOREM 3.2

*Proof.* According to Assumption 3.3, and based on the descent lemma, we have

$$f(\mathbf{X}_{t+1}) \leq f(\mathbf{X}_t) + \langle \nabla f(\mathbf{X}_t), \mathbf{X}_{t+1} - \mathbf{X}_t \rangle + \frac{L}{2}\|\mathbf{X}_{t+1} - \mathbf{X}_t\|_F^2$$

$$\leq f(\mathbf{X}_t) + \langle \mathbf{M}_t, \mathbf{X}_{t+1} - \mathbf{X}_t \rangle + \langle \nabla f(\mathbf{X}_t) - \mathbf{M}_t, \mathbf{X}_{t+1} - \mathbf{X}_t \rangle + \frac{L}{2}\|\mathbf{X}_{t+1} - \mathbf{X}_t\|_F^2$$

$$\overset{(\circ)}{\leq} f(\mathbf{X}_t) - \eta_t\|\mathbf{M}_t\|_* + \langle \nabla f(\mathbf{X}_t) - \mathbf{M}_t, \mathbf{X}_{t+1} - \mathbf{X}_t \rangle + \frac{L}{2}\|\mathbf{X}_{t+1} - \mathbf{X}_t\|_F^2$$

$$\leq f(\mathbf{X}_t) - \eta_t\|\mathbf{M}_t\|_* + \frac{1}{2\alpha}\|\nabla f(\mathbf{X}_t) - \mathbf{M}_t\|_F^2 + \frac{\alpha + L}{2}\|\mathbf{X}_{t+1} - \mathbf{X}_t\|_F^2$$

$$\overset{(\star)}{\leq} f(\mathbf{X}_t) - \eta_t\|\mathbf{M}_t\|_F + \frac{\sqrt{\eta_t}}{2L}\|\nabla f(\mathbf{X}_t) - \mathbf{M}_t\|_F^2 + \frac{\frac{L}{\sqrt{\eta_t}} + L}{2}\|\mathbf{X}_{t+1} - \mathbf{X}_t\|_F^2,$$

where $(\circ)$ holds because, by the definition of $\mathbf{O}_t$ and the property of norms $\|\mathbf{M}_t\|_* \geq \|\mathbf{M}_t\|_F$, we have:

$$\langle \mathbf{M}_t, \mathbf{X}_{t+1} - \mathbf{X}_t \rangle = -\eta_t\langle \mathbf{M}_t, \mathbf{O}_t \rangle = -\eta_t\langle \mathbf{M}_t, \mathbf{U}_r\mathbf{V}_r^\top \rangle = -\eta_t\|\mathbf{M}_t\|_* \leq -\eta_t\|\mathbf{M}_t\|_F,$$

and $(\star)$ holds by setting $\alpha = \frac{L}{\sqrt{\eta_t}}$. Thus, we have

$$\sum_{t=1}^T \eta_t\mathbb{E}\|\mathbf{M}_t\|_F \leq \sum_{t=1}^T \left(\mathbb{E}[f(\mathbf{X}_t)] - \mathbb{E}[f(\mathbf{X}_{t+1})]\right) + \sum_{t=1}^T \frac{\sqrt{\eta_t}}{2L}\mathbb{E}\|\nabla f(\mathbf{X}_t) - \mathbf{M}_t\|_F^2$$

$$+ \sum_{t=1}^T \frac{\frac{L}{\sqrt{\eta_t}} + L}{2}\mathbb{E}\|\mathbf{X}_{t+1} - \mathbf{X}_t\|_F^2$$

$$\leq f(\mathbf{X}_1) - f^* + \sum_{t=1}^T \frac{\sqrt{\eta_t}}{2L}\mathbb{E}\|\nabla f(\mathbf{X}_t) - \mathbf{M}_t\|_F^2 + \sum_{t=1}^T \frac{L(\eta_t^{3/2} + \eta_t^2)}{2}n$$

$$\leq f(\mathbf{X}_1) - f^* + \frac{1}{2L}\sum_{t=1}^T \sqrt{\eta_t}\mathbb{E}\|\mathbf{S}_t\|_F^2 + \frac{Ln}{2}\sum_{t=1}^T t^{-1} + \frac{Ln}{2}\sum_{t=1}^T t^{-4/3}$$

$$\overset{(\circ)}{\leq} f(\mathbf{X}_1) - f^* + \frac{4\sigma^2 + 8(4L^2n + \sigma^2)(1 + \ln T)}{2L}$$

$$+ \frac{Ln}{2}(1 + \ln T) + \frac{Ln}{2}\sum_{t=1}^T t^{-4/3}$$

$$\overset{(\star)}{\leq} f(\mathbf{X}_1) - f^* + \frac{2\sigma^2}{L} + 2Ln + 4(4Ln + \sigma^2L^{-1})(1 + \ln T) + \frac{Ln}{2}(1 + \ln T),$$

where $(\circ)$ uses Lemma C.2; $(\star)$ follows from the fact that $\sum_{t=1}^T \frac{1}{t^{4/3}} \leq 4$. Next, we let

$$G = f(\mathbf{X}_1) - f^* + \frac{2\sigma^2}{L} + \left(16Ln + 4\sigma^2L^{-1} + \frac{Ln}{2}\right)(1 + \ln T) + 2Ln.$$

Thus, we have

$$\frac{1}{T}\sum_{t=1}^T \mathbb{E}\|\mathbf{M}_t\|_F \leq \frac{1}{T}\sum_{t=1}^T \frac{t^{2/3}}{t^{2/3}}\mathbb{E}\|\mathbf{M}_t\|_F$$

$$\leq \frac{T^{2/3}}{T}\sum_{t=1}^T \frac{1}{t^{2/3}}\mathbb{E}\|\mathbf{M}_t\|_F = \frac{1}{T^{1/3}}\sum_{t=1}^T \eta_t\mathbb{E}\|\mathbf{M}_t\|_F$$

$$\leq \frac{G}{T^{1/3}}.$$

Next, we have

$$\frac{1}{T}\sum_{t=1}^{T}\mathbb{E}\|\nabla f(\mathbf{X}_t) - \mathbf{M}_t\|_F \overset{(\circ)}{\leq} \sqrt{\frac{1}{T}\sum_{t=1}^{T}\mathbb{E}\|\nabla f(\mathbf{X}_t) - \mathbf{M}_t\|_F^2}$$

$$\overset{(\star)}{\leq} \sqrt{\frac{4\sigma^2 + 8(4L^2 n + \sigma^2)(1 + \ln T)}{T^{2/3}}}.$$

where $(\circ)$ uses Jensen's inequality; $(\star)$ uses Lemma C.3 by letting $\mathbf{S}_t = \nabla f(\mathbf{X}_t) - \mathbf{M}_t$. Thus, we have

$$\frac{1}{T}\sum_{t=1}^{T}\mathbb{E}\|\nabla f(\mathbf{X}_t)\|_F \leq \frac{1}{T}\sum_{t=1}^{T}\mathbb{E}\|\nabla f(\mathbf{X}_t) - \mathbf{M}_t\|_F + \frac{1}{T}\sum_{t=1}^{T}\mathbb{E}\|\mathbf{M}_t\|_F$$

$$\leq \sqrt{\frac{4\sigma^2 + 8(4L^2 n + \sigma^2)(1 + \ln T)}{T^{2/3}}} + \frac{G}{T^{1/3}}$$

$$= \mathcal{O}\left(\frac{\ln T}{T^{1/3}}\right).$$

This completes the proof. □

# E LEMMAS FOR THEOREMS 3.3 AND 3.4

## E.1 LEMMA E.1

**Lemma E.1.** *Let $\{\Gamma_t\}_{t\geq 1}$ be a non-negative sequence whose partial sum $D_t = \sum_{t=1}^{T}\Gamma_t$ is bounded by $D_t \leq A_1 \ln T + A_2$ for some positive constants $A_1$ and $A_2$. Define the function $f(t) = \frac{t^{1-p}}{\ln t}$ where $p \in (0,1)$. Fix $T_0 = 2e^{1/(1-p)}$, where $e$ is the base of the natural logarithm. Then, the following inequality holds for all $T > T_0$:*

$$\sum_{t=T_0}^{T-1}\Gamma_t f(t) \leq (A_1 + A_2)(T-1)^{1-p}.$$

*Proof.* We choose an integer $T_0 = 2e^{1/(1-p)} > e^{1/(1-p)}$ large enough such that for all $t \geq T_0$, the function $f(t)$ is positive and monotonically increasing. We use summation by parts to bound the sum $\sum_{t=T_0}^{T-1}\Gamma_t f(t)$:

$$\sum_{t=T_0}^{T-1}\Gamma_t f(t) = \sum_{t=T_0}^{T-1}(D_t - D_{t-1})f(t) = D_{T-1}f(T-1) - D_{T_0-1}f(T_0) - \sum_{t=T_0}^{T-1}D_{t-1}(f(t) - f(t-1)).$$

Since $D_{t-1} \geq 0$ and $f(t)$ is monotonically increasing for $t \geq T_0$ (i.e., $f(t) - f(t-1) \geq 0$), the last term, $\sum_{t=T_0}^{T-1}D_{t-1}(f(t) - f(t-1))$, is non-negative. Therefore, we have:

$$\sum_{t=T_0}^{T-1}\Gamma_t f(t) \leq D_{T-1}f(T-1).$$

Substituting the given bound on the partial sums, $D_{T-1} \leq A_1 \ln(T-1) + A_2$, we obtain:

$$\sum_{t=T_0}^{T-1}\Gamma_t f(t) \leq (A_1 \ln(T-1) + A_2)\frac{(T-1)^{1-p}}{\ln(T-1)}$$

$$= \left(A_1 + \frac{A_2}{\ln(T-1)}\right)(T-1)^{1-p}$$

$$\overset{(\circ)}{\leq} (A_1 + A_2)(T-1)^{1-p},$$

where $(\circ)$ follows from $\ln(T-1) \geq 1$ for sufficiently large $T$.

Consequently, for any given $\epsilon > 0$, the expression can be bounded by $(A_1 + A_2 + \epsilon)T^{1-p}$ for sufficiently large $T$. □

## E.2 LEMMA E.2

**Lemma E.2.** *Suppose a positive sequence $\{\Delta_t\}_{t \geq 1}$ satisfies the following recursive inequality:*

$$\Delta_{t+1} \leq \Delta_t - \frac{\sqrt{2\mu}}{t^p}\sqrt{\Delta_t} + \Gamma_t,$$

*where $p \in (0,1)$ and $\mu > 0$ are constants. The non-negative noise sequence $\{\Gamma_t\}_{t \geq 1}$ satisfies the condition from Lemma E.1 that $\sum_{t=1}^{T} \Gamma_t = A_1 \ln T + A_2$. Then, the sequence $\Delta_t$ converges to 0 with the following rate:*

$$\Delta_t \leq \frac{\mathcal{A}^2(1-p)^2}{C^2} \cdot \frac{(\ln t)^2}{t^{2(1-p)}},$$

*where $\mathcal{A} := A_1 + A_2$ and $C = \frac{\sqrt{2\mu}}{2}$.*

*Proof.* We define the variable $G_t = \sqrt{\Delta_t}$.

Since $\Delta_t > 0$, it follows that $G_t > 0$. The function $f(x) = \sqrt{x}$ is concave, which implies $\sqrt{y} - \sqrt{x} \leq \frac{1}{2\sqrt{x}}(y - x)$ for any $x, y > 0$. By setting $y = \Delta_{t+1}$ and $x = \Delta_t$, we obtain:

$$G_{t+1} - G_t \leq \frac{\Delta_{t+1} - \Delta_t}{2G_t}.$$

Substituting the original recurrence relation $\Delta_{t+1} - \Delta_t \leq -\frac{\sqrt{2\mu}}{t^p}\sqrt{\Delta_t} + \Gamma_t$ and noting that $\sqrt{\Delta_t} = G_t$, we have:

$$G_{t+1} - G_t \leq \frac{1}{2G_t}\left(-\frac{\sqrt{2\mu}}{t^p}G_t + \Gamma_t\right).$$

Letting $C = \frac{\sqrt{2\mu}}{2}, T_0 = 2e^{1/(1-p)}$, we arrive at the core recurrence for $G_t$:

$$G_{t+1} \leq G_t - \frac{C}{t^p} + \frac{\Gamma_t}{2G_t} \tag{18}$$

For $p \in (0,1)$, we will prove that $G_t \leq \frac{\mathcal{A}(1-p)}{C} \cdot \frac{\ln t}{t^{1-p}}$ by contradiction.

Assume the proposition is false. This means that for any constant $A > 0$, there exist infinitely many time steps $t$ such that $G_t > B_t$, where $B_t = A \cdot \frac{\ln t}{t^{1-p}}$.

We choose a constant $A$ large enough (the specific condition will be derived later). By our assumption, there must exist an arbitrarily large time $T$ such that $G_T > B_T$. Consider an interval $[T_0, T-1]$ and assume that $G_t > B_t$ for all $t$ in this interval.

Summing the inequality (18) from $t = T_0$ to $T-1$ gives:

$$G_T - G_{T_0} \leq \sum_{t=T_0}^{T-1}(G_{t+1} - G_t) \leq \sum_{t=T_0}^{T-1}\left(-\frac{C}{t^p} + \frac{\Gamma_t}{2G_t}\right).$$

Over this interval, since $G_t > B_t = A\frac{\ln t}{t^{1-p}}$, we have $\frac{1}{G_t} < \frac{1}{B_t} = \frac{t^{1-p}}{A \ln t}$. Substituting this bound yields:

$$G_T < G_{T_0} - C\underbrace{\sum_{t=T_0}^{T-1}\frac{1}{t^p}}_{V_1} + \frac{1}{2A}\underbrace{\sum_{t=T_0}^{T-1}\frac{\Gamma_t t^{1-p}}{\ln t}}_{V_2}.$$

Next, we analyze the asymptotic behavior of the two sums, $V_1$ and $V_2$.

For the negative drift term $V_1$, we use an integral approximation:

$$\sum_{t=T_0}^{T-1}\frac{1}{t^p} \geq \int_{T_0}^{T} x^{-p}dx = \frac{1}{1-p}(T^{1-p} - T_0^{1-p}).$$

The leading term of $V_1$ is thus on the order of $\frac{C}{1-p}T^{1-p}$.

For the positive noise term $V_2$, we use the condition on the partial sums of $\Gamma_t$, $S_t = \sum_{k=1}^t \Gamma_k \le A_1 \ln t + A_2$. Applying summation by parts (in a manner analogous to Lemma E.1), the dominant behavior of the sum in $V_2$ is given by $S_{T-1} \cdot \frac{(T-1)^{1-p}}{\ln(T-1)}$. This can be bounded by $((A_1 + A_2)\ln(T-1))\frac{(T-1)^{1-p}}{\ln(T-1)} = (A_1 + A_2)(T-1)^{1-p}$. For simplicity in asymptotics, let us denote the effective constant as $\mathcal{A} := A_1 + A_2$. The leading behavior of $V_2$ is thus on the order of $\frac{\mathcal{A}}{2A}T^{1-p}$.

Substituting the leading terms of these bounds back into the inequality for $G_T$:

$$G_T < G_{T_0} - \frac{C}{1-p}T^{1-p} + \frac{\mathcal{A}}{2A}T^{1-p} = G_{T_0} + T^{1-p}\left(\frac{\mathcal{A}}{2A} - \frac{C}{1-p}\right).$$

We now choose the constant $A$ such that the coefficient of the leading term is negative:

$$\frac{\mathcal{A}}{2A} - \frac{C}{1-p} < 0 \implies A > \frac{\mathcal{A}(1-p)}{2C}.$$

With an $A$ satisfying this condition, for a sufficiently large $T$, the negative term proportional to $T^{1-p}$ will dominate, forcing the right-hand side to become negative. This contradicts the fact that $G_T = \sqrt{\Delta_T}$ must be positive.

This contradiction proves that our assumption that $G_t$ can remain above $B_t$ for an arbitrarily long interval is false. Therefore, there must exist a time $T_A$ such that for all $t > T_A > T_0 = 2e^{1/(1-p)}$, $G_t \le B_t$. This establishes that $G_t = \mathcal{O}\left(\frac{\ln t}{t^{1-p}}\right)$.

Finally, by choosing the constant $A = \frac{\mathcal{A}(1-p)}{C}$ , squaring both sides of this result yields the convergence rate for $\Delta_t$:

$$\Delta_t = G_t^2 \le \frac{\mathcal{A}^2(1-p)^2}{C^2} \cdot \frac{(\ln t)^2}{t^{2(1-p)}}$$
$$= \mathcal{O}\left(\frac{(\ln t)^2}{t^{2(1-p)}}\right).$$

$\square$

## F    PROOFS OF THEOREM 3.3

*Proof.* By Theorem 3.1 inequality (7), we have

$$\mathbb{E}[f(\mathbf{X}_{t+1})] \le \mathbb{E}[f(\mathbf{X}_t)] - \eta_t \mathbb{E}[\|\nabla f(\mathbf{X}_t)\|_F] + \Gamma_t \tag{19}$$

where

$$\Gamma_t = \frac{\eta_t^{2/3}}{L}\mathbb{E}\|\mathbf{M}_t - \nabla f(\mathbf{X}_t)\|_F^2 + (L/2 + Ln)\eta_t^{4/3}. \tag{20}$$

Case 1: $\gamma = 0$.

Let $\eta_t = t^{-3/4}, \beta_t = 1 - t^{-1/2}, \alpha_t = t^{-1/2}$. According to Theorem 3.1 inequality (9), we have

$$\sum_{t=1}^T \Gamma_t \le A_1 \ln T + A_2,$$

where $A_1 = 2L^{-1}\sigma^2 + 4\sqrt{2}Ln + Ln + L/2$ and $A_2 = 4L^{-1}\sigma^2 + 4\sqrt{2}Ln + Ln + L/2$.

Case 2: $\gamma \ne 0$.

Let $\eta_t = t^{-3/4}, \beta_t = 1 - (1+t)^{-1/2}, \alpha_t = t^{-1/2}$. According to Theorem 3.1 inequality (13), we have

$$\sum_{t=1}^T \Gamma_t \le A_1 \ln T + A_2,$$

where $A_1 = 4L^{-1}\sigma^2 + 8\sqrt{2}Ln + Ln + L/2$ and $A_2 = 10L^{-1}\sigma^2 + 8\sqrt{2}Ln + Ln + L/2$.

Then, we define $\Delta_t = \mathbb{E}[f(\mathbf{X}_t)] - f^*$. Applying Assumption 3.5 to inequality (19), we have

$$\Delta_{t+1} \leq \Delta_t - \eta_t\sqrt{2\mu\Delta_t} + \Gamma_t$$
$$= \Delta_t - \frac{\sqrt{2\mu}}{t^{3/4}}\sqrt{\Delta_t} + \Gamma_t.$$

By Lemma E.2, let $G_t = \sqrt{\Delta_t}$, $C = \sqrt{2\mu}/2$ and $p = 3/4$.

$$G_{t+1} \leq G_t - \frac{C}{t^{3/4}} + \frac{\Gamma_t}{2G_t}.$$

Setting $p = 3/4$, for any constant $A > \frac{(A_1+A_2)(1-p)}{2C} = \frac{A_1+A_2}{8C}$, there exists a time step $T_A$ such that for all $t > T_A$, we have $G_t \leq A \cdot \frac{\ln t}{t^{1/4}}$. By choosing $A = \frac{\mathcal{A}}{4C}$, we satisfy the condition $A > \frac{\mathcal{A}}{8C}$, which ensures this bound holds. Thus, we obtain the convergence rate for $\Delta_t$:

$$\Delta_t \leq \frac{\mathcal{A}^2(1-3/4)^2}{C^2} \cdot \frac{(\ln t)^2}{t^{2(1-3/4)}} = \frac{\mathcal{A}^2}{8\mu} \cdot \frac{(\ln t)^2}{t^{1/2}}$$

We can now summarize the results for both scenarios. In either case ($\gamma = 0$ or $\gamma \neq 0$), the analysis yields the same asymptotic convergence rate. For any number of iterations $T > T_0 = 2e^4$, the analysis leads to the same asymptotic upper bound:

$$\mathbb{E}[f(\mathbf{X}_{T+1})] - f^* \leq \frac{\mathcal{A}^2}{8\mu} \cdot \frac{(\ln T)^2}{T^{1/2}} = \mathcal{O}\left(\frac{(\ln T)^2}{T^{1/2}}\right).$$

The specific definition of $\mathcal{A}$ is as follows: (i) Case 1 ($\gamma = 0$): $\mathcal{A} = 6L^{-1}\sigma^2 + 8\sqrt{2}Ln + 2Ln + L$. (ii) Case 2 ($\gamma \neq 0$): $\mathcal{A} = 14L^{-1}\sigma^2 + 16\sqrt{2}Ln + 2Ln + L$.

This completes the proof. $\qquad\square$

## G  PROOFS OF THEOREM 3.4

*Proof.* According to Theorem 3.2, we have

$$f(\mathbf{X}_{t+1}) \leq f(\mathbf{X}_t) - \eta_t\|\mathbf{M}_t\|_F + \frac{\sqrt{\eta_t}}{2L}\|\nabla f(\mathbf{X}_t) - \mathbf{M}_t\|_F^2 + \frac{\frac{L}{\sqrt{\eta_t}} + L}{2}\|\mathbf{X}_{t+1} - \mathbf{X}_t\|_F^2.$$

Since $\eta_t \leq 1$, we have $\frac{L}{\sqrt{\eta_t}} \geq L$. Thus, taking the expectation yields

$$\mathbb{E}[f(\mathbf{X}_{t+1})] \leq \mathbb{E}[f(\mathbf{X}_t)] - \eta_t\mathbb{E}[\|\mathbf{M}_t\|_F] + \frac{\sqrt{\eta_t}}{2L}\mathbb{E}[\|\nabla f(\mathbf{X}_t) - \mathbf{M}_t\|_F^2] + \frac{L}{\sqrt{\eta_t}}\mathbb{E}[\|\mathbf{X}_{T+1} - \mathbf{X}_t\|_F^2]$$

$$\leq \mathbb{E}[f(\mathbf{X}_t)] - \eta_t\mathbb{E}[\|\mathbf{M}_t\|_F] + \frac{\sqrt{\eta_t}}{2L}\mathbb{E}[\|\nabla f(\mathbf{X}_t) - \mathbf{M}_t\|_F^2] + Ln\eta_t^{3/2}$$

$$\overset{(\circ)}{\leq} \mathbb{E}[f(\mathbf{X}_t)] - \eta_t\mathbb{E}[\|\nabla f(\mathbf{X}_t)\|_F] + \eta_t\mathbb{E}[\|\nabla f(\mathbf{X}_t) - \mathbf{M}_t\|_F]$$
$$+ \frac{\sqrt{\eta_t}}{2L}\mathbb{E}[\|\nabla f(\mathbf{X}_t) - \mathbf{M}_t\|_F^2] + Ln\eta_t^{3/2}$$

$$\overset{(\star)}{\leq} \mathbb{E}[f(\mathbf{X}_t)] - \eta_t\mathbb{E}[\|\nabla f(\mathbf{X}_t)\|_F] + \frac{1}{2\epsilon}\eta_t^2 + \frac{\epsilon}{2}\mathbb{E}[\|\nabla f(\mathbf{X}_t) - \mathbf{M}_t\|_F^2]$$
$$+ \frac{\sqrt{\eta_t}}{2L}\mathbb{E}[\|\nabla f(\mathbf{X}_t) - \mathbf{M}_t\|_F^2] + Ln\eta_t^{3/2}$$

$$\overset{(\bullet)}{=} \mathbb{E}[f(\mathbf{X}_t)] - \eta_t\mathbb{E}[\|\nabla f(\mathbf{X}_t)\|_F]$$
$$+ \underbrace{\frac{L\eta_t^{3/2}}{2} + \left(\frac{\epsilon}{2} + \frac{\sqrt{\eta_t}}{2L}\right)\mathbb{E}[\|\nabla f(\mathbf{X}_t) - \mathbf{M}_t\|_F^2] + Ln\eta_t^{3/2}}_{\Gamma_t},$$

where ($\circ$) follows from the reverse triangle inequality $-\|\mathbf{M}_t\|_F \leq \|\nabla f(\mathbf{X}_t) - \mathbf{M}_t\|_F - \|\nabla f(\mathbf{X}_t)\|_F$; ($\star$) applies Young's inequality to the term $\eta_t\mathbb{E}[\|\nabla f(\mathbf{X}_t) - \mathbf{M}_t\|_F]$; and ($\bullet$) collects the residual terms into $\Gamma_t$. By setting $\epsilon = \frac{\sqrt{\eta_t}}{L}$, we can simplify the expression for $\Gamma_t$:

$$\Gamma_t = \frac{\sqrt{\eta_t}}{L}\mathbb{E}[\|\mathbf{S}_t\|_F^2] + (L/2 + Ln)\eta_t^{3/2}, \tag{21}$$

where $\mathbf{S}_t = \nabla f(\mathbf{X}_t) - \mathbf{M}_t$. The sum of these terms can be bounded. Let $A_1 = 33Ln + 8L^{-1}\sigma^2 + L/2$ and $A_2 = 33Ln + 12L^{-1}\sigma^2 + L/2$. We have

$$\sum_{t=1}^{T}\Gamma_t = \frac{1}{L}\sum_{t=1}^{T}\sqrt{\eta_t}\mathbb{E}[\|\mathbf{S}_t\|_F^2] + \sum_{t=1}^{T}(L/2 + Ln)\eta_t^{3/2}$$

$$\leq \frac{1}{L}\sum_{t=1}^{T}\sqrt{\eta_t}\mathbb{E}[\|\mathbf{S}_t\|_F^2] + (L/2 + Ln)\sum_{t=1}^{T}\frac{1}{t}$$

$$\overset{(\circ)}{\leq} \frac{4\sigma^2 + 8(4L^2n + \sigma^2)(1 + \ln T)}{L} + (L/2 + Ln)(1 + \ln T)$$

$$= A_1\ln T + A_2,$$

where $(\circ)$ holds by Lemma C.3 inequality (17).

Next, let $\Delta_t = \mathbb{E}[f(\mathbf{X}_t)] - f^*$. By applying the Assumption 3.5, $\|\nabla f(\mathbf{X}_t)\|_F^2 \geq 2\mu(f(\mathbf{X}_t) - f^*)$, and Jensen's inequality to the main recurrence, we obtain

$$\Delta_{t+1} \leq \Delta_t - \eta_t\mathbb{E}[\|\nabla f(\mathbf{X}_t)\|_F] + \Gamma_t$$

$$\leq \Delta_t - \eta_t\sqrt{2\mu\mathbb{E}[f(\mathbf{X}_t) - f^*]} + \Gamma_t$$

$$\leq \Delta_t - \eta_t\sqrt{2\mu\Delta_t} + \Gamma_t.$$

Setting the step size $\eta_t = 1/t^{2/3}$, the recurrence becomes

$$\Delta_{t+1} \leq \Delta_t - \frac{\sqrt{2\mu}}{t^{2/3}}\sqrt{\Delta_t} + \Gamma_t.$$

This recurrence is in the form required by Lemma E.2. Let $G_t = \sqrt{\Delta_t}$, $C = \sqrt{2\mu}/2$ and $p = 2/3$. According to Lemma E.2, for any constant $A > \frac{(A_1 + A_2)(1-p)}{2C} = \frac{A_1 + A_2}{6C}$, there exists a time $T_A$ such that for all $t > T_A$, $G_t \leq A \cdot \frac{\ln t}{t^{1/3}}$. We choose a specific constant $A = \frac{\mathcal{A}}{3C}$, where $\mathcal{A} := A_1 + A_2 = 20L^{-1}\sigma^2 + 66Ln + L$. Squaring both sides of the inequality for $G_t$ establishes the convergence rate for $\Delta_t = G_t^2$:

$$\Delta_t \leq A^2 \cdot \frac{(\ln t)^2}{t^{2/3}} = \frac{\mathcal{A}^2}{9C^2}\frac{(\ln t)^2}{t^{2/3}} = \frac{2\mathcal{A}^2}{9\mu}\frac{(\ln t)^2}{t^{2/3}}$$

Therefore, for any number of iterations $T > T_0 = 2e^3$, the expected suboptimality has the following asymptotic bound:

$$\mathbb{E}[f(\mathbf{X}_{T+1})] - f^* \leq \frac{2\mathcal{A}^2}{9\mu} \cdot \frac{(\ln T)^2}{T^{2/3}} = \mathcal{O}\left(\frac{(\ln T)^2}{T^{2/3}}\right).$$

This completes the proof. $\qquad\square$

## H    EXPERIMENTAL DETAILS

### H.1    TRAINING ON CIFAR10

The ResNet18 model He et al. (2016) undergoes pretraining on the CIFAR-10 dataset with comprehensive hyperparameter specifications provided in Table 2. For each optimizer, the learning rate is selected via a grid search over the set $\{1 \times 10^{-4}, 5 \times 10^{-4}, 10^{-3}, 5 \times 10^{-3}, 10^{-2}, 5 \times 10^{-2}, 10^{-1}\}$. To ensure a robust comparison, all experiments are repeated over five different random seeds, and we report the mean results with one standard deviation shaded. For the ResNet-18 model, we reshape each convolutional kernel into a 2D matrix and apply a Muon-type optimizer to these parameters, while the remaining 1D vector parameters are optimized with AdamW.

Table 2: Hyperparameters used for training ResNet18 on CIFAR10

|  | SGD | Adam | Muon | Muon-MVR1 | Muon-MVR2 |
|---|---|---|---|---|---|
| Model Size | | | 42.7M | | |
| Training Epochs | | | 100 | | |
| Batch Size | | | 128 | | |
| Learning Rate | 0.1 | 0.01 | | 0.05 | |
| Learning Rate Scheduling | | | cosine to 10% | | |
| Numerical precision | | | float32 | | |
| Weight Decay | | | 0.01 | | |
| $(\beta_1, \beta_2)$ | ✗ | (0.9,0.999) | | ✗ | |
| Muon-Momentum | ✗ | ✗ | | 0.9 | |
| Gamma | ✗ | ✗ | ✗ | | 0.1 |

Table 3: Hyperparameters used for training LLaMA2-130M on C4

| Hyper-parameter | AdamW | MARS-AdamW | Muon | Muon-MVR1 | Muon-MVR2 |
|---|---|---|---|---|---|
| Max Learning Rate | 8e-4 | 1e-3 | 8e-4 | 2e-3 | 2e-3 |
| Warmup Ratio | | | 0.1 | | |
| Batch Size | | | 128 | | |
| Maximum Length | | | 4096 | | |
| Weight Decay | | | 0.1 | | |
| $(\beta_1, \beta_2)$ | | | (0.9,0.98) | | |
| Muon-Momentum | ✗ | ✗ | | 0.95 | |
| Gamma | ✗ | 0.025 | ✗ | | 0.05 |

## H.2 PRETRAINING ON C4

▶ **Experimental setup.** We use 48 Ascend 910C (64GB) NPUs for all experiments. For the additional experiments, we conduct hyperparameter sweeps for LLaMA2-130M Touvron et al. (2023) trained for 12B tokens on the C4 (Colossal Clean Crawled Corpus) dataset Raffel et al. (2020). For all optimizers (AdamW, MARS-AdamW, Muon, Muon-MVR1, and Muon-MVR2), we keep the model architecture and training data fixed. For LLaMA2-130M, we use a global batch size of 128 and a maximum sequence length of 4096. For each optimizer we train all configurations on the 4× Chinchilla data about 12B tokens for 20,000 steps and select the hyperparameters that achieve the best validation performance.

▶ **Hyperparameter search.** Comprehensive experimental specifications are tabulated in Table 3. For AdamW, we set $(\beta_1, \beta_2) = (0.9, 0.98)$, $\epsilon = 10^{-8}$, and a weight decay of 0.1. The learning rate $\eta$ is selected from the set $\{3e-4, 5e-4, 8e-4, 1e-3, 2e-3, 4e-3, 6e-3, 8e-3\}$. For MARS-AdamW, we use the same $(\beta_1, \beta_2) = (0.9, 0.98)$, $\epsilon = 10^{-8}$, and weight decay of 0.1, and we search over the same learning-rate set for $\eta$. In addition, we sweep over the algorithmic parameter $\gamma \in \{0.01, 0.025, 0.05\}$.

For Muon, we set $\beta = 0.95$ and a weight decay of 0.1, and we again choose the learning rate $\eta$ from $\{3e-4, 5e-4, 8e-4, 1e-3, 2e-3, 4e-3, 6e-3, 8e-3\}$. Muon-MVR1 and Muon-MVR2 use the same settings $\beta = 0.98$ and weight decay of 0.1, and share the same learning-rate search space as Muon. For both Muon-MVR1 and Muon-MVR2, we additionally perform a sweep over $\gamma \in \{0.01, 0.025, 0.05\}$. We use the Muon implementation from Moonlight[1]. For LLaMA model, we optimize all 2D matrix parameters (except the embedding layers) using a Muon-type optimizer, while the remaining 1D vector parameters (including the embedding layers) are optimized with AdamW.

## I LLM USAGE

A large language model (LLM) was used to aid in the polishing of this paper's writing.

---

[1] https://github.com/MoonshotAI/Moonlight

