# OpenReview forum: "On the Convergence of Muon and Beyond"
_ICLR.cc/2026/Conference — Submitted to ICLR 2026_

### Official Review · Reviewer_DRyM · 2025-10-26

**Soundness:** 2
**Presentation:** 3
**Contribution:** 2
**Rating:** 2
**Confidence:** 4

**Summary:**

This work studies the convergence properties of the Muon optimizer and two variants under suitable assumptions. In particular, for the stochastic non-convex setting, the authors provide convergence guarantees for the standard Muon algorithm and demonstrate improved guarantees for two variance-reduced versions. Additionally, the convergence of Muon and its two variants is examined under the Polyak-Lojasiewicz condition. Finally, the authors present experimental results on vision and language tasks, evaluating Muon and its two variants alongside other popular optimizers.

**Strengths:**

- The paper is for the most part clearly written and its structure is easy to follow.
- The last-iterate convergence results under the PL condition are interesting.
- The convergence guarantees of the approximate variance reduction scheme (MVR1) without the additive error-term improve previous results.
- The experimental results showcase the improved performance of the proposed algorithms alongside other popular optimizers.

**Weaknesses:**

- The proposed algorithmic variants of Muon are not novel. Prior work [1] has proposed Shampoo (a generalization of Muon) with both one-batch and two-batch momentum variance reduction and also uses the additional $\gamma$ parameter to control the variance-reduction term. In addition, the algorithm in [1] uses weight decay. However, it is important to note that [1] does not provide convergence guarantees for their algorithm.
- The convergence of standard Muon in the non-convex setting has been studied in previous works, as also mentioned in the paper. For example, [2] shows an iteration complexity of $O(T^{-1/4})$ with a batch size of $O(1)$. In contrast, this works shows an iteration complexity of $O(\log(T)/ T^{1/4})$, that is with an additional $\log(T)$ factor. Additionally, [3] also shows that the stochastic first-order oracle calls in Muon are $O( T^{-1/4})$. However, [3] studies Muon with weight decay.
- The convergence of the variant MVR2 (with $\gamma =1$ and additional weight decay) in the non-convex setting was also studied in a previous work [3]. This work claims that their advantage lies in the use of a batch size of $O(1)$, as opposed to [3]. However, [3] demonstrates an interaction complexity of $O(T^{-1/3})$ using a large batch size only in the first iteration and for all subsequent iterations the batch size is set to $O(1)$. Therefore, the amortized average batch size is also $O(1)$. Furthermore, the complexity in [3] does not involve a logarithmic factor, as opposed to this work, which demonstrates a complexity of $O(\log(T)/ T^{1/3})$. Additionally, Assumption 3.3 of this work is stronger than Assumption 2 in [3].
- This work claims that the MVR1 variant is equivalent to Muon with Nesterov momentum for a specific choice of parameters (equation (4)). I do not see how this is possible, though I may be wrong.
- Some parts of the non-ergodic section are a bit unclear. For example, $\Gamma_t$ is not specified in Remark 3.5.
- The introduction of an additional hyperparameter $\gamma$ makes tuning even more tedious in practice.
- The optimizers in the experiments use weight decay, which is not supported by the theoretical results.
- It is not specified how the variance parameter $\gamma$, momentum parameter $\beta$ and weight decay parameter were tuned in the experiments.
- One minor presentation issue is that standard Muon is referred to as Muon-MVR1 with $\gamma=0$, which is a bit misleading, since in that case no variance reduction is applied.


[1] MARS: Unleashing the Power of Variance Reduction for Training Large Models - Yuan et al.

[2] Convergence Bound and Critical Batch Size of Muon Optimizer - Sato et al.

[3] Lions and Muons: Optimization via Stochastic Frank-Wolfe - Sfyraki and Wang

**Questions:**

- I understand that $\gamma$ is used to control the variance reduction term, as in [1]. However, this scheme is different from the originally proposed STORM scheme in [4]. I also see that in practice $\gamma$ is set to a very small value, which effectively reduces the impact of the variance reduction. Can the authors explain the motivation for including $\gamma$?
- Can the authors elaborate on how equation (4) is obtained from Algorithm 1?



[1] MARS: Unleashing the Power of Variance Reduction for Training Large Models - Yuan et al.

[4] Momentum-Based Variance Reduction in Non-Convex SGD - Cutkosky and Orabona

---

> ### Author Response · Authors · 2025-11-19
>
> We appreciate the reviewers' feedback and provide a detailed, point-by-point response below.
>
> ---
>
> **Q1.** The proposed algorithmic variants of Muon are not novel. Prior work [1] has proposed Shampoo (a generalization of Muon) with both one-batch and two-batch momentum variance reduction and also uses the additional $\gamma$ parameter to control the variance-reduction term. In addition, the algorithm in [1] uses weight decay. However, it is important to note that [1] does not provide convergence guarantees for their algorithm.
>
> **A1.** The goal of this paper is not to propose a completely new optimization algorithm, but to systematically study the convergence properties of a class of Muon-style algorithms, as clearly stated in both the title and the introduction. The variance-reduction scheme we adopt has already been introduced in prior work such as STORM; our work follows this line of research and provides a unified convergence analysis for this family of methods.
>
> Reference [1] mainly discusses how the MARS method can be used as a plug-in within Muon, but does not provide convergence guarantees for MARS-Muon, or even for standard Muon. In contrast, our main contribution (already clearly stated in the Introduction) is to give a systematic convergence analysis for Muon and its variance-reduced variants, and to validate the practical effectiveness of these algorithms through experiments, rather than to pursue novelty in the algorithmic form.
>
> [1]MARS: Unleashing the Power of Variance Reduction for Training Large Models - Yuan et al.
>
> ---
>
> **Q2.** The convergence of standard Muon in the non-convex setting has been studied in previous works, as also mentioned in the paper. For example, [2] shows an iteration complexity of $O(T^{-1/4})$ with a batch size of  $O(1)$. In contrast, this works shows an iteration complexity of $O(\log(T)/T^{1/4})$, that is with an additional $\log(T)$ factor. Additionally, [3] also shows that the stochastic first-order oracle calls in Muon are $O(T^{-1/4})$ . However, [3] studies Muon with weight decay.
>
> **A2.**  As shown in Table 1 of this paper (lines 76–77) and Table 1 of [2], the upper bound of $\frac{1}{T}\sum _ {t\in[T]}\mathbb{E}||\nabla f(W _ t)|| _ F$ does not converge to 0, but is  $\mathcal{O}\left(\frac{1}{T}+\frac{1-\beta}{b}+n\right)$ and $\mathcal{O}\left(\frac{1}{T}+\frac{(2\beta+1)(1-\beta)}{2}\cdot\frac{1}{b}+n\right)$ , where $n$ is the matrix dimension and $b$ is the batch size. When $T\to \infty$, since $b$ and $n$ are fixed, the upper bound of  $\frac{1}{T}\sum _ {t\in[T]}\mathbb{E}||\nabla f(W_t)|| _ F$ does not  converge to 0, but instead converges to $\mathcal{O}\left(\frac{1-\beta}{b}+n\right)$ or $\mathcal{O}\left(\frac{(2\beta+1)(1-\beta)}{2}\cdot\frac{1}{b}+n\right)$, respectively. Therefore, the convergence scale is not the same as in our work, where the upper bound converges to 0 as $T$ increases.
>
> Regarding the comparison of our work with reference Sfyraki & Wang (2025)[3], the following differences exist:
>
> -   **Assumptions on the optimization problem.** Sfyraki & Wang (2025) require the solution set to be compact, while our analysis does not rely on this compactness assumption.
> -   **Algorithmic form.** Sfyraki & Wang (2025) substantially modify Muon and use a linear convex-combination update, which is fundamentally different from the Muon-style update considered in our work.
> -   **Convergence criterion.** Sfyraki & Wang (2025) use the Frank–Wolfe gap $\mathcal{G}(x)$ as the convergence measure, whereas our results are based on $\frac{1}{T}\sum_{t\in[T]}\mathbb{E}||\nabla f(X_t)||_F$.
> -   **Step-size schedule.** Sfyraki & Wang (2025) adopt a constant step size, while we use decaying step sizes, which better reflect common practice in large-scale training.
> -   **Convergence results.** We additionally prove non-ergodic convergence, which is not covered in Sfyraki & Wang (2025).
>
> Thus, although both papers study Muon-related methods, the algorithmic form, theoretical setting, and convergence guarantees differ substantially and are not directly comparable.
>
> [2] Convergence Bound and Critical Batch Size of Muon Optimizer
>
> [3] Lions and Muons: Optimization via Stochastic Frank-Wolfe - Sfyraki and Wang
>
> ---

---

> ### Author Response · Authors · 2025-11-19
>
> **Q3.** The convergence of the variant MVR2 (with $\gamma=1$ and additional weight decay) in the non-convex setting was also studied in a previous work [3]. This work claims that their advantage lies in the use of a batch size of $O(1)$ , as opposed to [3]. However, [3] demonstrates an interaction complexity of  $O(T^{-1/3})$ using a large batch size only in the first iteration and for all subsequent iterations the batch size is set to $O(1)$. Therefore, the amortized average batch size is also . Furthermore, the complexity in [3] does not involve a logarithmic factor, as opposed to this work, which demonstrates a complexity of $O(\log(T)/T^{1/3})$. Additionally, Assumption 3.3 of this work is stronger than Assumption 2 in [3].
>
> **A3.** First, regarding the strength of assumptions, our Assumption 3.3 and Assumption 2 in [3] are essentially the same type of sample-wise L-smoothness assumption, both following the standard setting in STORM [1], and do not introduce any strictly stronger additional conditions.
>
> Moreover, as mentioned in our response to **W2**, our work and [3] differ substantially in the following aspects: assumptions on the optimization problem, the update rule of the algorithm ([3] adopts a stochastic Frank–Wolfe–type linear convex combination), the convergence criterion (Frank–Wolfe gap vs. gradient norm), the stepsize policy, and whether non-ergodic convergence is established. Therefore, even though both complexity bounds contain terms of order $T^{-1/3}$, the objects under analysis and the theoretical frameworks are quite different, making a direct one-to-one comparison difficult.
>
> [1] Cutkosky, Ashok, and Francesco Orabona. "Momentum-based variance reduction in non-convex sgd." *
>
> [3] Sfyraki, Maria-Eleni, and Jun-Kun Wang. "Lions and muons: Optimization via stochastic frank-wolfe."
>
> ---

---

> ### Author Response · Authors · 2025-11-19
>
> **Q4.** This work claims that the MVR1 variant is equivalent to Muon with Nesterov momentum for a specific choice of parameters (equation (4)). I do not see how this is possible, though I may be wrong.
>
> Can the authors elaborate on how equation (4) is obtained from Algorithm 1?
>
> **A4.**  We indeed omitted a scaling factor in the original manuscript. In the revision (lines 162-166, marked in blue), we more rigorously clarify the equivalence between Eq. (4) and Algorithm 1, with the derivation outlined as follows.
>
> 1.Obtaining a form equivalent to Nesterov momentum via rescaling.
>  Setting $\beta _ 1 = \mu$ and $\gamma = 1-\mu$, substituting into the MVR1 formula gives
> $$
> \mathbf{M} _ t=\mu\mathbf{M} _ {t-1}+(1-\mu)\nabla f(\mathbf{X} _ t;\xi _ t)+(1-\mu)\cdot \mu\cdot(\nabla f(\mathbf{X} _ t;\xi _ t)-\nabla f(\mathbf{X} _ {t-1};\xi _ {t-1})).
> $$
> Multiplying both sides by $1/(1-\mu)$ yields
> $$
> \frac{\mathbf{M} _ t}{1-\mu}=\frac{\mu}{1-\mu}\mathbf{M} _ {t-1}+\nabla f(\mathbf{X} _ t;\xi_t)+\mu\cdot(\nabla f(\mathbf{X} _ t;\xi_t)-\nabla f(\mathbf{X} _ {t-1};\xi _ {t-1})).
> $$
> Letting $\tilde{\mathbf{M}} _ t = \mathbf{M} _ t/(1-\mu)$, we obtain
> $$
> \tilde{\mathbf{M}} _ t = \mu \tilde{\mathbf{M}} _ {t-1}+\nabla f(\mathbf{X} _ t;\xi _ t)+\mu\cdot(\nabla f(\mathbf{X} _ t;\xi _ t)-\nabla f(\mathbf{X} _ {t-1};\xi _ {t-1})),
> $$
> which is consistent with the standard Nesterov momentum form.
>
> On the other hand, consider the following iterative system:
> $$
> \mathbf{C} _ t=\mu\mathbf{C} _ {t-1}+\nabla f(\mathrm{X} _ t;\xi _ t),
> \mathbf{M} _ t=\mu\mathbf{C} _ t+\nabla f(\mathbf{X} _ t;\xi _ t).
> $$
> Then
> $$
> \mathbf{M} _ {t-1}-\nabla f(\mathbf{X} _ {t-1};\xi _ {t-1})=\mu\mathbf{C} _ {t-1}.
> $$
> Substituting back gives
> $$
> \mathbf{M} _ t=\mu(\mathbf{M} _ {t-1}-\nabla f(\mathbf{X} _ {t-1};\xi _ {t-1})+\nabla f(\mathrm{X} _ t;\xi _ t))+\nabla f(\mathbf{X} _ t;\xi _ t)
> = \mu\mathbf{M} _ {t-1}+\nabla f(\mathbf{X} _ t;\xi _ t) + \mu(\nabla f(\mathrm{X} _ t;\xi _ t)-\nabla f(\mathbf{X} _ {t-1};\xi _ {t-1})),
> $$
> which is exactly an equivalent formulation of Nesterov momentum. Similar equivalence transformations have been discussed in the literature [1] (Page 8, Section 3, Eqs. (3.22)–(3.25)), and we follow this perspective in clarifying our wording in the revision.
>
> 2.Connection to standard Nesterov acceleration.
>  The standard Nesterov-accelerated gradient iteration can be written as
> $$
> \mathbf{G} _ t = \nabla f(\mathbf{X} _ t-\eta \cdot\mu \mathbf{M} _ {t-1}),
> \mathbf{M} _ t = \mu \mathbf{M} _ {t-1} + \mathbf{G} _ t,
> \mathbf{X} _ {t+1} = \mathbf{X} _ t-\eta \mathbf{M} _ t.
> $$
> Applying a first-order Taylor expansion to $\mathbf{G} _ t$ gives
> $$
> \mathbf{G} _ t = \nabla f(\mathbf{X} _ t-\eta \cdot\mu \mathbf{M} _ {t-1})
> \approx\nabla f(\mathbf{X} _  t) + \mu \nabla^2 f(\mathbf{X} _ t)(-\eta \mathbf{M} _ {t-1})
> =\nabla f(\mathbf{X} _ t) + \mu \nabla^2 f(\mathbf{X} _ t)(\mathbf{X} _ t-\mathbf{X} _ {t-1})
> \approx \nabla f(\mathbf{X} _ t)+ \mu(\nabla f(\mathbf{X} _ t)-\nabla f(\mathbf{X} _ {t-1})),
> $$
> which leads to
> $$
> \mathbf{g} _ t = \nabla f(\mathbf{X} _ t),
> \mathbf{M} _ t = \mu \mathbf{M} _ {t-1} + \mathbf{g} _ t + \mu (\mathbf{g} _ t-\mathbf{g} _ {t-1}),
> \mathbf{X} _ {t+1} = \mathbf{X} _ t-\eta \mathbf{M} _ t,
> $$
> matching the rescaled MVR1 momentum update derived above. Similar derivations are used in Adan (Xie et al., 2024) and related work.
>
> [1] MARS: Unleashing the Power of Variance Reduction for Training Large Models - Yuan et al.
>
> [2] Xie, Xingyu, et al. "Adan: Adaptive nesterov momentum algorithm for faster optimizing deep models." *IEEE Transactions on Pattern Analysis and Machine Intelligence* 2024
>
> ---
>
> **Q5.** Some parts of the non-ergodic section are a bit unclear. For example, $\Gamma_t$ is not specified in Remark 3.5.
>
> **A5.** We have now explicitly given the precise form of $\Gamma_t$ in the main text, and in Remark 3.5 (Remark 3.6 in the revised version) we have added references to the corresponding definitions and derivations in the appendix to avoid ambiguity.
>
> ---
>
> **Q6.** The introduction of an additional hyperparameter $\gamma$ makes tuning even more tedious in practice.
>
> **A6.** Note that when $\gamma = 0$, the momentum update reduces to the standard EMA form; when $\gamma \neq 0$, we use the variance-reduction correction term proposed in MARS. As explained on page 4 of MARS, Section “Why $\gamma_t$ improves convergence”, “dynamically tuning $\gamma_t$ improves the variance of the gradient estimator, leading to better convergence behavior.” Thus, our motivation for introducing $\gamma$ is precisely to control the variance of the gradient estimator and improve convergence. In the experiments, we indeed set $\gamma$ to a small value, both to be consistent with the MARS setup and to strike a balance between numerical stability and generalization performance, rather than to abandon the variance-reduction effect.
>
> [1]MARS: Unleashing the Power of Variance Reduction for Training Large Models - Yuan et al.
>
> ---

---

> ### Author Response · Authors · 2025-11-19
>
> **Q7.** It is not specified how the variance parameter  $\gamma$, momentum parameter $\beta$ and weight decay parameter were tuned in the experiments.
>
> **A7.** In the revised manuscript (lines 1646-1668, marked in blue), we have added the specific tuning strategies for $\gamma,\beta,\lambda$: we choose $\gamma \in \{0.01,0.025,0.05\}$, set the momentum coefficient of the Muon variants to $\beta = 0.95$, and use a weight decay coefficient $\lambda = 0.1$ for all optimizers
>
> ---
>
> **Q8.** One minor presentation issue is that standard Muon is referred to as Muon-MVR1 with $\gamma=0$, which is a bit misleading, since in that case no variance reduction is applied.
>
> **A8.** In the revised manuscript (line 124, marked in blue) , we refer to this class of methods collectively as “Muon-style” algorithms, and we have updated Algorithm 1 and the related text to clarify that $\gamma = 0$ corresponds to standard Muon, while the variance-reduction term is introduced only when $\gamma \neq 0$, in order to avoid ambiguity.
>
> ---
>
> **Q9.** I understand that is used to control the variance reduction term, as in [1]. However, this scheme is different from the originally proposed STORM scheme in [4]. I also see that in practice is set to a very small value, which effectively reduces the impact of the variance reduction. Can the authors explain the motivation for including $\gamma$ ?
>
> **A9.** As in [1], $\gamma$ is used to control the variance-reduction term:
>
> -   when $\gamma = 0$, the update reduces to the classical EMA form;
> -   when $\gamma \neq 0$, a MARS-like variance-reduction correction is introduced.
>
> As explained in MARS [1] (page 4, Section “Why $\gamma_t$ improves convergence”), properly chosen $\gamma_t$ reduces the variance of the gradient estimator and enhances convergence. In our experiments, we adopt a small but fixed $\gamma$ to balance theoretical motivation with practical training stability. The goal is to retain the beneficial variance-reduction effect, not to weaken its role.
>
> [1]MARS: Unleashing the Power of Variance Reduction for Training Large Models - Yuan et al.
>
> ---

---

### Official Review · Reviewer_XfDc · 2025-10-30

**Soundness:** 3
**Presentation:** 2
**Contribution:** 3
**Rating:** 6
**Confidence:** 4

**Summary:**

This paper analyzes the theoretical convergence of the Muon optimizer and introduces two momentum-based variance-reduced variants, Muon-MVR1 and Muon-MVR2. The authors show that Muon-MVR2 attains the optimal stochastic non-convex iteration complexity of $\tilde{\mathcal{O}}(T^{-1/3})$ and establish last-iterate convergence under the Polyak–Łojasiewicz condition. The analysis is elegant and unified: it extends the stochastic momentum framework to the orthogonalized Muon setting, with proofs that are logically consistent and mathematically transparent. Empirical results on CIFAR-10 and C4 further support the theoretical claims.

**Strengths:**

1. **Clear theoretical contribution.** The paper provides a rigorous convergence analysis for Muon-MVR, bridging the gap between Muon’s empirical success and previously incomplete theory. The Lyapunov-style recursion and variance-control arguments are clean and easy to follow.

2. **Well-structured and consistent analysis.** The logical progression from assumptions to lemmas and theorems is smooth, and the proofs are well-motivated and technically precise. The framework appears general enough to extend to other momentum–variance-reduction methods.

3. **Readable exposition.** Despite the mathematical density, the writing is accessible and the notation—aside from minor issues—remains consistent. The paper is pleasant to read, and the proofs are presented with notable clarity.

**Weaknesses:**

1. **Overstated claims in presentation.** The statement that the method “overcomes the bottleneck of traditional momentum methods, achieving the optimal iteration complexity” is exaggerated. Sfyraki & Wang (2025) already analyze a Muon+VR variant (under a stochastic Frank–Wolfe formulation) achieving the same $\tilde{\mathcal{O}}(T^{-1/3})$ rate. The novelty here lies primarily in the unconstrained SGD-style analysis, not in attaining a new theoretical complexity.

2. **Ambiguity in Table 1.** The superscripts $c$ and $d$ are undefined. Their meaning should be clearly explained in the caption or a footnote.

3. **Formula inconsistency (Eq. (4)).** As written,
   $$
   M_t = \mu M_{t-1} + \nabla f(X_t, \xi_t) + \mu \big(\nabla f(X_t, \xi_t) - \nabla f(X_{t-1}, \xi_{t-1})\big)
   $$
   is not equivalent to Eq. (2). It should instead be
   $$
   C_t = \mu C_{t-1} + (1-\mu)\nabla f(X_t,\xi_t), \quad
   M_t = \mu C_t + (1-\mu)\nabla f(X_t,\xi_t).
   $$

4. **Possible overclaim in Remark 3.2.** Although the analysis avoids a growing-batch assumption, the comparison with Sfyraki & Wang (2025) is not fully aligned, as their problem setting, objective, and convergence metric differ. Characterizing this as a “significant theoretical advantage” is therefore overstated—especially without empirical evidence that fixed-batch training achieves the claimed $\tilde{\mathcal{O}}(T^{-1/3})$ scaling.

5. **Experimental limitations.** The empirical section primarily plots standard training curves and does not directly test the predicted scaling laws. Moreover, there is no comparison against other Muon-type optimizers. As a result, the experiments feel empirical rather than diagnostic.

6. **Typographical and constant issues.**
   - (1) Page 15 (L799): should read $\epsilon = \frac{1-\beta_{t+1}}{\beta_{t+1}}$.
   - (2) Page 15 (L801), Eq. (6): the right-hand side contains an extra “$t$”.
   - (3) Page 22 (L1187): the second term’s constant should be $2\eta_{t+1}^2(4L^2 n + \sigma^2)$; assuming $\eta_t \le 2\eta_{t+1}$, we have $2\eta_t^2 L^2 n \le 8\eta_{t+1}^2 L^2 n$, which affects multiple subsequent constants.

**Questions:**

The main improvement of Muon lies in its orthogonalization step. Could the authors elaborate on why orthogonalization improves optimization dynamics over standard SGD? Is there a formal argument or empirical evidence that the equalization of singular values accelerates convergence?

---

> ### Author Response · Authors · 2025-11-19
>
> We thank the reviewers for their constructive comments. We have addressed these issues point-by-point below.
>
> ---
>
> **Q1.** Overstated claims in presentation. The statement that the method “overcomes the bottleneck of traditional momentum methods, achieving the optimal iteration complexity” is exaggerated. Sfyraki & Wang (2025) already analyze a Muon+VR variant (under a stochastic Frank–Wolfe formulation) achieving the same $\tilde{\mathcal{O}}(T^{-1/3})$ rate. The novelty here lies primarily in the unconstrained SGD-style analysis, not in attaining a new theoretical complexity.
>
> **A1.** In the revision (lines 59-61, marked in blue), we have changed the description to: in the unconstrained Muon-style setting, Muon-MVR2 achieves, for the first time, a convergence rate of $\tilde{\mathcal{O}}(T^{-1/3})$. The specific change appears in lines 58–62 of the manuscript.
>
> ---
>
> **Q2.** Ambiguity in Table 1. The superscripts $c$ and $d$ are undefined. Their meaning should be clearly explained in the caption or a footnote.
>
> **A2.**  As noted in lines 88–90, we have already defined $c$ and $d$ in the note to Table 1.
>
> ---
>
> **Q3.** Formula inconsistency (Eq. (4)). As written, $M_t=\mu M_{t-1}+\nabla f(X_t,\xi_t)+\mu\left(\nabla f(X_t,\xi_t)-\nabla f(X_{t-1},\xi_{t-1})\right)$，is not equivalent to Eq. (2). It should instead be
>
>
> **A3.** Thank the reviewer for pointing out the inconsistency in Eq. (4). You are correct that we indeed omitted a scaling factor in the original manuscript. In the revision (lines 162-166, marked in blue), we more rigorously clarify the equivalence between Eq. (4) and Algorithm 1, with the derivation outlined as follows.
>
> Setting $\beta _ 1 = \mu$ and $\gamma = 1-\mu$, substituting into the MVR1 formula gives
> $$
> \mathbf{M} _ t=\mu\mathbf{M} _ {t-1}+(1-\mu)\nabla f(\mathbf{X} _ t;\xi _ t)+(1-\mu)\cdot \mu\cdot(\nabla f(\mathbf{X} _ t;\xi _ t)-\nabla f(\mathbf{X} _ {t-1};\xi _ {t-1})).
> $$
> Multiplying both sides by $1/(1-\mu)$ yields
> $$
> \frac{\mathbf{M} _ t}{1-\mu}=\frac{\mu}{1-\mu}\mathbf{M} _ {t-1}+\nabla f(\mathbf{X} _ t;\xi_t)+\mu\cdot(\nabla f(\mathbf{X} _ t;\xi_t)-\nabla f(\mathbf{X} _ {t-1};\xi _ {t-1})).
> $$
> Letting $\tilde{\mathbf{M}} _ t = \mathbf{M} _ t/(1-\mu)$, we obtain
> $$
> \tilde{\mathbf{M}} _ t = \mu \tilde{\mathbf{M}} _ {t-1}+\nabla f(\mathbf{X} _ t;\xi _ t)+\mu\cdot(\nabla f(\mathbf{X} _ t;\xi _ t)-\nabla f(\mathbf{X} _ {t-1};\xi _ {t-1})),
> $$
> which is consistent with the standard Nesterov momentum form.
>
> On the other hand, consider the following iterative system:
> $$
> \mathbf{C} _ t=\mu\mathbf{C} _ {t-1}+\nabla f(\mathbf{X} _ t;\xi _ t),
> \mathbf{M} _ t=\mu\mathbf{C} _ t+\nabla f(\mathbf{X} _ t;\xi _ t).
> $$
> Then
> $$
> \mathbf{M} _ {t-1}-\nabla f(\mathbf{X} _ {t-1};\xi _ {t-1})=\mu\mathbf{C} _ {t-1}.
> $$
> Substituting back gives
> $$
> \mathbf{M} _ t=\mu(\mathbf{M} _ {t-1}-\nabla f(\mathbf{X} _ {t-1};\xi _ {t-1})+\nabla f(\mathbf{X} _ t;\xi _ t))+\nabla f(\mathbf{X} _ t;\xi _ t)
> = \mu\mathbf{M} _ {t-1}+\nabla f(\mathbf{X} _ t;\xi _ t) + \mu(\nabla f(\mathbf{X} _ t;\xi _ t)-\nabla f(\mathbf{X} _ {t-1};\xi _ {t-1})),
> $$
> which is exactly an equivalent formulation of Nesterov momentum. Similar equivalence transformations have been discussed in the literature [1] (Page 8, Section 3, Eqs. (3.22)–(3.25)), and we follow this perspective in clarifying our wording in the revision.
>
> [1] Yuan, Huizhuo, et al. "Mars: Unleashing the power of variance reduction for training large models, 2024."
>
> ---
>
> **Q4.** Possible overclaim in Remark 3.2. Although the analysis avoids a growing-batch assumption, the comparison with Sfyraki & Wang (2025) is not fully aligned, as their problem setting, objective, and convergence metric differ. Characterizing this as a “significant theoretical advantage” is therefore overstated—especially without empirical evidence that fixed-batch training achieves the claimed $\tilde{\mathcal{O}}(T^{-1/3})$ scaling.
>
> **A4.**  In the revision (lines 270-273, marked in blue), we have changed the wording to: our convergence complexity matches the current best-known results, and we provide the corresponding analysis in the setting of unconstrained Muon-style algorithms with momentum variance reduction.
>
> [1] Sfyraki, Maria-Eleni, and Jun-Kun Wang. "Lions and muons: Optimization via stochastic frank-wolfe." *arXiv preprint arXiv:2506.04192* (2025).
>
> ---

---

> ### Author Response · Authors · 2025-11-19
>
> **Q5.** Experimental limitations. The empirical section primarily plots standard training curves and does not directly test the predicted scaling laws. Moreover, there is no comparison against other Muon-type optimizers. As a result, the experiments feel empirical rather than diagnostic.
>
> **A5.** The main contribution of this paper is a systematic theoretical analysis of the convergence of Muon-type optimizers; the experiments are intended to use a limited compute budget to verify and illustrate the overall behavior predicted by the theory, rather than to build a comprehensive empirical benchmark.
>
> Regarding the point that we do not directly test scaling laws, a rigorous scaling study would require systematically sweeping over multiple model sizes, data scales, and training budgets, with careful hyperparameter tuning for each configuration. At the LLaMA/C4 scale, this workload exceeds the compute budget we can afford in the rebuttal stage. Under this constraint, we chose to prioritize (1) longer training runs (10B tokens) and (2) a more extensive hyperparameter search for the proposed methods.
>
> We clearly state the limitations of the current experiments in the revised manuscript (lines 528-536, marked in blue), and regard a systematic comparison with other Muon-type optimizers and a more complete empirical study of scaling laws as important directions for future work.
>
> ---
>
> **Q6.** Typographical and constant issues. (1) Page 15 (L799): should read
>
> (2) Page 15 (L801), Eq. (6): the right-hand side contains an extra “t”.
>
> (3) Page 22 (L1187): the second term’s constant should be $2\eta_{t+1}^2(4L^2n+\sigma^2)$; assuming , we have , which affects multiple subsequent constants.
>
> **A6.** Thank the reviewer for the careful checking. We have corrected the symbol and constant errors mentioned in the manuscript one by one and have correspondingly verified the affected constants in the subsequent derivations.
>
> ---
>
> **Q7.** The main improvement of Muon lies in its orthogonalization step. Could the authors elaborate on why orthogonalization improves optimization dynamics over standard SGD? Is there a formal argument or empirical evidence that the equalization of singular values accelerates convergence?
>
> **A7.** Theoretically, [1,2] interpret simplified variants of Muon, Shampoo, and related methods as instances of steepest descent under a spectral-norm constraint. This is arguably well aligned with the nature of neural network matrix-parameter optimization: one performs gradient descent in a geometry induced by a matrix/spectral norm, which naturally tends to “equalize” the singular values of the parameter matrices.
>
> Empirically, SGD performs much worse than Adam for training Transformers, as has been systematically documented in a number of works [3]. Some studies analyze this phenomenon from the viewpoint of the Hessian spectrum of Transformers [3,4], while others attribute it to heavy-tailed data and class imbalance [5].
>
> Our empirical claim is that plain SGD, lacking adaptive step sizes, struggles to timely adjust step lengths across different directions under ill-conditioned Hessians or heavy-tailed, imbalanced data, which leads to slow or even unstable convergence. The orthogonalization step in Muon inherits the above “spectral/norm control” idea: by approximately orthogonalizing the momentum/gradient matrix, it makes its singular values more balanced, improves the effective condition number, and partially mitigates the optimization difficulties induced by pathological Hessians and heavy-tailed, imbalanced data. Conceptually, this is consistent with the theoretical and empirical findings in norm-SGD, Clip-SGD, and singular-value–reweighting methods such as Shampoo, which all observe that “controlling the gradient norm or singular values accelerates convergence” [6,7,8].
>
> [1] Jeremy Bernstein and Laker Newhouse.  "Old Optimizer, New Norm: An Anthology ".
>
> [2] Thomas Pethick, et al.  "Training Deep Learning Models with Norm-Constrained LMOs ".
>
> [3] Yushun Zhang, et al.  "Why Transformers Need Adam: A Hessian Perspective ".
>
> [4] Yushun Zhang, et al.  "Adam-mini: Use Fewer Learning Rates To Gain More ".
>
> [5] Frederik Kunstner, et al.  "Heavy-Tailed Class Imbalance and Why Adam Outperforms Gradient Descent on Language Models ".
>
> [6] Alexander Cutkosky.  "Momentum Improves Normalized SGD ".
>
> [7] Savelii Chezhegov, et al.  "Convergence of Clipped-SGD for Convex (L_0, L_1)-Smooth Optimization with Heavy-Tailed Noise".
>
> [8] Vineet Gupta, Tomer Koren, Yoram Singer.  "Shampoo: Preconditioned Stochastic Tensor Optimization".
>
>
> ---

---

### Official Review · Reviewer_twoR · 2025-10-30

**Soundness:** 3
**Presentation:** 3
**Contribution:** 2
**Rating:** 4
**Confidence:** 4

**Summary:**

This work integrates Momentum Variance Reduction (MVR)—including a MARS‑style $\gamma$ factor—into Muon. It analyzes three variants (standard Muon/MVR1 with $\gamma=0$, MVR1 with $\gamma_t=t^{-1/2}$, and MVR2 with two gradients on the same mini‑batch), proving ergodic rates $\tilde{\mathcal O}(T^{-1/4})$ for MVR1 and $\tilde{\mathcal O}(T^{-1/3})$ for MVR2, plus PŁ last‑iterate rates $\tilde{\mathcal O}(T^{-1/2})$ (MVR1) and $\tilde{\mathcal O}(T^{-2/3})$ (MVR2). Experiments on CIFAR‑10/ResNet‑18 and LLaMA2‑130M/C4 are provided.

**Strengths:**

- The core idea is reasonable and timely: MARS-style variance reduction has worked well in practice, and combining MVR with Muon yields provable benefits, making the approach promising for practical impact.
- The analysis of a more practical, one-batch MVR variant appears novel at the algorithmic level; however, I am uncertain about the theoretical novelty/difficulty of this result.
- The main results are stated cleanly and appear correct.
- The algorithmic setup is presented clearly: MVR1 vs. MVR2, schedules, and update rules.
- The empirical section covers reasonable benchmarks (CIFAR-10/ResNet-18, LLaMA2-130M on C4) and broadly supports the claimed advantages, especially for MVR2.

**Weaknesses:**

- Despite its practical motivation, the paper’s theoretical analysis appears to offer limited novelty. The proof technique closely mirrors that of normalized SGD with momentum (Cutkosky & Mehta, 2020)—which is, surprisingly, not cited—and is similar to prior analyses of Muon’s convergence. Moreover, MVR in this context has already been discussed (e.g., the OptML notes: https://optmlclass.github.io/notes/optforml_notes.pdf), and parameter-agnostic results under comparable assumptions (e.g., Yang et al., 2023) are not addressed. As a result, beyond analyzing a more practical $\gamma$-factor variant, the contribution seems incremental: with a decreasing schedule, $\gamma_t$, the analysis does not improve the bound over $\gamma=0$ (it essentially recovers the SGD rate). The paper should add a clear, dedicated discussion of technical novelty, explicitly situating its results relative to these works and clarifying what is genuinely new in the Muon-specific setting.

- The $\gamma$‑factor with a decreasing schedule ($\gamma_t=t^{-1/2}$) recovers the same $\tilde{\mathcal O}(T^{-1/4})$ rate as $\gamma=0$, so the analysis does not show a theoretical gain from the MARS‑like correction.
- Related‑work positioning: compares against relatively weak baselines and omits important works such as Kovalev (2025); needs a more detailed comparison to Sfyraki & Wang (2025).
- Optimality beyond the $T$‑exponent is unclear: the dependence on $L,\sigma,n$ (especially for Theorem 3.2) is not discussed relative to known parameter‑agnostic results.
- Missing important baseline: MARS‑AdamW should be included.
- Missing/unclear experimental details: whether Muon is combined with AdamW for some parameter groups, and the separate learning rates; absence of ablations on $\gamma$, weight decay, and batch size; on C4, a single fixed LR is used for all optimizers, which is hard to justify.
- “Nesterov‑accelerated” wording for Option MVR1 is incorrect for the presented update.

---
- Cutkosky, A., & Mehta, H. “Momentum improves normalized SGD.” ICML, 2020.
- Yang, Junchi, et al. “Two sides of one coin: the limits of untuned SGD and the power of adaptive methods.” NeurIPS 36, 2023.
- Kovalev, Dmitry. “Understanding gradient orthogonalization for deep learning via non-Euclidean trust‑region optimization.” arXiv:2503.12645, 2025.
- OptML notes on MVR in this context: https://optmlclass.github.io/notes/optforml_notes.pdf￼

**Questions:**

- Technical novelty: how do the proofs differ materially from normalized‑SGD‑with‑momentum (Cutkosky & Mehta, 2020) and prior Muon analyses; why are these not cited? Please also discuss connections to the OptML MVR notes and parameter‑agnostic results (Yang et al., 2023).
- Positioning vs. prior work: add a careful comparison to Sfyraki & Wang (2025) and include Kovalev (2025); specify which assumptions/estimators.
-  Baselines: add MARS‑AdamW on CIFAR‑10 and C4.
- Empirical details: clarify if Muon is used only for matrix‑shaped parameters while AdamW is used elsewhere; list parameter groups and per‑group learning rates; report weight decay values.
- LLaMA2/C4 setup: justify using a single LR for all optimizers or include at least a small per‑optimizer LR sweep.
- Constants: discuss the dependence on $L,\sigma,n$ in Theorems 3.1–3.2 and whether any factors are tight or can be improved.

---

> ### Author Response · Authors · 2025-11-19
>
> We appreciate the reviewers' feedback and provide a detailed, point-by-point response below.
>
> ---
>
> **Q1.** Despite its practical motivation, the paper’s theoretical analysis appears to offer limited novelty. The proof technique closely mirrors that of normalized SGD with momentum (Cutkosky & Mehta, 2020)—which is, surprisingly, not cited—and is similar to prior analyses of Muon’s convergence. Moreover, MVR in this context has already been discussed (e.g., the OptML notes: https://optmlclass.github.io/notes/optforml_notes.pdf), and parameter-agnostic results under comparable assumptions (e.g., Yang et al., 2023) are not addressed. As a result, beyond analyzing a more practical -factor variant, the contribution seems incremental: with a decreasing schedule, , the analysis does not improve the bound over (it essentially recovers the SGD rate). The paper should add a clear, dedicated discussion of technical novelty, explicitly situating its results relative to these works and clarifying what is genuinely new in the Muon-specific setting.
>
> Technical novelty: how do the proofs differ materially from normalized‑SGD‑with‑momentum (Cutkosky & Mehta, 2020) and prior Muon analyses; why are these not cited? Please also discuss connections to the OptML MVR notes and parameter‑agnostic results (Yang et al., 2023).
>
> **A1.**
>
> 1.In the related work section (lines 483–485, marked in blue), we systematically add references on **normalization-based gradient-scaling methods**, including normalized SGD with momentum (Cutkosky & Mehta, 2020), clipped SGDM, Lion, etc.
>
> 2.**Substantive differences from normalized SGD with momentum (Cutkosky & Mehta, 2020):**
>
> -   Step-size schedules. NormSGD uses a step size of order $O(T^{-1/2})$ (or constant steps), whereas MVR1 and MVR2 use decaying steps $\eta_t = O(t^{-3/4})$ and $\eta_t = O(t^{-2/3})$, leading to different convergence structures and complexity analyses.
> -   PL convergence. Under the PL condition, we establish non-ergodic convergence for Muon/MVR1/MVR2, which is not covered in NormSGDM.
> -   Complexity guarantees. For Muon-MVR2, we prove an ergodic **optimal** iteration complexity of $\tilde{O}(\varepsilon^{-3})$, improving on the $\tilde{O}(\varepsilon^{-4})$ complexity of vanilla Muon/SGD-type methods.
>
> 3.**Algorithmic level.** Cutkosky & Mehta analyze scalar/vector NormSGDM in Euclidean space, whereas we study a Muon matrix optimizer with an orthogonalization step, which introduces a matrix-plus-projection structure and MVR-specific errors that require re-establishing the convergence inequalities. Consequently, the NormSGDM proofs cannot be reused; our analysis instead relies on new arguments that critically combine **Lemma A.3** and **Lemma C.1**.
>
> 4.**Relation to the MVR analysis in the OptML notes.** The MVR in the OptML notes is developed for standard SGD/Adam-type optimizers in Euclidean space and does not involve the orthogonalization structure of Muon. Our main contributions are:
> -   Combining MVR with the orthogonalization step in Muon to propose Muon-MVR1 and Muon-MVR2;
> -   Proving that a **constant mini-batch size** suffices to achieve the **optimal** rate $\tilde{\mathcal{O}}(T^{-1/3})$, matching STORM/SPIDER. Prior Muon variants typically require batch sizes that depend on the target accuracy or iteration index, or yield weaker guarantees. To our knowledge, this is the first work to obtain **optimal complexity** for an unconstrained Muon-style method through a practical “two-batch’’ MVR scheme.
>
> 5.**Novelty of the iterative structure in our PL non-ergodic analysis.** Under the PL inequality, we derive a new recursion: $\Delta_{t+1} \le \Delta_t - \frac{\sqrt{2\mu}}{t^p} \sqrt{\Delta_t} + \Gamma_t$, whereas prior analyses  typically rely on the linear form $\Delta_{t+1} \le (1-\rho)\Delta_t + \Gamma_t$. The additional square-root term $\sqrt{\Delta_t}$ term fundamentally changes the proof structure and technical details, so existing PL-type analyses cannot be applied.
>
> 6.**Relation to parameter-free results such as Yang et al. (2023).** Our algorithm does **not** require Lipschitz constants or other problem-dependent parameters. Instead, we focus on resolving the **complexity of unconstrained Muon-style matrix optimization** and achieving iteration bounds that match known lower limits. In contrast, Yang et al. (2023) study vector-parameterized adaptive SGD methods.
>
> [1] Cutkosky, A., & Mehta, H. “Momentum improves normalized SGD.” ICML, 2020.
>
> [2] OptML notes: https://optmlclass.github.io/notes/optforml_notes.pdf)
>
> [3] Yang, Junchi, et al. “Two sides of one coin: the limits of untuned SGD and the power of adaptive methods.”
>
> ---

---

> ### Author Response · Authors · 2025-11-19
>
> **Q2.** The $\gamma$‑factor with a decreasing schedule ($\gamma_t=t^{-1/2}$) recovers the same $\tilde{\mathcal{O}}(T^{-1/4})$ rate as , so the analysis does not show a theoretical gain from the MARS‑like correction.
>
> **A2.** We clarify the distinction as follows:
>
> 1. **On existing MARS theory.** Current theoretical results for MARS apply only to the *exact* variance-reduced scheme (MARS-AdamW), for which a rate of $\tilde{\mathcal{O}}(T^{-1/3})$ is known. In contrast, the *inexact* variant widely used in practice (MARS-AdamW-approx) lacks any formal convergence analysis to the best of our knowledge.
>
> 2. **Our contribution: the first theory for inexact VR in Muon-MVR1.** Under standard assumptions, we provide the first convergence analysis for **inexact variance reduction** in a Muon-style algorithm. We show that Muon-MVR1 achieves a rate of $\tilde{\mathcal{O}}(T^{-1/4})$, thereby formally characterizing the gap between the theoretically ideal MVR2 and the practically used inexact MVR1. This is, to our knowledge, the first result establishing such a rate in an inexact MVR setting.
>
> 3.  **Structural insight: relation to Nesterov-type acceleration.** Our analysis shows that the inexact MVR term is uniformly bounded, closely mirrors the structure of the exact MVR2 correction, and resembles Nesterov-style extrapolation—consistent with observations in recent work such as Adan (Xie et al., 2024). Under standard assumptions, our $\tilde{\mathcal{O}}(T^{-1/4})$ bound matches the best rates currently known for comparable inexact schemes.
>
> 4. **On potential sharper rates.**  We agree that the $\tilde{\mathcal{O}}(T^{-1/4})$ rate may not be tight. Achieving sharper guarantees (e.g., approaching $\tilde{\mathcal{O}}(T^{-1/3})$) typically requires stronger sample-wise smoothness or Hessian-regularity assumptions—conditions that are hard to leverage for inexact MVR1. Strengthening these assumptions (e.g., via **Hessian smoothness**, as in [1]) may allow future work to improve the rate further, and we will highlight this as a future direction.
>
> [1] Xie, Xingyu, et al. "Adan: Adaptive nesterov momentum algorithm for faster optimizing deep models." *IEEE Transactions on Pattern Analysis and Machine Intelligence* 46.12 (2024): 9508-9520.
>
> ---

---

> ### Author Response · Authors · 2025-11-19
>
> **Q3.** Related‑work positioning: compares against relatively weak baselines and omits important works such as Kovalev (2025); needs a more detailed comparison to Sfyraki & Wang (2025).
>
> Positioning vs. prior work: add a careful comparison to Sfyraki & Wang (2025) and include Kovalev (2025); specify which assumptions/estimators.
>
> **A3.**  We have added a new section titled 'Clarification of Theoretical Novelty' in the revised PDF (lines 342–362, marked in blue). Here we clarify how our work differs from theirs.
>
> **1.Comparison with Sfyraki & Wang (2025)**
>
> The two works study related ideas but under **substantially different assumptions, update rules, and convergence criteria**, making the results non-comparable in a direct sense.
>
> * **Assumptions.** Sfyraki & Wang (2025) require the solution set to be compact; our analysis does **not** impose this compactness assumption.
>
> * **Algorithmic form.** Their method uses a **linear convex-combination update**, which is fundamentally different from the Muon-style update we analyze.
>
> * **Convergence measure.** They adopt the Frank–Wolfe gap $\mathcal{G}(x)$; we work with the standard gradient-norm criterion $\frac{1}{T}\sum_{t=1}^T\mathbb{E}|\nabla f(X_t)|_F$.
>
> * **Stepsize policy.** They use **constant** stepsizes; our analysis focuses on **decaying** stepsizes, which more closely match modern large-scale training practice.
>
> * **Form of results.** Our paper additionally proves **non-ergodic** convergence guarantees, which are not covered in Sfyraki & Wang (2025).
>
> In summary, although both papers study Muon-related methods, the algorithmic form, theoretical setting, and convergence guarantees are fundamentally different.
>
> ---
>
> **2. Comparison with Kovalev (2025)**
>
> Kovalev (2025) provides a broad **non-Euclidean trust-region** framework, whereas our analysis focuses specifically on **Muon, MVR1, and MVR2** and yields sharper results in this context.
>
> * **General framework vs. specialized analysis.** Muon appears only as a special case in Kovalev’s unified framework. In contrast, we develop a **refined, variance-reduced, and non-ergodic** analysis tailored specifically to Muon-style methods.
>
> * **Complexity.** Kovalev obtains a standard SGD-type $\mathcal{O}(1/\varepsilon^4)$ complexity for both the unified framework and vanilla Muon. By exploiting the specific structure of Muon and incorporating variance-reduction, we improve this to $\mathcal{O}(1/\varepsilon^3)$ under the same nonconvex stochastic assumptions. These improvements **cannot** be derived from Kovalev’s general framework.
>
> * **Stepsizes.** Kovalev analyzes **constant** stepsizes, whereas we study **decaying** stepsizes for MVR1/MVR2—again more aligned with practical training.
>
> * **Additional guarantees.** We further establish **non-ergodic PL-style convergence** for Muon/MVR1/MVR2, which is not addressed in Kovalev (2025).
>
> [1] Sfyraki, Maria-Eleni, and Jun-Kun Wang. "Lions and muons: Optimization via stochastic frank-wolfe."
>
> [2] Kovalev, Dmitry. “Understanding gradient orthogonalization for deep learning via non-Euclidean trust‑region optimization.” arXiv:2503.12645, 2025.
>
> ---
>
> **Q4.** Optimality beyond the ‑exponent is unclear: the dependence on $L,\sigma,n$(especially for Theorem 3.2) is not discussed relative to known parameter‑agnostic results.
>
> Constants: discuss the dependence on in Theorems 3.1–3.2 and whether any factors are tight or can be improved.
>
> **A4.** In the revision (lines 284–292, marked in blue), we have added the following clarifications:
>
> 1. **New Remark 3.4 summarizing parameter dependence.** Immediately after Theorem 3.2, we introduce Remark 3.4, which explicitly states how the convergence bounds depend on $L$, $\sigma$, and $n$.
>
> 2. **Parameter dependence for MVR1.** For MVR1, the upper bound depends linearly on the noise variance $\sigma^2$ and on the dimension $n$ with coefficients controlled by $L$; there is no higher-order polynomial amplification.
>
>
> 3. **Parameter dependence for MVR2.** For MVR2, the dependence on $\sigma,\sigma^2$ is also linear in the stochastic and constant terms. The dependence on $n$ appears as $\sqrt{n}$ from the term with $L\sqrt{n}$ and as $n$ through $G$, which contains an $Ln$ term.
>
> 4.  **Tightness and potential improvements.** We expect our dependence on $L,\sigma,n$ to be of the same order as related nonconvex MVR methods, though some constants (especially those involving $\sigma$) may be improvable. For example, STORM-style [1] adaptive variance-reduction strategies could further weaken the dependence on $\sigma$; since this would require more complex updates and analysis, we will mention it as a direction for future work.
>
> [1]Cutkosky, Ashok, and Francesco Orabona. "Momentum-based variance reduction in non-convex sgd."
>
> ---

---

> ### Author Response · Authors · 2025-11-19
>
> **Q5.** Missing important baseline: MARS‑AdamW should be included.
>
> Baselines: add MARS‑AdamW on CIFAR‑10 and C4.
>
> **A5.** We have added MARS-AdamW comparison results in the LLaMA2-130M / C4 experiments and explicitly highlighted this baseline in the experimental section, so as to more fairly assess the performance differences between Muon-MVR1/MVR2 and existing MVR-type optimizers.
>
> ---
>
> **Q6.** Missing/unclear experimental details: whether Muon is combined with AdamW for some parameter groups, and the separate learning rates; absence of ablations on , weight decay, and batch size; on C4, a single fixed LR is used for all optimizers, which is hard to justify.
>
> Empirical details: clarify if Muon is used only for matrix‑shaped parameters while AdamW is used elsewhere; list parameter groups and per‑group learning rates; report weight decay values.
>
> **A6.** In the revision (lines 1646-1668, marked in blue), we provide a more detailed description of the experiments and add corresponding hyperparameter searches, as follows:
>
> 1.  **Parameter split between Muon and AdamW.** For 2D matrix-shaped parameters we use a Muon-type optimizer; for 1D vector parameters and the lm_head, we use AdamW. This split follows common practice in current Muon implementations.
>
> 2.  **Parameter groups and learning rates.** All parameter groups share a single global learning rate in implementation (the per-group learning rates are listed in the tables for reproducibility). For parameters optimized by Muon, we use $0.2\cdot \sqrt{\max(n,m)}$, where $n,m$ are the matrix dimensions, which has been shown to be an effective practical rule in Liu et al. (2025). We add to the appendix the detailed parameter-group partition and the exact learning-rate and weight-decay values for each group.
>
> 3.  **Learning-rate and $\gamma$ search on C4.** In response to W6’s concern that “all optimizers use a single learning rate on C4,” we now perform a more systematic hyperparameter search on the C4 task. We search the learning rate for all optimizers and conduct a grid search over $\gamma\in{0.01,0.025,0.05}$. The corresponding results and best configurations are reported in both the main text and the appendix.
>
> [1] Liu, Jingyuan, et al. "Muon is scalable for LLM training." *arXiv preprint arXiv:2502.16982* (2025).
>
> ---

---

> ### Author Response · Authors · 2025-11-19
>
> **Q7.**“Nesterov‑accelerated” wording for Option MVR1 is incorrect for the presented update.
>
> **A7.** Our intention was to emphasize that, after a suitable reparameterization, the momentum update in MVR1 is close in form to Nesterov momentum. To avoid misunderstanding, we have tightened and clarified the wording in the revision (lines 162-166 , marked in blue) and added a more rigorous derivation, as follows:
>
> 1.**Obtaining a form equivalent to Nesterov momentum via rescaling.**
>  Setting $\beta _ 1 = \mu$ and $\gamma = 1-\mu$, substituting into the MVR1 formula gives
> $$
> \mathbf{M} _ t=\mu\mathbf{M} _ {t-1}+(1-\mu)\nabla f(\mathbf{X} _ t;\xi _ t)+(1-\mu)\cdot \mu\cdot(\nabla f(\mathbf{X} _ t;\xi _ t)-\nabla f(\mathbf{X} _ {t-1};\xi _ {t-1})).
> $$
> Multiplying both sides by $1/(1-\mu)$ yields
> $$
> \frac{\mathbf{M} _ t}{1-\mu}=\frac{\mu}{1-\mu}\mathbf{M} _ {t-1}+\nabla f(\mathbf{X} _ t;\xi_t)+\mu\cdot(\nabla f(\mathbf{X} _ t;\xi_t)-\nabla f(\mathbf{X} _ {t-1};\xi _ {t-1})).
> $$
> Letting $\tilde{\mathbf{M}} _ t = \mathbf{M} _ t/(1-\mu)$, we obtain
> $$
> \tilde{\mathbf{M}} _ t = \mu \tilde{\mathbf{M}} _ {t-1}+\nabla f(\mathbf{X} _ t;\xi _ t)+\mu\cdot(\nabla f(\mathbf{X} _ t;\xi _ t)-\nabla f(\mathbf{X} _ {t-1};\xi _ {t-1})),
> $$
> which is consistent with the standard Nesterov momentum form.
>
> On the other hand, consider the following iterative system:
> $$
> \mathbf{C} _ t=\mu\mathbf{C} _ {t-1}+\nabla f(\mathrm{X} _ t;\xi _ t),
> \mathbf{M} _ t=\mu\mathbf{C} _ t+\nabla f(\mathbf{X} _ t;\xi _ t).
> $$
> Then
> $$
> \mathbf{M} _ {t-1}-\nabla f(\mathbf{X} _ {t-1};\xi _ {t-1})=\mu\mathbf{C} _ {t-1}.
> $$
> Substituting back gives
> $$
> \mathbf{M} _ t=\mu(\mathbf{M} _ {t-1}-\nabla f(\mathbf{X} _ {t-1};\xi _ {t-1})+\nabla f(\mathrm{X} _ t;\xi _ t))+\nabla f(\mathbf{X} _ t;\xi _ t)
> = \mu\mathbf{M} _ {t-1}+\nabla f(\mathbf{X} _ t;\xi _ t) + \mu(\nabla f(\mathrm{X} _ t;\xi _ t)-\nabla f(\mathbf{X} _ {t-1};\xi _ {t-1})),
> $$
> which is exactly an equivalent formulation of Nesterov momentum. Similar equivalence transformations have been discussed in the literature [1] (Page 8, Section 3, Eqs. (3.22)–(3.25)), and we follow this perspective in clarifying our wording in the revision.
>
> 2.**Connection to standard Nesterov acceleration.**
>  The standard Nesterov-accelerated gradient iteration can be written as
> $$
> \mathbf{G} _ t = \nabla f(\mathbf{X} _ t-\eta \cdot\mu \mathbf{M} _ {t-1}),
> \mathbf{M} _ t = \mu \mathbf{M} _ {t-1} + \mathbf{G} _ t,
> \mathbf{X} _ {t+1} = \mathbf{X} _ t-\eta \mathbf{M} _ t.
> $$
> Applying a first-order Taylor expansion to $\mathbf{G} _ t$ gives
> $$
> \mathbf{G} _ t = \nabla f(\mathbf{X} _ t-\eta \cdot\mu \mathbf{M} _ {t-1})
> \approx\nabla f(\mathbf{X} _  t) + \mu \nabla^2 f(\mathbf{X} _ t)(-\eta \mathbf{M} _ {t-1})
> =\nabla f(\mathbf{X} _ t) + \mu \nabla^2 f(\mathbf{X} _ t)(\mathbf{X} _ t-\mathbf{X} _ {t-1})
> \approx \nabla f(\mathbf{X} _ t)+ \mu(\nabla f(\mathbf{X} _ t)-\nabla f(\mathbf{X} _ {t-1})),
> $$
> which leads to
> $$
> \mathbf{g} _ t = \nabla f(\mathbf{X} _ t),
> \mathbf{M} _ t = \mu \mathbf{M} _ {t-1} + \mathbf{g} _ t + \mu (\mathbf{g} _ t-\mathbf{g} _ {t-1}),
> \mathbf{X} _ {t+1} = \mathbf{X} _ t-\eta \mathbf{M} _ t,
> $$
> matching the rescaled MVR1 momentum update derived above. Similar derivations are used in Adan (Xie et al., 2024) and related work.
>
> [1] MARS: Unleashing the Power of Variance Reduction for Training Large Models - Yuan et al.
>
> [2] Xie, Xingyu, et al. "Adan: Adaptive nesterov momentum algorithm for faster optimizing deep models." *IEEE Transactions on Pattern Analysis and Machine Intelligence* 46.12 (2024): 9508-9520.
>
> ---
>
> **Q8.** LLaMA2/C4 setup: justify using a single LR for all optimizers or include at least a small per‑optimizer LR sweep.
>
> **A8.**  In the revision (lines 377-388 ,marked in blue), we conduct a simple but reasonably broad per-optimizer learning-rate search: in the LLaMA2-130M / C4 (12B tokens) experiments, we grid-search each optimizer over $\\{3\mathrm{e}{-4}, 5\mathrm{e}{-4}, 8\mathrm{e}{-4}, 1\mathrm{e}{-3}, 2\mathrm{e}{-3}, 4\mathrm{e}{-3}, 6\mathrm{e}{-3}, 8\mathrm{e}{-3}\\}$ and report its best result. This mitigates the potential bias of using a single learning rate and makes the comparison across optimizers more fair.
>
> ---

---

### Official Review · Reviewer_RVgi · 2025-10-31

**Soundness:** 4
**Presentation:** 3
**Contribution:** 3
**Rating:** 6
**Confidence:** 3

**Summary:**

This paper aims to establish a rigorous theoretical foundation for the Muon optimizer, which has recently gained attention for its remarkable empirical performance. The authors point out the limitations of existing Muon analyses and propose two variants to address these issues.

**Strengths:**

The primary strength is the first proof that Muon-MVR2 achieves optimal iteration complexity in the stochastic non-convex function. The paper doesn’t stop at presenting theoretical results. It validates the algorithm through experiments on CIFAR-10 and C4 datasets.

**Weaknesses:**

While it's commendable that this study used a large dataset like C4, the testing was conducted on only one model, LLaMA2-130M. Although the paper's main contribution is theoretical, the lack of evaluation on more than one model and more than one dataset remains its biggest drawback.

**Questions:**

1.	Why wasn't the wall-clock time shown in Figure 2? While O-notation is important, what is practically significant is essentially the wall-clock time.

2.	Muon is primarily known as an optimizer frequently used for LLMs. Therefore, I question the choice of using an older model from 2016 (ResNet-18) specifically for image classification experiments. If the goal was to demonstrate its effectiveness in the vision domain as well, why weren't large-scale datasets like ImageNet/MS COCO and more recent vision models such as CLIP or DINO employed? The chosen combination of ResNet18 & CIFAR-10 might inadvertently lead readers to suspect that the method failed on more modern and larger-scale vision tasks, and thus only results from this simpler setup were presented.

---

> ### Author Response · Authors · 2025-11-19
>
> We sincerely appreciate the reviewer’s thoughtful comments and suggestions.
>
> ---
>
> **Q1.** While it's commendable that this study used a large dataset like C4, the testing was conducted on only one model, LLaMA2-130M. Although the paper's main contribution is theoretical, the lack of evaluation on more than one model and more than one dataset remains its biggest drawback.
>
> **A1.**  We understand and appreciate the reviewer’s concern.
>
> 1. The primary contribution of this paper is theoretical: we analyze the iteration complexities of two Muon variants—**Muon-MVR1** and **Muon-MVR2**—and show that neither universally dominates the other.
>
> 2. We substantially expanded our experiments in the revision, training AdamW, Muon, MARS-AdamW, Muon-MVR1, and Muon-MVR2 on LLaMA2-130M/C4 for **12B tokens** ((\approx 4\times) Chinchilla-optimal) with systematic tuning, and reporting loss vs. wall-clock time under fixed compute budgets.
>
> 3. We also clarify when each variant is preferable: **Muon-MVR2** is recommended for very high accuracy or when the extra batch cost is negligible, whereas **Muon-MVR1** is more appropriate in standard or compute-constrained settings.
>
>
> ---
>
> **Q2.** Why wasn't the wall-clock time shown in Figure 2? While O-notation is important, what is practically significant is essentially the wall-clock time.
>
> **A2.** In the revision, we added wall-clock time statistics for the LLAMA2-130M experiments on 4× Chinchilla data and report the corresponding comparison in Figure 2 C.
>
> ---
>
> **Q3.** Muon is primarily known as an optimizer frequently used for LLMs. Therefore, I question the choice of using an older model from 2016 (ResNet-18) specifically for image classification experiments. If the goal was to demonstrate its effectiveness in the vision domain as well, why weren't large-scale datasets like ImageNet/MS COCO and more recent vision models such as CLIP or DINO employed? The chosen combination of ResNet18 & CIFAR-10 might inadvertently lead readers to suspect that the method failed on more modern and larger-scale vision tasks, and thus only results from this simpler setup were presented.
>
> **A3.**
>
> Our goal was to study the optimization behavior of Muon-MVR2 in a controlled vision setting, using ResNet-18 on CIFAR-10 as a baseline.
>
> We recognize that Muon-MVR2’s main practical drawback is its two-gradient cost per iteration, and we do not claim it is efficient for large-scale vision tasks; this limitation is stated explicitly in Remark 4.1.
>
> The two variants serve different purposes: Muon-MVR2 is suitable when very high accuracy is required and the extra batch cost is negligible, whereas Muon-MVR1 is preferable otherwise.
>
> Extending the experiments to larger datasets (e.g., ImageNet, MS COCO) and modern vision models (e.g., CLIP, DINO) is an important direction, and this has been added to the “Limitations and Future Work” section (lines 528–536).
>
> ---

---

### Official Review · Reviewer_Xhb7 · 2025-10-31

**Soundness:** 3
**Presentation:** 2
**Contribution:** 3
**Rating:** 4
**Confidence:** 3

**Summary:**

This paper presents analysis of two momentum-based variance-reduced Muon-MVR1 and Muon-MVR2 algorithms, which achieve sublinear convergence rates. In addition, the paper also provides convergence analysis details for the standard Muon and Muon-MVR1 methods. Muon-MVR2 variant offers some additional analysis on achieving optimal iteration complexity of $O(T^{-1/3})$ for non-convex optimization. Primary comparisons are between Muon-MVR2 versus the theoretical lower bound. The paper presents some small-scale experiments (such as with CIFAR-10 and LLaMA2), and practical results imply that those looking for the best theoretical convergence rate should use the Muon-MVR2 variant.

**Strengths:**

- The main contribution of this paper is it's rigorous theoretical proof of optimality for the Muon-class optimizer. There has been some recent interest in understanding the performance of Muon [1], and aiming to close the gap between Muon's practical usability versus its theoretical understanding is valuable.

- The proposed Muon-MVR2 achieves its optimal convergence using a constant batch size $O(1)$. This is a significant improvement over prior work [2] that needed large initial batch sizes. The last-iterate non-ergodic convergence analysis is also rigorous, showing $O(T^{-2/3})$.

- There are experiments in both vision and language domains, although the main contribution of the paper remains its theoretical clarification of the Muon framework. There is a clearly expressed tradeoff between the "cheaper" Muon-MVR1 variant and the theoretically backed MVR2 variant.



[1] Shen, Wei, et al. "On the convergence analysis of muon." arXiv preprint arXiv:2505.23737 (2025).

[2] Sfyraki, Maria-Eleni, and Jun-Kun Wang. "Lions and muons: Optimization via stochastic frank-wolfe." arXiv preprint arXiv:2506.04192 (2025).

**Weaknesses:**

- The most concerning point is that the proposed Muon-MVR2 variant requires 2 gradient evaluations per step. This practically introduces almost double the amount of computational overhead.

- This practical-vs-theory tradeoff is shown in the paper's own experiments. The authors transparently note that the overhead can be prohibitive, especially visible in Fig 1C where the theoretically slower MVR1 is actually faster in practice per wall-clock time. Since the more practically usable MVR1 variant is stuck at the suboptimal $O(T^{-1/4})$ convergence rate, this means most people will likely use this lower-cost variant. Making the implications of the paper's main headline results less impactful.

- Experiments are small-scale. CIFAR-10 is an outdated and small dataset, but more critically LLaMA2-130M is only trained for 10,000 steps. At this very small scale, it is unclear how the benefits outweigh the costs of Muon-MVR2. The variance reduction technique also introduces new hyperparameters ($\gamma$), which becomes another expensive step to tune in any practical setting.

**Questions:**

- When is Muon-MVR2 the better choice? Fig1 A/B seems to show it's better per-epoch, but it appears slower than Muon-MVR1 in wall-clock time on Fig 1C. Is there a point after N number of epochs where its faster convergence overcomes its 2x per-step cost? A larger experiment is needed to examine this possibility.

- How well does Muon-MVR2 compare against variance-reduced optimizers? Such as SPIDER[3] and SVRG?

- The theoretical contributions of the paper largely depend on perfect orthogonalization. But in practice this is an approximation (Newton-Schultz) for efficiency. Does using the approximation break these theoretical guarantees?

[3] Fang, Cong, et al. "Spider: Near-optimal non-convex optimization via stochastic path-integrated differential estimator." Advances in neural information processing systems 31 (2018).

[4] Gower, Robert M., et al. "Variance-reduced methods for machine learning." Proceedings of the IEEE 108.11 (2020): 1968-1983.

---

> ### Author Response · Authors · 2025-11-19
>
> We thank the reviewers for their constructive comments. We have addressed these issues point-by-point below.
>
> ---
>
> **Q1.** The most concerning point is that the proposed Muon-MVR2 variant requires 2 gradient evaluations per step. This practically introduces almost double the amount of computational overhead.
>
> **A1**. Based on the per-iteration computational cost and the (theoretical) total complexities of MVR1 and MVR2, we note the following:
>
> - **Choice under different accuracy requirements.** When a high-accuracy target $\epsilon$ is required, **MVR2** is recommended because its iteration complexity, $O(\epsilon^{-3})$, is asymptotically better than the $O(\epsilon^{-4})$ complexity of **MVR1**. Conversely, for moderate accuracy requirements, **MVR1** may be preferable: although its iteration bound is weaker, each update is more economical, relying on only a single stochastic gradient evaluation.
>
> - **A concrete example.**  Assume the iteration complexities of **MVR1** and **MVR2** are exactly $1\times \epsilon^{-4}$ and $1\times \epsilon^{-3}$, respectively. Consider a target accuracy of $\epsilon = 1000^{-1/3}$. At this accuracy level, **MVR2** requires roughly **1,000 iterations**, whereas **MVR1** needs about **10,000 iterations** to achieve the same precision. Although each MVR2 update incurs two gradient evaluations, this extra per-step cost is well-justified: its faster convergence reduces the total number of iterations by an order of magnitude. Consequently, the overall computational workload is still lower for MVR2.
>
>
> ---
>
> **Q2.** This practical-vs-theory tradeoff is shown in the paper's own experiments. The authors transparently note that the overhead can be prohibitive, especially visible in Fig 1C where the theoretically slower MVR1 is actually faster in practice per wall-clock time. Since the more practically usable MVR1 variant is stuck at the suboptimal $O(T^{-1/4})$ convergence rate, this means most people will likely use this lower-cost variant. Making the implications of the paper's main headline results less impactful.
>
> **A2.**  We agree that this reflects a standard tradeoff.
>
> 1. Our paper analyzes the iteration complexities of two Muon variants—**Muon-MVR1** and **Muon-MVR2**—and shows that neither method universally dominates the other.
>
> 2. **Muon-MVR2** attains the optimal $T^{-1/3}$ rate using two gradient evaluations per step, while **Muon-MVR1** uses only one and thus trades its slower $O(T^{-1/4})$ rate for efficiency. As Figs. 1C and 2C show, this often makes Muon-MVR1 faster in wall-clock time despite weaker asymptotics.
>
> 3. Theoretically, providing the first method with a $T^{-1/3}$ optimal rate is intrinsically valuable.
>
> 4. Empirically, we recommend **Muon-MVR2** for high-accuracy settings and **Muon-MVR1** for lower-accuracy or compute-constrained regimes.
>
> ---
>
> **Q3.** Experiments are small-scale. CIFAR-10 is an outdated and small dataset, but more critically LLaMA2-130M is only trained for 10,000 steps. At this very small scale, it is unclear how the benefits outweigh the costs of Muon-MVR2. The variance reduction technique also introduces new hyperparameters ($\gamma$), which becomes another expensive step to tune in any practical setting.
>
> **A3.** To address this, we significantly enhanced our experiments in the revision: we trained AdamW, Muon, MARS-AdamW, Muon-MVR1, and Muon-MVR2 on LLaMA2-130M/C4 for **12B tokens** ($\approx 4\times$ Chinchilla optimal) with systematic tuning, reporting loss vs. wall-clock time to reflect performance under fixed budgets.
>
> Regarding the hyperparameter $\gamma$, joint tuning reveals that performance is relatively insensitive to $\gamma$ near the optimal learning rate. We provide a stable default setting, keeping extra tuning costs limited.
>
> Consistent with our previous statement, we recommend Muon-MVR1 for efficiency, and Muon-MVR2 when high accuracy is required or the extra batch cost is negligible. The corresponding changes are marked in blue in lines 426–431 and shown in Figure 3b.
>
> ---

---

> ### Author Response · Authors · 2025-11-19
>
> **Q4**. When is Muon-MVR2 the better choice? Fig1 A/B seems to show it's better per-epoch, but it appears slower than Muon-MVR1 in wall-clock time on Fig 1C. Is there a point after N number of epochs where its faster convergence overcomes its 2x per-step cost? A larger experiment is needed to examine this possibility.
>
> **A4.**  While theoretically optimal, Muon-MVR2 is $2\times$ slower per iteration than Muon-MVR1 in our LLaMA2-130M experiments, often yielding comparable initial precision. This highlights a practical limitation of exact variance reduction.
>
> We analyze this using the average gradient norm: $\mathcal{G}(T):= \frac{1}{T}\sum_{t=1}^T\mathbb{E}||\nabla f(\mathbf{X}_{t})||_F$ . With theoretical rates $$\mathcal{G} _ {MVR1} \propto T^{-1/4},~\mathcal{G} _ {MVR2} \propto T^{-1/3},$$ and assuming a fixed budget $B$ where Muon-MVR1 runs $T$ steps and Muon-MVR2 runs $T/2$ steps, we compare: $$\mathcal{G} _ {MVR1}(T) \lesssim \frac{A _ 1 \log T}{T^{1/4}},\mathcal{G} _ {MVR2}(T) \lesssim A _ 2 2^{1/3} \frac{\log(T/2)}{T^{1/3}}.$$ Because $T^{-1/3}$ decays faster than $T^{-1/4}$, there exists a threshold $T _ 0$. When $T \ge T _ 0$ (sufficiently large budget), Muon-MVR2 dominates ($\mathcal{G} _ {MVR2} < \mathcal{G} _ {MVR1}$) despite the $2\times$ overhead. For limited budgets $T < T _ 0,$ the cheaper iterations of Muon-MVR1 provide better efficiency.
>
> ---
> **Q5.** How well does Muon-MVR2 compare against variance-reduced optimizers? Such as SPIDER[3] and SVRG?
>
> **A5.** Muon uses **matrix preconditioning** to mitigate ill-conditioning and scale mismatch, whereas SPIDER and SVRG apply variance reduction to **un-preconditioned** SGD. Lacking the stability and convergence benefits of preconditioning, SPIDER and SVRG are therefore expected to perform markedly worse than Muon-MVR2 in practice.
>
> ---
>
> **Q6.** The theoretical contributions of the paper largely depend on perfect orthogonalization. But in practice this is an approximation (Newton-Schultz) for efficiency. Does using the approximation break these theoretical guarantees?
>
> **A6.** Our analysis assumes ideal “perfect orthogonalization,” while the implementation uses a 5-step Newton–Schulz approximation.
>
> Empirically, this scheme is numerically stable and closely matches exact orthogonalization, with no observed performance loss.
>
> A rigorous treatment of the induced approximation error is beyond our current scope. We have added a ''Limitations and Future Work'' section (lines 528–536) and incorporated this discussion as a direction for future research.
>
> ---

---

### Author Response · Authors · 2025-11-30
**Author Response Summary**

Dear Area Chair and Reviewers,

We thank the reviewers (Xhb7, RVgi, twoR, XfDc, DRyM) for recognizing our contribution in providing a **systematic theoretical analysis of Muon-style optimizers with momentum variance reduction (MVR)** and establishing the optimal $\tilde{\mathcal{O}}(T^{-1/3})$ rate. We have incorporated your feedback to strengthen both theory and experiments.

**1.Practical overhead and usage regimes (Reviewers Xhb7, RVgi, DRyM)**

Addressing concerns about Muon-MVR2’s $2\times$ cost, we clarified the usage regimes and expanded LLaMA2-130M/C4 experiments (12B tokens) to explicitly trade off theoretical optimality against wall-clock efficiency:

-   **Muon-MVR2:** Recommended for high-accuracy budgets where the optimal convergence rate justifies the extra cost.
-   **Muon-MVR1:** Recommended for most standard scenarios; it serves as an efficient default that is often faster in wall-clock time while maintaining competitive performance.
-   **Validation:** We now report loss vs. wall-clock time, confirming these trade-offs. We also show performance is robust to $\gamma$ near optimal learning rates.

**2.Experimental scale and baselines (Reviewers Xhb7, RVgi, twoR, XfDc)**

We addressed concerns regarding small-scale runs and missing baselines by:

-   **Scaling Up:** Extended LLaMA2-130M/C4 training to 12B tokens ($\approx$4× Chinchilla-optimal) with full wall-clock comparisons (updated Fig. 2C/3b).
-   **Baselines & Details:** Added MARS-AdamW to fairly compare VR-type optimizers and documented comprehensive hyperparameter sweeps (LR and $\gamma$).
-   **Scope:** Clarified that ResNet/CIFAR are controlled testbeds. We explicitly list scaling to ImageNet/CLIP-style models as future work.

**3.Theoretical novelty and positioning (Reviewers twoR, DRyM, XfDc)**

We significantly refined the related work and claims:

-   **Differentiation:**  We explicitly contrast our work with normalized SGDM and prior Muon analyses (step-size schedules, PL-type results, and complexity guarantees), and we distinguish our setting from Sfyraki & Wang (2025) based on differences in assumptions, algorithmic form, and convergence metrics.
-   **Contribution:** Novelty lies in combining MVR with Muon’s orthogonalization. We prove constant mini-batch sizes suffice for optimal $\tilde{\mathcal{O}}(\varepsilon^{-3})$ complexity.
-   **MVR1 Analysis:** We provide the first convergence analysis for inexact Muon-MVR $\tilde{\mathcal{O}}(T^{-1/4})$, formally characterizing the gap to MVR2.
-   **Corrections:** We softened claims to align with best-known rates, corrected the Nesterov-equivalence formula, and expanded on how orthogonalization aids geometry in ill-conditioned settings.

**4.Summary**

**The revised manuscript (with changes highlighted in blue)** clarifies the theoretical-practical trade-offs, strengthens empirical rigor with wall-clock metrics on larger runs, and transparently positions our work against prior art. We once again sincerely thank the reviewers and the Area Chair for their diligent efforts, and we remain confident in the theoretical and practical contributions of this paper.

---

### Meta-Review · Area_Chair_Ptrk · 2026-01-08

**Summary:**

Reviewers generally agree the paper provides a careful convergence analysis of Muon and two MVR variants, with a clear theoretical highlight for MVR2 and additional PL last-iterate results. The recommended rejection is mainly driven by:
- (i) limited practical impact of the headline result because MVR2 requires two gradient evaluations per step and often underperforms in wall-clock time, with unclear regimes where it reliably wins;
- (ii) empirical evidence that remains narrow relative to the claims (limited model/dataset coverage and lack of diagnostic scaling/ablation studies);
- (iii) theory–practice mismatch, since the theory assumes exact orthogonalization while implementations use approximations and experiments include ingredients (e.g., weight decay) not covered by the analysis;
- and (iv) ongoing concerns about incremental novelty and positioning versus closely related work.

**Reviewer Concerns:**

Addressed by the rebuttal: clearer related-work positioning and added citations; softened overstated novelty language; fixes to equation/notation issues and clarification of the “Nesterov” wording; more experimental details and broader hyperparameter searches; added wall-clock reporting; longer LLaMA2-130M/C4 experiments and inclusion of MARS-AdamW as a baseline.

Still outstanding: no rigorous treatment of approximate orthogonalization and its effect on guarantees; incomplete theoretical coverage of practical training components used in experiments (notably weight decay); limited evidence that MVR2’s improved iteration complexity translates into wall-clock advantages under realistic budgets; mostly qualitative (rather than aligned) comparisons to classical VR methods (e.g., SPIDER/SVRG); and empirical breadth still too limited (single small LLM scale, outdated vision setup) to substantiate general claims.

**Reviewer Scores:**

Xhb7 (4): likely to 6, acknowledging improved experiments and clearer guidance, but still concerned about wall-clock practicality and approximation gaps.

RVgi (6): likely unchanged at 6; wall-clock additions help, but limited breadth remains.

twoR (4): likely to 6; many concrete requests were addressed, though novelty and practical value concerns persist.

XfDc (6): likely unchanged at 6; technical fixes help, but diagnostics/scaling and theory–practice alignment remain weak.

DRyM (2): likely to 3 at most; clarifications reduce ambiguity, but core objections (comparability/claims and theory–practice mismatch) remain.

---

### Decision · Program_Chairs · 2026-01-26

Reject